# Neuron Activation Coverage: Rethinking Out-of-distribution Detection and Generalization

**Yibing Liu[1], Chris Xing Tian[1], Haoliang Li[1,*], Lei Ma[2,3], Shiqi Wang[1]**
City University of Hong Kong[1]   The University of Tokyo[2]   University of Alberta[3]
lyibing112@gmail.com,xingtian4-c@my.cityu.edu.hk
{haoliang.li,shiqiwang}@cityu.edu.hk,ma.lei@acm.org

## Abstract

The out-of-distribution (OOD) problem generally arises when neural networks encounter data that significantly deviates from the training data distribution, *i.e.*, in-distribution (InD). In this paper, we study the OOD problem from a neuron activation view. We first formulate neuron activation states by considering both the neuron output and its influence on model decisions. Then, to characterize the relationship between neurons and OOD issues, we introduce the *neuron activation coverage* (NAC) – a simple measure for neuron behaviors under InD data. Leveraging our NAC, we show that 1) InD and OOD inputs can be largely separated based on the neuron behavior, which significantly eases the OOD detection problem and beats the 21 previous methods over three benchmarks (CIFAR-10, CIFAR-100, and ImageNet-1K). 2) a positive correlation between NAC and model generalization ability consistently holds across architectures and datasets, which enables a NAC-based criterion for evaluating model robustness. Compared to prevalent InD validation criteria, we show that NAC not only can select more robust models, but also has a stronger correlation with OOD test performance. Our code is available at: https://github.com/BierOne/ood_coverage.

## 1 Introduction

Recent advances in machine learning systems hinge on an implicit assumption that the training and test data share the same distribution, known as in-distribution (InD) (Dosovitskiy et al., 2021; Szegedy et al., 2015; He et al., 2016; Simonyan & Zisserman, 2015). However, this assumption rarely holds in real-world scenarios due to the presence of out-of-distribution (OOD) data, *e.g.*, samples from unseen classes (Blanchard et al., 2011). Such distribution shifts between OOD and InD often drastically challenge well-trained models, leading to significant performance drops (Recht et al., 2019; D'Amour et al., 2020).

Prior efforts tackling this OOD problem mainly arise from two avenues: 1) OOD detection and 2) OOD generalization. The former one targets at designing tools that differentiate between InD and OOD data inputs, thereby refraining from using unreliable model predictions (Hendrycks & Gimpel, 2017; Liang et al., 2018; Liu et al., 2020; Huang et al., 2021b). In contrast, OOD generalization focuses on developing robust networks to generalize unseen OOD data, relying solely on InD data for training (Blanchard et al., 2011; Sun & Saenko, 2016; Sagawa et al., 2020; Kim et al., 2021; Shi et al., 2022). Despite the emergence of numerous studies, it is shown that existing approaches are still arguable to provide insights into the fundamental cause and mitigation of OOD issues (Sun et al., 2021; Gulrajani & Lopez-Paz, 2021).

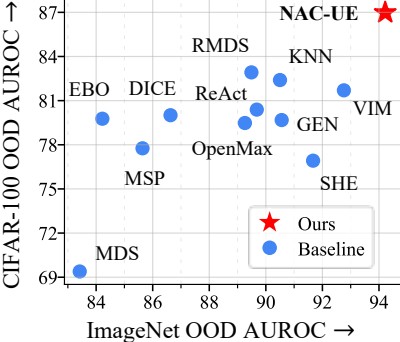

Figure 1: OOD detection performance on CIFAR-100 and ImageNet. AUROC scores (%) are averaged over the OOD datasets and backbones.

As suggested by Sun et al. (2021); Ahn et al. (2023), neurons could exhibit distinct activation patterns when exposed to data inputs from InD and OOD (See Figure 4). This reveals the potential of leveraging neuron behavior to characterize model status in terms of the OOD problem. Yet, though several studies recognize this significance, they either choose to modify neural networks (Sun et al., 2021), or lack the suitable definition of neuron activation states (Ahn et al., 2023; Tian et al., 2023).

---

*Corresponding author.

For instance, Sun et al. (2021) proposes a neuron truncation strategy that clips neuron output to separate the InD and OOD data, improving OOD detection. However, such truncation unexpectedly decrease the model classification ability (Djurisic et al., 2023)[1]. More recently, Ahn et al. (2023) and Tian et al. (2023) employ a threshold to characterize neurons into binary states (*i.e.*, activated or not) based on the neuron output. This characterization, however, discards valuable neuron distribution details. Unlike them, in this paper, we show that by leveraging natural neuron activation states, a simple statistical property of neuron distribution could effectively facilitate the OOD solutions.

We first propose to formulate the neuron activation state by considering both the neuron output and its influence on model decisions. Specifically, inspired by Huang et al. (2021b), we model neuron influence as the gradients derived from Kullback-Leibler (KL) divergence (Kullback & Leibler, 1951) between network output and a uniform vector. Then, to characterize the relationship between neuron behavior and OOD issues, we draw insights from coverage analysis in system testing (Pei et al., 2017; Ma et al., 2018), which reveals that *rarely-activated (covered) neurons by a training set can potentially trigger undetected bugs, such as misclassifications, during the test stage*. In this sense, we introduce the concept of *neuron activation coverage* (NAC), which quantifies the coverage degree of neuron states under InD training data (See Figure 2). In particular, if a neuron state is frequently activated by InD training inputs, NAC would assign it with a higher coverage score, indicating fewer underlying defects in this state. This paper applies NAC to two OOD tasks:

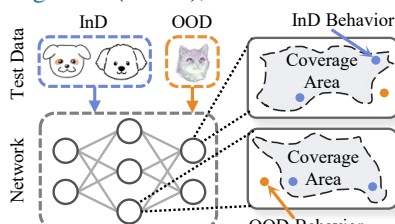

Figure 2: NAC models *coverage area* in neuron activation space using InD training data. Upon receiving OOD data, neurons tend to behave outside the expected coverage area, thus with lower coverage scores.

**OOD detection.** Since OOD data potentially trigger abnormal neuron activations, they should present smaller coverage scores compared to the InD test data (Figure 2). As such, we present **NAC** for **U**ncertainty **E**stimation (`NAC-UE`), which directly averages coverage scores over all neurons as data uncertainty. We evaluate `NAC-UE` over three benchmarks (CIFAR-10, CIFAR-100, and ImageNet-1k), establishing new state-of-the-art performance over the 21 previous best OOD detection methods. Notably, our `NAC-UE` achieves a 10.60% improvement on FPR95 (with a 4.58% gain on AUROC) over CIFAR-100 compared to the competitive ViM (Wang et al., 2022) (See Figure 1).

**OOD generalization.** Given that underlying defects can exist outside the coverage area (Pei et al., 2017), we hypothesize that the robustness of the network increases with a larger coverage area. To this end, we employ **NAC** for **M**odel **E**valuation (`NAC-ME`), which measures model robustness by integrating the coverage distribution of all neurons. Through experiments on DomainBed (Gulrajani & Lopez-Paz, 2021), we find that a positive correlation between NAC and model generalization ability consistently holds across architectures and datasets. Moreover, compared to InD validation criteria, `NAC-ME` not only selects more robust models, but also exhibits stronger correlation with OOD test performance. For instance, on the Vit-b16 (Dosovitskiy et al., 2021), `NAC-ME` outperforms validation criteria by 11.61% in terms of rank correlation with OOD test accuracy.

## 2 NAC: NEURON ACTIVATION COVERAGE

This paper studies OOD problems in multi-class classification, where $\mathcal{D} = \mathbb{R}^d$ denotes the input space and $\mathcal{Y} = \{1, 2, ..., C\}$ is the output space. Let $X = \{(\mathbf{x}_i, y_i)\}_{i=1}^n$ be the training set, comprising *i.i.d.* samples from the joint distribution $\mathcal{P} = \mathcal{X} \times \mathcal{Y}$. A neural network parameterized by $\theta$, $F(\mathbf{x}; \theta) : \mathcal{X} \to \mathbb{R}^{|\mathcal{Y}|}$, is trained on samples drawn from $\mathcal{P}$, producing a logit vector for classification. We illustrate our NAC-based approaches in Figure 3. In the following, we first formulate the neuron activation state (Section 2.1), and then introduce the details of our NAC (Section 2.2). We finally show how to apply NAC to two OOD problems (Section 2.3): OOD detection and generalization.

### 2.1 FORMULATION OF NEURON ACTIVATION STATE

Neuron outputs generally depend on the propagation from network input to the layer where the neuron resides. However, this does not consider the neuron influence in subsequent propagations.

---

[1]While it may be argued that maintaining neuron outputs for double-propagation preserves InD accuracy with low computational cost, it relies on the assumption that only later layers are utilized in neuron pruning, thus undermining the potential of these neuron-based methods.

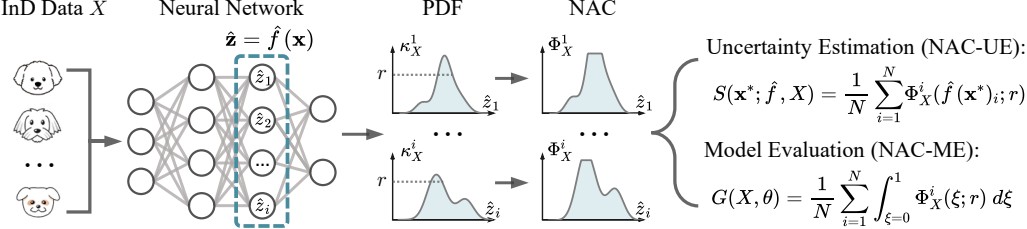

Figure 3: Illustration of our NAC-based methods. NAC is derived from the probability density function (PDF), which quantifies the coverage degree of neuron states under the InD training set $X$. Building upon NAC, we devise two approaches for tackling different OOD problems: OOD Detection (`NAC-UE`) and OOD Generalization (`NAC-ME`).

As such, we introduce gradients backpropagated from the KL divergence between network output and a uniform vector (Huang et al., 2021b), to model the neuron influence. Formally, we denote by $f(\mathbf{x}) = \mathbf{z} \in \mathbb{R}^N$ the output vector of a specific layer (Section 3.1 discusses this layer choice), where $N$ is the number of neurons and $z_i$ is the raw output of $i$-th neuron in this layer. By setting the uniform vector $\mathbf{u} = [1/C, 1/C, ..., 1/C] \in \mathbb{R}^C$, the desired KL divergence can be given as:

$$D_{\mathrm{KL}}(\mathbf{u}||\mathbf{p}) = \sum_{i=1}^{C} u_i \log \frac{u_i}{p_i} = -\sum_{i=1}^{C} u_i \log p_i - H(\mathbf{u}), \tag{1}$$

where $\mathbf{p} = \mathrm{softmax}(F(\mathbf{x}))$, and $p_i$ denotes $i$-element in $\mathbf{p}$. $H(\mathbf{u}) = -\sum_{i=1}^{C} u_i \log u_i$ is a constant. By combining the KL gradient with neuron output, we then formulate *neuron activation state* as,

$$\hat{\mathbf{z}} = \sigma(\mathbf{z} \odot \frac{\partial D_{\mathrm{KL}}(\mathbf{u}||\mathbf{p})}{\partial \mathbf{z}}), \tag{2}$$

where $\sigma(x) = 1/(1 + e^{-\alpha x})$ is the sigmoid function with a steepness controller $\alpha$. In the rest of this paper, we will also use the notation $\hat{f}(\mathbf{x}) := \hat{\mathbf{z}}$ to represent the neuron state function.

**Rationale of $\hat{\mathbf{z}}$.** Here, we further analyze the gradients from KL divergence to show how this part contributes to the neuron activation state $\hat{\mathbf{z}}$. Without loss of generality, let the network be $F = f \circ g$, where $g(\cdot)$ is the predictor following $\mathbf{z}$. Since $\partial D_{\mathrm{KL}}/\partial \mathbf{g}(\mathbf{z}) = \mathbf{p} - \mathbf{u}$, we can rewrite the Eq. (2) as follows (more details are provided in Appendix B):

$$\hat{\mathbf{z}} = \sigma(\mathbf{z} \odot \frac{\partial D_{\mathrm{KL}}}{\partial \mathbf{z}}) = \sigma(\mathbf{z} \odot (\frac{\partial g(\mathbf{z})}{\partial \mathbf{z}} \cdot \frac{\partial D_{\mathrm{KL}}}{\partial g(\mathbf{z})})) = \sigma\Big(\sum_{i=1}^{C}(\mathbf{z} \odot \frac{\partial g(\mathbf{z})_i}{\partial \mathbf{z}}) \cdot (p_i - u_i)\Big), \tag{3}$$

where (1) $\mathbf{z} \odot (\partial g(\mathbf{z})_i/\partial \mathbf{z})$ corresponds the simple explanation method known as *Input $\odot$ Gradient* (Ancona et al., 2018), which quantifies the contribution of neurons to the model prediction $g(\mathbf{z})_i$. It is also the general form of many prevalent explanation methods, such as $\epsilon$-LRP (Bach et al., 2015), DeepLIFT (Shrikumar et al., 2017), and IG (Sundararajan et al., 2017); (2) $p_i - u_i$ measures the deviation of model predictions from a uniform distribution, thus denoting sample confidence (Huang et al., 2021b). In this way, we builds $\hat{\mathbf{z}}$ by considering both the significance of neurons on model predictions, and model confidence in input data. Intuitively, if a neuron contributes less to the output (or the model lacks confidence in input data), the neuron would be considered less active.

## 2.2 NEURON ACTIVATION COVERAGE (NAC)

With the formulation of neuron activation state, we now introduce the *neuron activation coverage* (NAC) to characterize neuron behaviors under InD and OOD data. Inspired by system testing (Pei et al., 2017; Ma et al., 2018; Xie et al., 2019), NAC aims to quantify the coverage degree of neuron states under InD training data. The intuition is that *if a neuron state is rarely activated (covered) by any InD input, the chances of triggering bugs (e.g., misclassification) under this state would be high.* Since NAC directly measures the statistical property (*i.e.*, coverage) over neuron state distribution, we derive the NAC function from the probability density function (PDF). Formally, given a state $\hat{z}_i$ of $i$-th neuron, and its PDF $\kappa_X^i(\cdot)$ over an InD set $X$, the function for NAC can be given as:

$$\Phi_X^i(\hat{z}_i; r) = \frac{1}{r}\min(\kappa_X^i(\hat{z}_i), r), \tag{4}$$

where $\kappa_X^i(\hat{z}_i)$ is the probability density of $\hat{z}_i$ over the set $X$, and $r$ denotes the lower bound for achieving full coverage *w.r.t.* state $\hat{z}_i$. In cases where the neuron state $\hat{z}_i$ is frequently activated by InD training data, the coverage score $\Phi_X^i(\hat{z}_i; r)$ would be 1, denoting fewer underlying defects in this state. Notably, if $r$ is too low, noisy activations would dominate the coverage, reducing the significance of coverage scores. Conversely, an excessively large value of $r$ also makes the NAC function vulnerable to data biases. For example, given a homogeneous dataset comprising numerous similar samples, the coverage score of a neuron state $\hat{z}_i$ can be mischaracterized as abnormally high, marginalizing the effects of other meaningful states. We analyze the effect of $r$ in Section 3.1.

## 2.3 APPLICATIONS

After modeling the NAC function over InD training data, we can directly apply it to tackle existing OOD problems. In the following, we illustrate two application scenarios.

**Uncertainty estimation for OOD detection.** Since OOD data often trigger abnormal neuron behaviors (See Figure 4), we employ **NAC** for **U**ncertainty **E**stimation (`NAC-UE`), which directly averages coverage scores over all neurons as the uncertainty of test samples. Formally, given a test data $\mathbf{x}^*$, the function for `NAC-UE` can be given as,

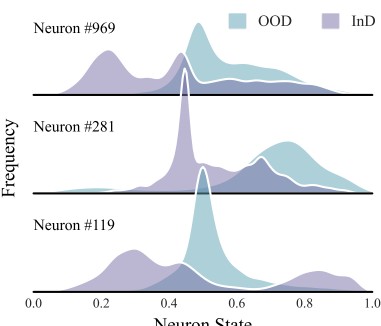

$$S(\mathbf{x}^*; \hat{f}, X) = \frac{1}{N} \sum_{i=1}^{N} \Phi_X^i(\hat{f}(\mathbf{x}^*)_i; r), \qquad (5)$$

where $N$ is the number of neurons; $\hat{f}(\mathbf{x}^*)_i := \hat{z}_i$ denotes the state of $i$-th neuron; $r$ is the controller of NAC function. If the neuron states triggered by $\mathbf{x}^*$ are frequently activated by InD training samples, the coverage score $S(\mathbf{x}^*; \hat{f}, X)$ would be high, suggesting that $\mathbf{x}^*$ is likely to come from InD distribution. By considering multiple layers in the network, we propose using `NAC-UE` for OOD detection following Liu et al. (2020); Huang et al. (2021b); Sun et al. (2021):

$$D(\mathbf{x}^*) = \begin{cases} \text{InD} & \text{if } \sum_l S(\mathbf{x}^*; \hat{f}_l, X) \geq \lambda; \\ \text{OOD} & \text{if } \sum_l S(\mathbf{x}^*; \hat{f}_l, X) < \lambda, \end{cases} \qquad (6)$$

Figure 4: OOD vs. InD neuron activation states. We employ PACS (Li et al., 2017) *Photo* domain as InD and *Sketch* as OOD. All neurons stem from the layer4 of ResNet-50.

where $\lambda$ is a threshold, and $\hat{f}_l$ denotes the neuron state function of layer $l$. The test sample with an uncertainty score $\sum_l S(\mathbf{x}^*; \hat{f}_l, X)$ less than $\lambda$ would be categorized as OOD; otherwise, InD.

**Model evaluation for OOD generalization.** OOD data potentially trigger neuron states beyond the coverage area of InD data (Figure 2 and Figure 4), thus leading to misclassifications. From this perspective, we hypothesize that the robustness of networks could positively correlate with the size of coverage area. For instance, as coverage area narrows, larger inactive space would remain, increasing the chances of triggering underlying bugs. Hence, we propose **NAC** for **M**odel **E**valuation (`NAC-ME`), which characterizes model generalization ability based on the integral of neuron coverage distribution. Formally, given an InD training set $X$, `NAC-ME` measures the generalization ability of a model (parameterized by $\theta$) as the average of integral *w.r.t.* NAC distribution:

$$G(X, \theta) = \frac{1}{N} \sum_{i=1}^{N} \int_{\xi=0}^{1} \Phi_X^i(\xi; r) \, d\xi, \qquad (7)$$

where $N$ is the number of neurons, and $r$ is the controller of NAC function. Given the training set $X$, if a neuron is consistently active throughout the activation space, we consider it to be well exercised by InD training data, thus with a lower probability of triggering bugs, i.e., favorable robustness.

**Approximation.** To enable efficient processing of large-scale datasets, we adopt a simple histogram-based approach for modeling the probability density function (PDF) function. This approach divides the neuron activation space into $M$ intervals, and naturally supports mini-batch approximation. We provide more details in Appendix C. In addition, we efficiently calculate $G(X, \theta)$ using the Riemman approximation (Krantz, 2005),

$$G(X, \theta) = \frac{1}{MN} \sum_{i=1}^{N} \sum_{k=1}^{M} \Phi_X^i\left(\frac{k}{M}; r\right). \qquad (8)$$

| Method | MNIST | | SVHN | | Textures | | Places365 | | Average | |
|---|---|---|---|---|---|---|---|---|---|---|
| | FPR95↓ | AUROC↑ | FPR95↓ | AUROC↑ | FPR95↓ | AUROC↑ | FPR95↓ | AUROC↑ | FPR95↓ | AUROC↑ |
| *CIFAR-10 Benchmark* | | | | | | | | | | |
| OpenMax | $23.33_{\pm4.67}$ | $90.50_{\pm0.44}$ | $25.40_{\pm1.47}$ | $89.77_{\pm0.45}$ | $31.50_{\pm4.05}$ | $89.58_{\pm0.60}$ | $38.52_{\pm2.27}$ | $88.63_{\pm0.28}$ | $29.69_{\pm1.21}$ | $89.62_{\pm0.19}$ |
| ODIN | $23.83_{\pm12.34}$ | $95.24_{\pm1.96}$ | $68.61_{\pm0.52}$ | $84.58_{\pm0.77}$ | $67.70_{\pm11.06}$ | $86.94_{\pm2.26}$ | $70.36_{\pm6.96}$ | $85.07_{\pm1.24}$ | $57.62_{\pm4.24}$ | $87.96_{\pm0.61}$ |
| MDS | $27.30_{\pm3.55}$ | $90.10_{\pm2.41}$ | $25.96_{\pm2.52}$ | $91.18_{\pm0.47}$ | $27.94_{\pm4.20}$ | $92.69_{\pm1.06}$ | $47.67_{\pm4.54}$ | $84.90_{\pm2.54}$ | $32.22_{\pm3.40}$ | $89.72_{\pm1.36}$ |
| MDSEns | $1.30_{\pm0.51}$ | $99.17_{\pm0.41}$ | $74.34_{\pm1.04}$ | $66.56_{\pm0.58}$ | $76.07_{\pm0.17}$ | $77.40_{\pm0.28}$ | $94.16_{\pm0.33}$ | $52.47_{\pm0.15}$ | $61.47_{\pm0.48}$ | $73.90_{\pm0.27}$ |
| RMDS | $21.49_{\pm2.32}$ | $93.22_{\pm0.80}$ | $23.46_{\pm1.48}$ | $91.84_{\pm0.26}$ | $25.25_{\pm0.53}$ | $92.23_{\pm0.23}$ | $31.20_{\pm0.28}$ | $91.51_{\pm0.11}$ | $25.35_{\pm0.73}$ | $92.20_{\pm0.21}$ |
| Gram | $70.30_{\pm8.96}$ | $72.64_{\pm2.34}$ | $33.91_{\pm17.35}$ | $91.52_{\pm4.45}$ | $94.64_{\pm2.71}$ | $62.34_{\pm8.27}$ | $90.49_{\pm1.93}$ | $60.44_{\pm3.41}$ | $72.34_{\pm6.73}$ | $71.73_{\pm3.20}$ |
| ReAct | $33.77_{\pm18.00}$ | $92.81_{\pm3.03}$ | $50.23_{\pm15.98}$ | $89.12_{\pm3.19}$ | $51.42_{\pm11.42}$ | $89.38_{\pm1.49}$ | $44.20_{\pm3.35}$ | $90.35_{\pm0.78}$ | $44.90_{\pm8.37}$ | $90.42_{\pm1.41}$ |
| VIM | $18.36_{\pm1.42}$ | $94.76_{\pm0.38}$ | $19.29_{\pm0.41}$ | $94.50_{\pm0.48}$ | $21.14_{\pm1.83}$ | $95.15_{\pm0.34}$ | $41.43_{\pm2.17}$ | $89.49_{\pm0.39}$ | $25.05_{\pm0.52}$ | $93.48_{\pm0.24}$ |
| KNN | $20.05_{\pm1.36}$ | $94.26_{\pm0.38}$ | $22.60_{\pm1.26}$ | $92.67_{\pm0.30}$ | $24.06_{\pm0.55}$ | $93.16_{\pm0.24}$ | $30.38_{\pm0.63}$ | $91.77_{\pm0.23}$ | $24.27_{\pm0.40}$ | $92.96_{\pm0.14}$ |
| ASH | $70.00_{\pm10.56}$ | $83.16_{\pm4.66}$ | $83.64_{\pm6.48}$ | $73.46_{\pm6.41}$ | $84.59_{\pm1.74}$ | $77.45_{\pm2.39}$ | $77.89_{\pm7.28}$ | $79.89_{\pm3.69}$ | $79.03_{\pm4.22}$ | $78.49_{\pm2.58}$ |
| SHE | $42.22_{\pm20.59}$ | $90.43_{\pm4.76}$ | $62.74_{\pm4.01}$ | $86.38_{\pm1.32}$ | $84.60_{\pm5.30}$ | $81.57_{\pm1.21}$ | $76.36_{\pm5.32}$ | $82.89_{\pm1.22}$ | $66.48_{\pm5.98}$ | $85.32_{\pm1.43}$ |
| GEN | $23.00_{\pm7.75}$ | $93.83_{\pm2.14}$ | $28.14_{\pm2.59}$ | $91.97_{\pm0.66}$ | $40.74_{\pm6.61}$ | $90.14_{\pm0.76}$ | $47.03_{\pm3.22}$ | $89.46_{\pm0.65}$ | $34.73_{\pm1.58}$ | $91.35_{\pm0.69}$ |
| **NAC-UE** | $15.14_{\pm2.60}$ | $94.86_{\pm1.36}$ | $14.33_{\pm1.24}$ | $96.05_{\pm0.47}$ | $17.03_{\pm0.59}$ | $95.64_{\pm0.44}$ | $26.73_{\pm0.80}$ | $91.85_{\pm0.28}$ | $18.31_{\pm0.92}$ | $94.60_{\pm0.50}$ |
| *CIFAR-100 Benchmark* | | | | | | | | | | |
| OpenMax | $53.82_{\pm4.74}$ | $76.01_{\pm1.39}$ | $53.20_{\pm1.78}$ | $82.07_{\pm1.53}$ | $56.12_{\pm1.91}$ | $80.56_{\pm0.09}$ | $54.85_{\pm1.42}$ | $79.29_{\pm0.40}$ | $54.50_{\pm0.68}$ | $79.48_{\pm0.41}$ |
| ODIN | $45.94_{\pm3.29}$ | $83.79_{\pm1.31}$ | $67.41_{\pm3.88}$ | $74.54_{\pm0.76}$ | $62.37_{\pm2.96}$ | $79.33_{\pm1.08}$ | $59.71_{\pm0.92}$ | $79.45_{\pm0.26}$ | $58.86_{\pm0.79}$ | $79.28_{\pm0.21}$ |
| MDS | $71.72_{\pm2.94}$ | $67.47_{\pm0.81}$ | $67.21_{\pm6.09}$ | $70.68_{\pm6.40}$ | $70.49_{\pm2.48}$ | $76.26_{\pm0.69}$ | $79.61_{\pm0.34}$ | $63.15_{\pm0.49}$ | $72.26_{\pm1.56}$ | $69.39_{\pm1.39}$ |
| MDSEns | $2.83_{\pm0.86}$ | $98.21_{\pm0.78}$ | $82.57_{\pm2.58}$ | $53.76_{\pm1.63}$ | $84.94_{\pm0.83}$ | $69.75_{\pm1.14}$ | $96.61_{\pm0.17}$ | $42.27_{\pm0.73}$ | $66.74_{\pm1.04}$ | $66.00_{\pm0.69}$ |
| RMDS | $52.05_{\pm6.28}$ | $79.74_{\pm2.49}$ | $51.65_{\pm3.68}$ | $84.89_{\pm1.10}$ | $53.99_{\pm1.06}$ | $83.65_{\pm0.51}$ | $53.57_{\pm0.43}$ | $83.40_{\pm0.46}$ | $52.81_{\pm0.63}$ | $82.92_{\pm0.42}$ |
| Gram | $53.53_{\pm7.45}$ | $80.71_{\pm4.15}$ | $20.06_{\pm1.96}$ | $95.55_{\pm0.60}$ | $89.51_{\pm2.54}$ | $70.79_{\pm1.32}$ | $94.67_{\pm0.60}$ | $46.38_{\pm1.21}$ | $64.44_{\pm2.37}$ | $73.36_{\pm1.08}$ |
| ReAct | $56.04_{\pm5.66}$ | $78.37_{\pm1.59}$ | $50.41_{\pm2.02}$ | $83.01_{\pm0.97}$ | $55.04_{\pm0.82}$ | $80.15_{\pm0.46}$ | $55.30_{\pm0.41}$ | $80.03_{\pm0.11}$ | $54.20_{\pm1.56}$ | $80.39_{\pm0.49}$ |
| VIM | $48.32_{\pm1.07}$ | $81.89_{\pm1.02}$ | $46.22_{\pm5.46}$ | $83.14_{\pm3.71}$ | $46.86_{\pm2.29}$ | $85.91_{\pm0.78}$ | $61.57_{\pm0.77}$ | $75.85_{\pm0.37}$ | $50.74_{\pm1.00}$ | $81.70_{\pm0.62}$ |
| KNN | $48.58_{\pm4.67}$ | $82.36_{\pm1.52}$ | $51.75_{\pm3.12}$ | $84.15_{\pm1.09}$ | $53.56_{\pm2.32}$ | $83.66_{\pm0.83}$ | $60.70_{\pm1.03}$ | $79.43_{\pm0.47}$ | $53.65_{\pm0.28}$ | $82.40_{\pm0.17}$ |
| ASH | $66.58_{\pm3.88}$ | $77.23_{\pm0.46}$ | $46.00_{\pm2.67}$ | $85.60_{\pm1.40}$ | $61.27_{\pm2.74}$ | $80.72_{\pm0.70}$ | $62.95_{\pm0.99}$ | $78.76_{\pm0.16}$ | $59.20_{\pm2.46}$ | $80.58_{\pm0.66}$ |
| SHE | $58.78_{\pm2.70}$ | $76.76_{\pm1.07}$ | $59.15_{\pm7.61}$ | $80.97_{\pm3.98}$ | $73.29_{\pm3.22}$ | $73.64_{\pm1.28}$ | $65.24_{\pm0.98}$ | $76.30_{\pm0.51}$ | $64.12_{\pm2.70}$ | $76.92_{\pm1.16}$ |
| GEN | $53.92_{\pm5.71}$ | $78.29_{\pm2.05}$ | $55.45_{\pm2.76}$ | $81.41_{\pm1.50}$ | $61.23_{\pm1.40}$ | $78.74_{\pm0.81}$ | $56.25_{\pm1.01}$ | $80.28_{\pm0.27}$ | $56.71_{\pm1.59}$ | $79.68_{\pm0.75}$ |
| **NAC-UE** | $21.97_{\pm6.62}$ | $93.15_{\pm1.63}$ | $24.39_{\pm4.66}$ | $92.40_{\pm1.26}$ | $40.65_{\pm1.94}$ | $89.32_{\pm0.55}$ | $73.57_{\pm1.16}$ | $73.05_{\pm0.68}$ | $40.14_{\pm1.86}$ | $86.98_{\pm0.37}$ |

Table 1: OOD detection performance on CIFAR-10 and CIFAR-100 benchmarks. We format **first**, second, and third results. Full results for all baselines are provided in Table 20 and Table 21.

## 3 EXPERIMENTS

### 3.1 CASE STUDY 1: OOD DETECTION

**Setup.** Our experimental settings align with the latest version of OpenOOD[2] (Yang et al., 2022; Zhang et al., 2023a). We evaluate our NAC-UE on three benchmarks: CIFAR-10, CIFAR-100, and ImageNet-1k. For CIFAR-10 and CIFAR-100, InD dataset corresponds to the respective CIFAR, and 4 OOD datasets are included: MNIST (Deng, 2012), SVHN (Netzer et al., 2011), Textures (Cimpoi et al., 2014), and Places365 (Zhou et al., 2018). For ImageNet experiments, ImageNet-1k serves as InD, along with 3 OOD datasets: iNaturalist (Horn et al., 2018), Textures (Cimpoi et al., 2014), and OpenImage-O (Wang et al., 2022). We use pretrained ResNet-50 and Vit-b16 for ImageNet experiments, and ResNet-18 for CIFAR. For all employed benchmarks, we compare our NAC-UE with 21 SoTA OOD detection methods. We provide more details in Appendix D.

**Metrics.** We utilize two threshold-free metrics in our evaluation: 1) FPR95: the false-positive-rate of OOD samples when the true positive rate of ID samples is at 95%; 2) AUROC: the area under the receiver operating characteristic curve. Throughout our implementations, all pretrained models are left unmodified, preserving their classification ability during the OOD detection phase.

**Implementation details.** We first build the NAC function using InD training data, utilizing 1,000 training images for ResNet-18 and ResNet-50, and 50,000 images for Vit-b16. Note that in this stage, we merely use training samples less than 5% of the training set (See Appendix G.1 for more analysis). Next, we employ NAC-UE to calculate uncertainty scores during the test phase. Following OpenOOD, we use the validation set to select hyperparameters and evaluate NAC-UE on the test set.

**Results.** Table 1 and Table 2 mainly illustrate our results on CIFAR and ImageNet benchmarks, where we compare NAC-UE with 21 SoTA methods. As can be seen, our NAC-UE consistently outperforms all of the SoTA methods on average performance, establishing record-breaking performance over 3 benchmarks. Specifically, NAC-UE reduces the FPR95 by 10.60% and 5.96% over the most competitive rival (Wang et al., 2022; Sun et al., 2022) on CIFAR-100 and CIFAR-10, respectively. On the large-scale ImageNet benchmark, NAC-UE also consistently improves AUROC scores across backbones and OOD datasets. Besides, since NAC-UE performs in a *post-hoc* fashion, it preserves model classification ability (*i.e.*, InD accuracy) during the OOD detection phase. In contrast, advanced methods such as ReAct (Sun et al., 2021) and ASH (Djurisic et al., 2023) exhibit promising OOD detection results at the expense of InD performance (Djurisic et al., 2023).

---

[2]https://github.com/Jingkang50/OpenOOD.

| Dataset | Backbone | OpenMax | MDS | RMDS | ReAct | VIM | KNN | ASH | SHE | GEN | **NAC-UE** |
|---------|----------|---------|-----|------|-------|-----|-----|-----|-----|-----|------------|
| iNaturalist | ResNet-50 | 92.05 | 63.67 | 87.24 | 96.34 | 89.56 | 86.41 | 97.07 | 92.65 | 92.44 | 96.52 |
| | Vit-b16 | 94.93 | 96.01 | 96.10 | 86.11 | 95.72 | 91.46 | 50.62 | 93.57 | 93.54 | 93.72 |
| | Average | 93.49 | 79.84 | 91.67 | 91.23 | 92.64 | 88.94 | 73.85 | 93.11 | 92.99 | **95.12** |
| OpenImage-O | ResNet-50 | 87.62 | 69.27 | 85.84 | 91.87 | 90.50 | 87.04 | 93.26 | 86.52 | 89.26 | 91.45 |
| | Vit-b16 | 87.36 | 92.38 | 92.32 | 84.29 | 92.18 | 89.86 | 55.51 | 91.04 | 90.27 | 91.58 |
| | Average | 87.49 | 80.83 | 89.08 | 88.08 | 91.34 | 88.45 | 74.39 | 88.78 | 89.77 | **91.52** |
| Textures | ResNet-50 | 88.10 | 89.80 | 86.08 | 92.79 | 97.97 | 97.09 | 96.90 | 93.60 | 87.59 | 97.9 |
| | Vit-b16 | 85.52 | 89.41 | 89.38 | 86.66 | 90.61 | 91.12 | 48.53 | 92.65 | 90.23 | 94.17 |
| | Average | 86.81 | 89.61 | 87.73 | 89.73 | 94.29 | 94.11 | 72.72 | 93.13 | 88.91 | **96.04** |

Table 2: OOD detection performance (AUROC↑) on ImageNet. See Table 22 for full results.

**NAC-UE with training methods.** Training-time regularization is one of the potential directions in OOD detection. Here, we further show that `NAC-UE` is pluggable to existing training methods. Table 3 illustrates our results using three training schemes: ConfBranch (DeVries & Taylor, 2018), RotPred (Hendrycks et al., 2019b), and GODIN (Hsu et al., 2020), where we compare NAC-UE with the detection method employed in the original paper, *i.e.*, *Baseline* in Table 3. Notably, `NAC-UE` significantly improves upon the baseline method across all three training approaches, which highlights its effectiveness for OOD detection again.

| Training | Method | FPR95↓ | AUROC↑ |
|----------|--------|--------|--------|
| ConfBranch | Baseline | 50.98 | 83.94 |
| | NAC-UE | **31.04** | **93.90** |
| RotPred | Baseline | 36.67 | 90.00 |
| | NAC-UE | **30.24** | **93.28** |
| GODIN | Baseline | 50.87 | 85.51 |
| | NAC-UE | **26.86** | **94.61** |

Table 3: ImageNet results of NAC-UE with different training methods.

**Where to apply NAC-UE?** Since `NAC-UE` performs based on neurons in a network, we further investigate its effect when using neurons from different layers. Table 4 exhibits the results, where the ResNet is utilized as the backbone for analysis. It can be drawn that (1) the performance of `NAC-UE` positively correlates with the number of employed layers. This is intuitive, as including more layers enables a greater number of neurons to be considered, thereby enhancing the accuracy of `NAC-UE` in estimating the model status; (2) even with a single layer of neurons, `NAC-UE` is able to achieve favorable performance. For instance, by employing *layer4*, `NAC-UE` already achieves 23.50% FPR95, which outperforms the previous best method KNN on CIFAR-10.

| Layer Combinations | | | | CIFAR-10 | | CIFAR-100 | | ImageNet | |
|--------|--------|--------|--------|---------|---------|---------|---------|---------|---------|
| Layer4 | Layer3 | Layer2 | Layer1 | FPR95↓ | AUROC↑ | FPR95↓ | AUROC↑ | FPR95↓ | AUROC↑ |
| ✓ | | | | 23.50 | 93.21 | 85.84 | 58.37 | 26.89 | 94.57 |
| ✓ | ✓ | | | 21.32 | 94.35 | 44.92 | 85.25 | 23.51 | 95.05 |
| ✓ | ✓ | ✓ | | 18.50 | 94.46 | **39.96** | 86.94 | 22.69 | 95.23 |
| ✓ | ✓ | ✓ | ✓ | **18.31** | **94.60** | 40.14 | **86.98** | **22.49** | **95.29** |

Table 4: Performance of NAC-UE with different layer choices.

**The superiority of neuron activation state $\hat{\mathbf{z}}$.** Section 2.1 formulates the neuron activation state $\hat{\mathbf{z}}$ by combining the neuron output $\mathbf{z}$ with its KL gradients $\partial D_{KL}/\partial \mathbf{z}$. Here, we ablate this formulation to examine the superiority of $\hat{\mathbf{z}}$. In particular, we analyze the neuron behaviors *w.r.t.* 1) raw neuron output: $\mathbf{z}$, 2) KL gradients of neuron output: $\partial D_{KL}/\partial \mathbf{z}$, and 3) ours neuron state: $\mathbf{z} \odot \partial D_{KL}/\partial \mathbf{z}$.

Figure 5 illustrates the results, where we visualize the InD and OOD distribution of different neurons in the ImageNet benchmark. As can be seen, under the form of $\mathbf{z} \odot \partial D_{KL}/\partial \mathbf{z}$, neurons tend to present distinct activation patterns when exposed to InD and OOD data. This distinctiveness greatly facilitates the separability between InD and OOD, thereby leading to the best OOD detection performance with `NAC-UE`, *e.g.*, 16.58% FPR95 ($\mathbf{z} \odot \partial D_{KL}/\partial \mathbf{z}$) *vs.* 35.72% FPR95 ($\mathbf{z}$) on *layer4*. Contrary to that, when considering the vanilla form of $\mathbf{z}$ and $\partial D_{KL}/\partial \mathbf{z}$, the neuron behaviors under InD and OOD are largely overlapped, which further spotlights the unique characteristic of our $\hat{\mathbf{z}}$. More detailed analysis can be found in Appendix G.2.

**Parameter analysis.** Table 5-7 presents a systematically analysis of the effect of sigmoid steepness ($\alpha$), lower bound ($r$) for full coverage, and the number of intervals ($M$) for PDF approximation. The following observations can be noted: 1) A relatively steep sigmoid function could make `NAC-UE` perform better. We conjecture this is due to that neuron activation states often distribute in a small range, thus requiring a steeper function to distinguish their finer variations; 2) `NAC-UE` is sensitive to the choice of $r$. As previously discussed, a small r would allows noisy activations to dominate

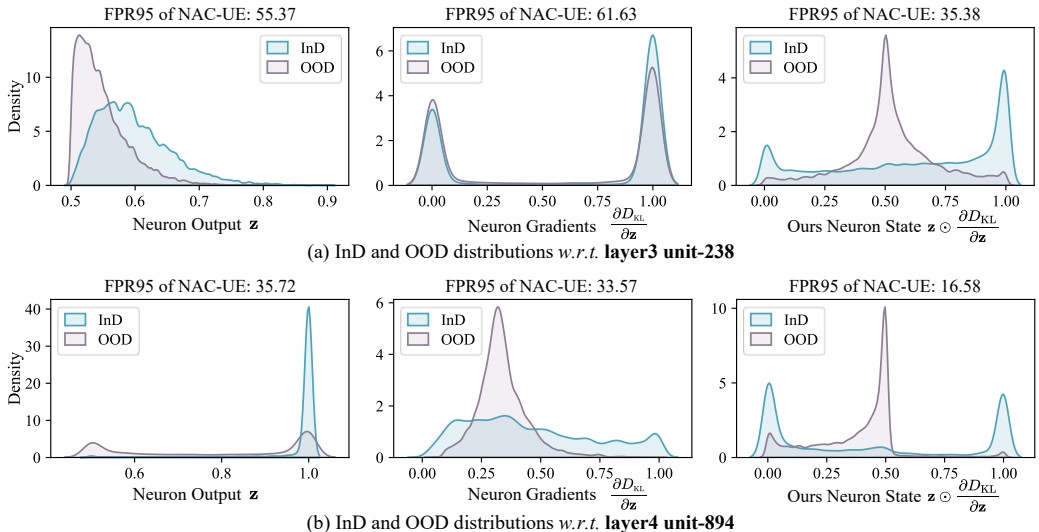

Figure 5: Ablation studies on the neuron activation state. We visualize InD (ImageNet) and OOD (iNaturalist) distributions *w.r.t.* (a) neuron output, $\mathbf{z}$; (b) KL gradients of neuron output, $\partial D_{\text{KL}}/\partial\mathbf{z}$; (c) our defined neuron state, $\mathbf{z} \odot \partial D_{\text{KL}}/\partial\mathbf{z}$. All states are normalized via the sigmoid function.

| Sigmoid Steepness ($\alpha$) | FPR95↓ | AUROC↑ |
|---|---|---|
| $\alpha = 1$ | 40.07 | 85.48 |
| $\alpha = 10$ | 25.64 | 92.11 |
| $\alpha = 100$ | **23.50** | **93.21** |
| $\alpha = 500$ | 48.99 | 86.00 |
| $\alpha = 1000$ | 92.69 | 54.69 |

| Lower Bound ($r$) | FPR95↓ | AUROC↑ |
|---|---|---|
| $r = 0.1$ | 27.10 | 91.51 |
| $r = 0.5$ | 24.16 | 92.79 |
| $r = 1$ | **23.50** | **93.21** |
| $r = 5$ | 28.35 | 92.17 |
| $r = 50$ | 36.70 | 90.38 |

| No. of Intervals ($M$) | FPR95↓ | AUROC↑ |
|---|---|---|
| $M = 10$ | 25.19 | 91.80 |
| $M = 50$ | **23.50** | **93.21** |
| $M = 100$ | 24.23 | 93.09 |
| $M = 500$ | 33.87 | 91.11 |
| $M = 1000$ | 40.36 | 89.69 |

Table 5: `NAC-UE` *w.r.t* different $\alpha$ over CIFAR-10.

Table 6: `NAC-UE` *w.r.t* different $r$ over CIFAR-10.

Table 7: `NAC-UE` *w.r.t* different $M$ over CIFAR-10.

NAC, thus diminishing the effect of coverage scores. Also, a large $r$ makes the NAC vulnerable to data biases, *e.g.*, in datasets with numerous similar samples, a neuron state can be inaccurately characterized with a high coverage score, disregarding other meaningful neuron states. 3) `NAC-UE` works better with a moderate $M$. This is intuitive as a lower $M$ may not sufficiently approximate the PDF function, while a higher $M$ can easily lead to overfitting on the utilized training samples.

## 3.2 CASE STUDY 2: OOD GENERALIZATION

**Setup.** Our experimental settings follow the Domainbed benchmark (Gulrajani & Lopez-Paz, 2021). Without employing digital images, we adopt four datasets: VLCS (Fang et al., 2013) (4 domains, 10,729 images) , PACS (Li et al., 2017) (4 domains, 9,991 images), OfficeHome (Venkateswara et al., 2017) (4 domains, 15,588 images), and TerraInc (Beery et al., 2018) (4 domains, 24,788 images). For all datasets, we report the *leave-one-out* test accuracy following (Gulrajani & Lopez-Paz, 2021), whereby results are averaged over cases that use a single domain for test and the others for training. For all employed backbones, we utilize the hyperparameters suggested by (Cha et al., 2021) to fine-tune them. The training strategy is ERM (Vapnik, 1999), unless stated otherwise. We set the total training steps as 5000, and the evaluation frequency as 300 steps for all models. We use the validation set to select hyperparameters of `NAC-ME`. See Appendix E for more details.

**Model evaluation criteria.** Since OOD data is assumed unavailable during model training, existing methods commonly resort to InD validation accuracy to evaluate a model (Ramé et al., 2022; Yao et al., 2022; Shi et al., 2022; Kim et al., 2021). Thus, we mainly compare `NAC-ME` with the prevalent *validation criterion* (Gulrajani & Lopez-Paz, 2021). We also leverage the *oracle criterion* (Gulrajani & Lopez-Paz, 2021) as the upper bound, which directly utilizes OOD test data for model evaluation.

**Metrics.** Here, we utilize two metrics: 1) Spearman Rank Correlation (RC) between OOD test accuracy and the model evaluation scores (*i.e.*, InD validation accuracy or `NAC-ME` scores), which are sampled at regular evaluation intervals (*i.e.*, every 300 steps) during the training process; 2) OOD Test Accuracy (ACC) of the best model selected by the criterion within a single run of training.

| Bakbone | Method | VLCS | | PACS | | OfficeHome | | TerraInc | | Average | |
|---|---|---|---|---|---|---|---|---|---|---|---|
| | | RC | ACC | RC | ACC | RC | ACC | RC | ACC | RC | ACC |
| ResNet-18 | Oracle | - | 77.67 | - | 80.51 | - | 56.18 | - | 44.51 | - | 64.72 |
| | Validation | 34.27 | 75.12 | 68.71 | **79.01** | 83.50 | 55.60 | 39.58 | 37.36 | 56.52 | 61.77 |
| | NAC-ME | **50.29** | **75.83** | **74.16** | 78.85 | **84.91** | **55.76** | **40.42** | **39.45** | **62.45** | **62.47** |
| | Δ | (+16.02) | (+0.71) | (+5.45) | (-0.16) | (+1.41) | (+0.16) | (+0.84) | (+2.09) | (+5.93) | (+0.70) |
| ResNet-50 | Oracle | - | 79.79 | - | 86.10 | - | 65.95 | - | 50.76 | - | 70.65 |
| | Validation | **31.43** | **77.70** | 58.54 | 84.57 | 67.93 | 65.04 | 37.07 | 46.07 | 48.74 | 68.34 |
| | NAC-ME | 28.68 | 76.41 | **62.07** | **85.28** | **69.16** | **65.23** | **40.16** | **47.10** | **50.02** | **68.51** |
| | Δ | (-2.75) | (-1.29) | (+3.53) | (+0.71) | (+1.23) | (+0.19) | (+3.09) | (+1.03) | (+1.28) | (+0.17) |
| Vit-t16 | Oracle | - | 79.11 | - | 71.99 | - | 61.44 | - | 41.29 | - | 63.46 |
| | Validation | 37.95 | 77.43 | 89.34 | 69.83 | 98.71 | 61.22 | 22.71 | 36.28 | 62.18 | 61.19 |
| | NAC-ME | **49.59** | **77.97** | **90.67** | **70.99** | **99.14** | **61.26** | **23.26** | **36.69** | **65.67** | **61.73** |
| | Δ | (+11.64) | (+0.54) | (+1.33) | (+1.16) | (+0.43) | (+0.04) | (+0.55) | (+0.41) | (+3.49) | (+0.54) |
| Vit-b16 | Oracle | - | 80.96 | - | 90.23 | - | 81.23 | - | 52.23 | - | 76.16 |
| | Validation | 18.81 | 78.70 | 41.38 | 87.80 | 58.29 | 80.11 | 0.92 | 45.49 | 29.85 | 73.03 |
| | NAC-ME | **37.42** | **79.20** | **45.04** | **88.83** | **63.17** | **80.52** | **20.22** | **47.86** | **41.46** | **74.10** |
| | Δ | (+18.61) | (+0.50) | (+3.66) | (+1.03) | (+4.88) | (+0.41) | (+19.30) | (+2.37) | (+11.61) | (+1.07) |

Table 8: OOD generalization results on DomainBed. *Oracle* denotes the upper bound, which uses OOD test data to evaluate models. Δ denotes the improvement of `NAC-ME` over the validation criterion. All scores are averaged over 3 random trials. Full results are provided in Appendix K.

**Results.** As illustrated in Table 8, we mainly compare our `NAC-ME` with the typical validation criterion over four backbones: ResNet-18, ResNet-50, Vit-t16, and Vit-b16. We provide the main observations in the following: 1) The positive correlation (*i.e.*, RC > 0) between the `NAC-ME` and OOD test performance consistently holds across architectures and datasets; 2) By comparison with the validation criterion, `NAC-ME` not only selects more robust models (with higher OOD accuracy), but also exhibits stronger correlation with OOD test performance. For instance, on the TerraInc dataset, `NAC-ME` achieves a rank correlation of 20.22% with OOD test accuracy, surpassing validation criterion by 19.30% on Vit-b16. Similarly, on the VLCS dataset, `NAC-ME` also shows a rank correlation of 52.29%, outperforming the validation criterion by 16.02% on ResNet-18. Such results highlight the potential of `NAC-ME` in evaluating model generalization ability.

**NAC-ME can co-work with SoTA learning algorithms.** Recent literature has suggested numerous learning algorithms to enhance the model robustness (Ganin et al., 2016; Shi et al., 2022; Ramé et al., 2022). In this sense, we further investigate the potential of `NAC-ME` by implementing it with two recent SoTA algorithms: CORAL (Sun & Saenko, 2016) and SelfReg (Kim et al., 2021). The results are shown in Table 9. We can see that `NAC-ME` as an evaluation criterion still presents better performance compared with the validation criterion, which spotlights its effectiveness again.

| Algorithm | Method | RC | ACC |
|---|---|---|---|
| SelfReg | Validation | 61.76 | 80.66 |
| | NAC-ME | **66.85** | **80.92** |
| | Δ | (+5.09) | (+0.26) |
| CORAL | Validation | 70.06 | 80.68 |
| | NAC-ME | **76.55** | **81.54** |
| | Δ | (+6.49) | (+0.86) |

Table 9: OOD generalization results on `PACS` (Li et al., 2017), averaged over 3 trials. Backbone: ResNet-18.

**Does the volume of OOD test data hinder the Rank Correlation (RC)?** As illustrated in Table 8, while in most cases `NAC-ME` outperforms the validation criterion on model selection, we can find that the Rank Correlation (RC) still falls short of its maximum value, *e.g.*, on the VLCS dataset using ResNet-18, RC only reaches 50% compared to the maximum of 100%. Given that Domainbed only provides 6 OOD domains at most, we hypothesize that the volume/variance of OOD test data may be the reason: insufficient OOD test data may be unreliable to reflect model generalization ability, thereby hindering the validity of RC. To this end, we conduct additional experiments on the iWildCam dataset (Koh et al., 2021), which includes 323 domains and 203,029 images in total. Figure 6 illustrates the results, where we analyze the relationship between RC and the volume of OOD test data

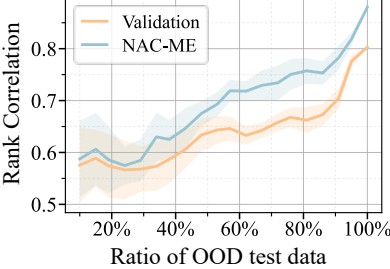

Figure 6: The positive relationship between RC and the volume of OOD test data. Dataset: `iWildCAM` (Koh et al., 2021). Backbone: ResNet-50.

by randomly sampling different ratios of OOD data for RC calculation. As can be seen, an increase in the ratio of test data also leads to an improvement in the RC, which confirms our hypothesis regarding the effect of OOD data. Furthermore, we can observe that in most cases, `NAC-ME` could still outperform the validation criterion. These observations spotlight the capability of our NAC again.

## 4 RELATED WORK

**Neuron coverage in system testing.** Traditional system testing commonly leverages coverage criteria to uncover defects in software programs (Ammann & Offutt, 2008). These criteria measure the degree to which certain codes or components have been exercised, thereby revealing areas with potential defects. To simulate such program testing in neural networks, Pei et al. (2017) first introduced neuron coverage, which measures the proportion of activated neurons within a given input set. The underlying idea is that if a network performs with larger neuron coverage during testing, it is likely to have fewer undetected bugs, *e.g.*, misclassification. In line with this, Ma et al. (2018) extended neuron coverage with fine-grained criteria by considering the neuron outputs from training data. Yuan et al. (2023) introduced layer-wise neuron coverage, focusing on interactions between neurons within the same layer. The most recent work related to our paper is Tian et al. (2023), where they proposed to improve model generalization ability by maximizing neuron coverage during training. However, these existing definitions of neuron coverage still focus on the proportion of activated neurons in the *entire network*, which disregards the activation details of individual neurons. Contrary to that, in this paper, we specifically define neuron activation coverage (NAC) for *individual neurons*, which characterizes the coverage degree of each neuron state under InD data. This provides a more comprehensive perspective on understanding neuron behaviors under InD and OOD scenarios.

**OOD detection.** The goal of OOD detection is to distinguish between InD and OOD data inputs, thereby refraining from using unreliable model predictions during deployment. Existing detection methods can be broadly categorized into three groups: 1) confidence-based (Bendale & Boult, 2016; Hendrycks & Gimpel, 2017; Huang & Li, 2021), 2) distance-based (Huang et al., 2021a; Chen et al., 2020; van Amersfoort et al., 2020), and 3) density-based (Zisselman & Tamar, 2020; Jiang et al., 2022; Kirichenko et al., 2020) approaches. Confidence-based methods commonly resort to the confidence level of model outputs to detect OOD samples, *e.g.*, maximum softmax probability (Hendrycks & Gimpel, 2017). In contrast, distance-based approaches identify OOD samples by measuring the distance (*e.g.*, Mahalanobis distance (Lee et al., 2018)) between input sample and typical InD centroids or prototypes. Likewise, density-based methods employ probabilistic models to explicitly model InD distribution and classify test data located in low-density regions as OOD.

Specific to neuron behaviors, ReAct (Sun et al., 2021) recently proposes the truncation of neuron activations to separate the InD and OOD data. However, such truncation can lead to a decrease in model classification ability (Djurisic et al., 2023). Similarly, LINe (Ahn et al., 2023) seeks to find important neurons using the Shapley value (Shapley, 1997) and then performs activation clipping. Yet, this approach relies on a threshold-based strategy that categorizes neurons into binary states, disregarding valuable neuron distribution details. Unlike them, in this work, we show that by using natural neuron states, a distribution property (*i.e.*, coverage) greatly facilitates the OOD detection.

**OOD generalization.** OOD generalization aims to train models that can overcome distribution shifts between InD and OOD data. While a myriad of studies has emerged to tackle this problem (Li et al., 2018b; Sun & Saenko, 2016; Sagawa et al., 2020; Parascandolo et al., 2021; Arjovsky et al., 2019; Ganin et al., 2016; Li et al., 2018a; Krueger et al., 2021), Gulrajani & Lopez-Paz (2021) recently put forth the importance of model evaluation criterion, and demonstrated that a vanilla ERM (Vapnik, 1999) along with a proper criterion could outperform most state-of-the-art methods. In line with this, Arpit et al. (2022) discovered that using validation accuracy as the evaluation criterion could be unstable for model selection, and thus proposed moving average to stabilize model training. Contrary to that, this work sheds light on the potential of neuron activation coverage for model evaluation, showing that it outperforms the validation criterion in various cases.

## 5 CONCLUSION

In this work, we have presented a neuron activation view to reflect the OOD problem. We have shown that through our formulated neuron activation states, the concept of neuron activation coverage (NAC) could effectively facilitate two OOD tasks: OOD detection and OOD generalization. Specifically, we have demonstrated that 1) InD and OOD inputs can be more separable based on the neuron activation coverage, yielding substantially improved OOD detection performance; 2) a positive correlation between NAC and model generalization ability consistently holds across architectures and datasets, which highlights the potential of NAC-based criterion for model evaluation. Along these lines, we hope this paper has further motivated the community to consider neuron behavior in the OOD problem. This is also the most considerable benefit eventually lies.

## ACKNOWLEDGMENTS

This work was supported in part by Research Grant Council 9229106, in part by ITF MSRP Grant ITS/018/22MS and ITF Project GHP/044/21SZ, in part by National Natural Science Foundation of China under Grant 62022002, in part by Shenzhen Science and Technology Program under Project JCYJ20220530140816037, in part by Hong Kong Research Grants Council General Research Fund 11203220, in part by CityU Strategic Interdisciplinary Research Grant 7020055, in part by Canada CIFAR AI Chairs Program, the Natural Sciences and Engineering Research Council of Canada (NSERC No. RGPIN-2021-02549, No. RGPAS-2021-00034, No. DGECR-2021-00019), in part by JST-Mirai Program Grant No. JPMJMI20B8, and in part by JSPS KAKENHI Grant No. JP21H04877, No. JP23H03372.

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

# Appendix

# Contents

## A  POTENTIAL SOCIAL IMPACT

This study introduces neuron activation coverage (NAC) as an efficient tool for facilitating out-of-distribution (OOD) solutions. By improving OOD detection and generalization, NAC has the potential to significantly enhance the dependability and safety of modern machine learning models. Thus, the social impact of this research can be far-reaching, spanning consumer and business applications in digital content understanding, transportation systems including driver assistance and autonomous vehicles, as well as healthcare applications such as identifying unseen diseases. Moreover, by openly sharing our code, we strive to offer machine learning practitioners a readily available resource for responsible AI development, ultimately benefiting society as a whole. Although we anticipate no negative repercussions, we are committed to expanding upon our framework in future endeavors.

## B  ADDITIONAL THEORETICAL DETAILS

In this section, we present additional theoretical details for Eq. (3) in the main paper. Concretely, we first elaborate on the calculation of gradients *w.r.t.* the sample confidence, *i.e.*, $\partial D_{\mathrm{KL}}/\partial \mathbf{g}(\mathbf{z}) = \mathbf{p} - \mathbf{u}$. Then, we show the detailed derivation of Eq. (3).

**Derivation of sample confidence.** As a reminder, in the main paper, we introduce the Kullback-Leibler (KL) divergence (Kullback & Leibler, 1951) between the network output and a uniform vector $\mathbf{u} = [1/C, 1/C, ..., 1/C] \in \mathbb{R}^C$ as follows:

$$D_{\mathrm{KL}}(\mathbf{u}\|\mathbf{p}) = \sum_{i=1}^{C} u_i \log \frac{u_i}{p_i}$$

$$= -\sum_{i=1}^{C} u_i \log p_i + \sum_{i=1}^{C} u_i \log u_i$$

$$= -\frac{1}{C} \sum_{i=1}^{C} \log p_i - H(\mathbf{u}),$$

where $\mathbf{p} = \mathrm{softmax}(F(\mathbf{x}))$, and $p_i$ denotes $i$-element in $\mathbf{p}$. $H(\mathbf{u}) = -\sum_{i=1}^{C} u_i \log u_i$ is a constant. Let $F(\mathbf{x})_i$ indicates $i$-th element in $F(\mathbf{x})$, we have $p_i = e^{F(\mathbf{x})_i}/\sum_{j=1}^{C} e^{F(\mathbf{x})_j}$. Then, by substituting the expression of $p_i$, we can rewrite KL divergence as:

$$D_{\mathrm{KL}}(\mathbf{u}\|\mathrm{softmax}(F(\mathbf{x}))) = -\frac{1}{C} \sum_{i=1}^{C} \log \frac{e^{F(\mathbf{x})_i}}{\sum_{j=1}^{C} e^{F(\mathbf{x})_j}} - H(\mathbf{u})$$

$$= -\frac{1}{C} \left( \sum_{i=1}^{C} F(\mathbf{x})_i - C \cdot \log \sum_{j=1}^{C} e^{F(\mathbf{x})_j} \right) - H(\mathbf{u}).$$

Subsequently, we can derive the gradients of KL divergence *w.r.t.* the output logit $F(\mathbf{x})_i$ as:

$$\frac{\partial D_{\mathrm{KL}}}{\partial F(\mathbf{x})_i} = -\frac{1}{C} \left( 1 - C \cdot \frac{\partial \log \sum_{j=1}^{C} e^{F(\mathbf{x})_j}}{\partial F(\mathbf{x})_i} \right)$$

$$= -\frac{1}{C} \left( 1 - C \cdot \frac{e^{F(\mathbf{x})_i}}{\sum_{j=1}^{C} e^{F(\mathbf{x})_j}} \right)$$

$$= -\frac{1}{C} + \frac{e^{F(\mathbf{x})_i}}{\sum_{j=1}^{C} e^{F(\mathbf{x})_j}}$$

$$= p_i - u_i.$$

Since $F(\mathbf{x}) = g(f(\mathbf{x})) = g(\mathbf{z})$, we finally have:

$$\frac{\partial D_{\mathrm{KL}}}{\partial g(\mathbf{z})} = \frac{\partial D_{\mathrm{KL}}}{\partial F(\mathbf{x})} = [p_1 - u_1, ..., p_c - u_c]^T = \mathbf{p} - \mathbf{u} \tag{9}$$

**Derivation of Eq.(3).** As shown above, we have $\partial D_{\mathrm{KL}}/\partial g(\mathbf{z}) = \mathbf{p} - \mathbf{u}$. By substituting this expression, we can rewrite the formulation of neuron activation state $\hat{\mathbf{z}}$ as:

$$\hat{\mathbf{z}} = \sigma(\mathbf{z} \odot \frac{\partial D_{\mathrm{KL}}}{\partial \mathbf{z}}) = \sigma(\mathbf{z} \odot (\frac{\partial g(\mathbf{z})}{\partial \mathbf{z}} \cdot \frac{\partial D_{\mathrm{KL}}}{\partial g(\mathbf{z})})) = \sigma(\mathbf{z} \odot (\frac{\partial g(\mathbf{z})}{\partial \mathbf{z}} \cdot (\mathbf{p} - \mathbf{u}))). \tag{10}$$

By expanding the expression of $\partial g(\mathbf{z})/\partial \mathbf{z}$, we have:

$$\frac{\partial g(\mathbf{z})}{\partial \mathbf{z}} = [\frac{\partial g(\mathbf{z})_1}{\partial \mathbf{z}}, \frac{\partial g(\mathbf{z})_2}{\partial \mathbf{z}}, ..., \frac{\partial g(\mathbf{z})_C}{\partial \mathbf{z}}] \in \mathbb{R}^{N \times C}, \tag{11}$$

where $\partial g(\mathbf{z})_i/\partial \mathbf{z} \in \mathbb{R}^N$ denotes the gradients of $i$-th element in the logit output $g(\mathbf{z})$. $N$ is the number of neurons in $\mathbf{z}$, and $C$ is the number of classes. In this way, we can reorganize Eq.(10) as:

$$\hat{\mathbf{z}} = \sigma\Big(\mathbf{z} \odot (\sum_{i=1}^{C} \frac{\partial g(\mathbf{z})_i}{\partial \mathbf{z}} \cdot (p_i - u_i))\Big) = \sigma\Big(\sum_{i=1}^{C} (\mathbf{z} \odot \frac{\partial g(\mathbf{z})_i}{\partial \mathbf{z}}) \cdot (p_i - u_i)\Big). \tag{12}$$

## C  APPROXIMATION DETAILS

In this section, we demonstrate details for the approximation of PDF function, and further show the insights for the choice of $r$ in our NAC function.

### C.1  PRELIMINARIES

**Probability density function (PDF).** The Probability Density Function (PDF), denoted by $\kappa(x)$, measures the probability of a continuous random variable taking on a specific value within a given range. Accordingly, $\kappa(x)$ should possess the following key properties:

(1) Non-Negativity: $\kappa(x) \geq 0$, for all $x \in \mathbb{R}$;

(2) Normalization: $\int_{-\infty}^{\infty} \kappa(x)dx = 1$;

(3) Probability Interpretation: $P(a \leq \mu \leq b) = \int_{a}^{b} \kappa(x)dx$,

where $P(a \leq \mu \leq b)$ denotes the probability that random variable $\mu$ has values within range $[a, b]$.

**Cumulative distribution function (CDF).** In line with PDF, the Cumulative Distribution Function (CDF), denoted by $K(x)$, calculates the cumulative probability for a given $x$-value. Formally, $K(x)$ gives the area under the probability density function up to the specified $x$,

$$K(x) = P(\mu \leq x) = \int_{-\infty}^{x} \kappa(t)dt. \tag{13}$$

By the Fundamental Theorem of Calculus, we can rewrite the function $\kappa(x)$ as,

$$\kappa(x) = K'(x) = \lim_{h \to 0} \frac{K(x+h) - K(x)}{h}. \tag{14}$$

Note that in the main paper, we denote by $\kappa_X^i(\cdot)$ the PDF, and $\Phi_X^i(\cdot)$ the NAC function of $i$-th neuron over the training dataset $X$. In this appendix, we will omit the superscript $i$ and subscript $X$ for simplicity.

### C.2  APPROXIMATION

In line with the main paper, we approximate the PDF of neuron states following a simple histogram-based approach, where the neuron activation space is partitioned into $M$ intervals/bins with logarithmic scales. Formally, suppose the width of a bin is $h$, we can rewrite the PDF function as,

$$\kappa(\hat{z}) \approx \frac{K(\hat{z}+h) - K(\hat{z})}{h} = \frac{P(\hat{z} < \mu \leq \hat{z}+h)}{h} \approx \frac{O(\hat{z})}{|X|} \cdot \frac{1}{h}, \tag{15}$$

where $\hat{z}$ is the neuron activation state, and $O(\hat{z})$ is the number of samples in the bin activating $\hat{z}$.

During the PDF modeling process, we iteratively take a random batch of neuron states as input and assign them corresponding bins.

**The choice of $r$.** With the approximation of PDF, we can rewrite the NAC function as,

$$\Phi(\hat{z}; r) = \frac{1}{r}\min(\kappa(\hat{z}), r) = \min(\frac{\kappa(\hat{z})}{r}, 1) \approx \min(\frac{O(\hat{z})}{|X|h} \cdot \frac{1}{r}, 1), \qquad (16)$$

where $r$ denotes the lower bound for achieving full coverage *w.r.t.* state $\hat{z}$. However, for the above formulation, it could be challenging to search for a suitable $r$, since various factors (*e.g.*, InD dataset size $|X|$) could affect the significance of NAC scores $\Phi(\hat{z}; r)$. In this sense, to further simplify this formulation in the practical deployment, we set $r = \frac{O^*}{|X|h}$, such that

$$\Phi(\hat{z}; r) \approx \min(\frac{O(\hat{z})}{|X|h} \cdot \frac{1}{r}, 1) = \min(\frac{O(\hat{z})}{O^*}, 1), \qquad (17)$$

where $O^*$ represents the minimum number of samples required for bin filling, and $O(\hat{z})$ is the number of samples activating the neuron state $\hat{z}$ in the bin. In this way, we can directly manipulate $O^*$ to control the NAC function in the practical deployment.

## D EXPERIMENTAL DETAILS FOR OOD DETECTION

We conduct experiments following the latest version of OpenOOD[3] (Yang et al., 2022; Zhang et al., 2023a). In this section, we first provide more details for the utilized baselines (Section D.1), datasets and evaluation protocol (Section D.2), and model architectures (Section D.3). Then, we demonstrate the hyperparameters of NAC-UE, and the corresponding search space (Section D.4).

### D.1 BASELINE METHODS

Since NAC-UE performs in a post-hoc fashion, we mainly compare our approach on three bechmarks with the 21 post-hoc OOD detection methods, including OpenMax (Bendale & Boult, 2016), MSP (Hendrycks & Gimpel, 2017), TempScale (Guo et al., 2017), ODIN (Liang et al., 2018), MDS (Lee et al., 2018), MDSEns (Lee et al., 2018), RMDS (Ren et al., 2021), Gram (Sastry & Oore, 2020), EBO (Liu et al., 2020), OpenGAN (Kong & Ramanan, 2021), GradNorm (Huang et al., 2021b), ReAct (Sun et al., 2021), MLS (Hendrycks et al., 2022), KLM (Hendrycks et al., 2022), VIM (Wang et al., 2022), KNN (Sun et al., 2022), DICE (Sun & Li, 2022), RankFeat (Song et al., 2022), ASH (Djurisic et al., 2023), SHE (Zhang et al., 2023b), GEN (Liu et al., 2023). In particular, ReAct and ASH are neuron-based methods, which modify the neuron activations for OOD detection. The results presented in Table 20-22 are from the OpenOOD implementations.

### D.2 OOD BENCHMARKS

We mainly utilize the Far-OOD track of OpenOOD for the evaluation, as it is well defined and supported by many existing studies, *e.g.*, Wang et al. (2022) and Bitterwolf et al. (2023).

**CIFAR benchmarks** CIFAR-10 and CIFAR-100 are widely employed as in-distribution (InD) datasets in existing studies. CIFAR-10 consists of 10 classes, while CIFAR-100 contains 100 classes. In line with OpenOOD, we adopt the same split setup for CIFAR-10 and CIFAR-100 benchmarks. Specifically, for both CIFAR-10 and CIFAR-100, we utilize the official train set with 50,000 training images, and hold out 1,000 samples from the test set as InD validation set. The remaining 9,000 test images are employed as *InD test* set. The 1,000 images covering 20 categories are held out from Tiny ImageNet (Le & Yang, 2015), serving as the *OOD validation* set. To assess the performance of OOD detection methods, we employ four commonly adopted datasets for *OOD test*, which are disjoint with the *OOD validation* set. The details of them are provided below:

1. MNIST (Deng, 2012): This is a 10-class handwriting digital dataset, contains 60,000 images for training and 10,000 for test. We utilize the entire test set for OOD detection.

---

[3]https://github.com/Jingkang50/OpenOOD.

| Architecture | Parameter | Denotation | Values |
|---|---|---|---|
| | - | layer choice | layer4 / layer3 / layer2 / layer1 |
| | $M$ | number of bins for PDF estimation | 50 / 500 / 50 / 500 |
| ResNet-18 | $\alpha$ | sigmoid steepness | 100 / 1000 / 0.001 / 0.001 |
| | $O^*$ | number of samples required for bin filling | 50 / 100 / 5 / 100 |

Table 10: Hyperparameters and their default values on the CIFAR-10 benchmark. Note that $r$ can be computed based on $O^*$, as illustrated in Appendix C.2

| Architecture | Parameter | Denotation | Values |
|---|---|---|---|
| | - | layer choice | layer4 / layer3 / layer2 / layer1 |
| | $M$ | number of bins for PDF estimation | 50 / 1000 / 50 / 50 |
| ResNet-18 | $\alpha$ | sigmoid steepness | 50 / 10 / 1 / 0.005 |
| | $O^*$ | number of samples required for bin filling | 50 / 500 / 500 / 5 |

Table 11: Hyperparameters and their default values on the CIFAR-100 benchmark. Note that $r$ can be computed based on $O^*$, as illustrated in Appendix C.2

2. SVHN (Netzer et al., 2011): This dataset consists of color images depicting house numbers, encompassing ten classes representing digits 0 to 9. We utilize the entire test set, containing 26,032 images.

3. Textures (Cimpoi et al., 2014): The Textures dataset comprises 5,640 real-world texture images classified into 47 categories. We employ the entire dataset for evaluation purposes.

4. Places365 (Zhou et al., 2018): Places365 contains a vast collection of photographs depicting scenes, classified into 365 scene categories. The test set consists of 900 images per category. For OOD detection, we utilize the entire test dataset with 1,305 images removed due to the semantic overlap following (Yang et al., 2022).

**Large-scale ImageNet benchmark**  We employ ImageNet-1k (Deng et al., 2009) as the in-distribution dataset, which contains about 1.2M training images. Following OpenOOD, we utilize 45,000 images from the ImageNet validation set as *InD test* set, and the remaining 5,000 samples as *InD validation* set. To search hyperparameters, 1,763 images from OpenImage-O (Wang et al., 2022) are picked out for *OOD validation*. Finally, we leverage three commonly adopted datasets as *OOD test* for evaluations:

1. iNaturalist (Horn et al., 2018): This dataset consists of 859,000 images of plants and animals, covering over 5,000 different species. Each image is resized to a maximum dimension of 800 pixels. Following (Huang & Li, 2021; Yang et al., 2022), we evaluate our method on a randomly selected subset of 10,000 images, which are drawn from 110 classes that do not overlap with ImageNet-1k.

2. Textures (Cimpoi et al., 2014): This dataset contains 5,640 real-world texture images categorized into 47 classes. We utilize the entire dataset for evaluation purposes.

3. OpenImage-O (Wang et al., 2022): This dataset is curated based on the test set of OpenImage-v3, thereby enjoying natural class statistics to avoid initial design biases. It contains 17,632 images with large scale. Following OpenOOD, we utilize the entire dataset for OOD detection, except the images selected for OOD validation.

### D.3  MODEL ARCHITECTURE

For CIFAR-10 and CIFAR-100 benchmarks, we employ the powerful ResNet-18 (He et al., 2016) architecture. In line with the OpenOOD (Yang et al., 2022; Zhang et al., 2023a), we train ResNet-18 for 100 epochs and evaluate OOD detection methods over three checkpoints. Pleas refer to OpenOOD for more training details.

Following OpenOOD, our experiments for ImageNet benchmark employ two model architectures:

| Architecture | Parameter | Denotation | Values |
|---|---|---|---|
| Vit-b16 | - | layer choice | before_head / block11 / block10 / block9 |
| | $M$ | number of bins for PDF estimation | 50 / 500 / 500 / 1000 |
| | $\alpha$ | sigmoid steepness | 100 / 1 / 10 / 1 |
| | $O^*$ | number of samples required for bin filling | 500 / 50 / 10 / 10 |
| ResNet-50 | - | layer choice | layer4 / layer3 / layer2 / layer1 |
| | $M$ | number of bins for PDF estimation | 50 / 50 / 500 / 1000 |
| | $\alpha$ | sigmoid steepness | 3000 / 300 / 0.01 / 1 |
| | $O^*$ | number of samples required for bin filling | 10 / 500 / 50 / 5000 |

Table 12: Hyperparameters and their default values on the ImageNet benchmark. Note that $r$ can be computed based on $O^*$, as illustrated in Appendix C.2

- ResNet-50 (He et al., 2016) is pretrained on ImageNet-1k. For this model, all images are resized to $224 \times 224$ at the test phase. We use the official checkpoints from Pytorch.
- Vit-b16 (Dosovitskiy et al., 2021) is also pretrained on ImageNet-1k. Similar to ResNet-50, test images are resized to $224 \times 224$. The checkpoints from Pytorch are employed.

### D.4 HYPERPARAMETERS

In all of our experiments, we utilize the InD and OOD validation sets to search for the best hyperparameters. In general, we search $M$ in [50, 500, 1000], and $O^*$ in [5, 10, 50, 100, 500, 5000] across architectures and benchmarks. Since neurons in deeper network layers (*e.g.*, layer4) often varies in a smaller range (See **z** in Figure 5 for an example), we search $\alpha$ in [50, 100, 300, 1000, 3000] for steeper sigmoid function. Otherwise, we search $\alpha$ in [0,001, 0.005, 0.01, 0.1, 1, 10].

In Table 10-12, we list the values of selected hyperparameters for different model architectures over CIFAR-10, CIFAR-100, and ImageNet benchmarks. As suggested in Table 4, we use layer4, layer3, layer2, and layer1 together for OOD detection regrading the ResNet architectures. For Vit-b16, we use the attention layer in block11, block10, block9, and the neurons before the head layer.

## E EXPERIMENTAL DETAILS FOR OOD GENERALIZATION

### E.1 DOMAINBED BENCHMARK

**Datasets** We conduct experiments on the DomainBed (Gulrajani & Lopez-Paz, 2021) benchmark, which is an arguably fairer benchmark in OOD generalization[4]. Without utilizing digital images, we utilize four datasets:

1. VLCS (Fang et al., 2013) is composed of photographic domains, namely Caltech101, LabelMe, SUN09, and VOC2007. This dataset consists of 10,729 examples with dimensions (3, 224, 224) and 5 classes.
2. PACS dataset (Li et al., 2017) consists of four domains: art, cartoons, photos, and sketches. It comprises a total of 9,991 examples with dimensions (3, 224, 224) and 7 classes.
3. OfficeHome (Venkateswara et al., 2017) includes domains: art, clipart, product, real. This dataset contains 15,588 examples of dimension (3, 224, 224) and 65 classes.
4. TerraInc (Beery et al., 2018) is a collection of wildlife photographs captured by camera traps at various locations: L100, L38, L43, and L46. Our version of this dataset contains 24,788 examples of dimension (3, 224, 224) and 10 classes.

**Settings** To ensure the reliability of final results, the data from each domain is partitioned into two parts: 80% for training or testing, and 20% for validation. This process is repeated three times with

---

[4]https://github.com/facebookresearch/DomainBed.

different seeds, such that reported numbers represent the mean and standard errors across these three runs. In our experiments, we report *leave-one-out* test accuracy scores, whereby results are averaged over cases that uses a single domain for test and the others for training. Besides, we set the total training steps as 5000, and the evaluation frequency as 300 steps for all runs.

**Model evaluation criteria**   For model evaluation, we mainly compare our method with the *validation criterion*, which measures model accuracy over 20% source-domain (*i.e.*, InD) validation data. In addition, we also employ the *oracle criterion* as the upper bound, which directly utilizes the accuracy over 20% test-domain data for model evaluation. For more details, we suggest to refer Gulrajani & Lopez-Paz (2021).

### E.2   METRIC: RANK CORRELATION

Rank correlation metrics are widely utilized to measure the relationship between two random variables. The purpose of these metrics is to provide a quantitative way to assess the similarity in rankings of observations across the variables. Following Arpit et al. (2022), we utilize the Spearman Rank Correlation (RC) for assessing the relationship between OOD test accuracy and the model evaluation scores, *i.e.*, InD validation accuracy or InD `NAC-ME` scores.

The rationale behind this choice is that during the training phase, the selection of the optimal model is frequently based on the ranking of model performance, such as validation accuracy. Therefore, utilizing the RC score enables us to directly measure the effectiveness of evaluation criteria in model selection (which naturally translates to early stopping). The value of RC ranges between -1 and 1, where a value of -1 signifies that the rankings of two random variables are exactly opposite to each other; whereas, a value of +1 indicates that the rankings are exactly the same. Furthermore, a RC score of 0 indicates no linear relationship between the two variables.

### E.3   MODEL ARCHITECTURE

In our experiments, we employ four model architectures: ResNet-18 (He et al., 2016), ResNet-50 (He et al., 2016), Vit-t16 (Dosovitskiy et al., 2021), and Vit-b16 (Dosovitskiy et al., 2021). All of them are pretrained on the ImageNet dataset, and are employed as the initial weight. For parameter choices, we suggest to refer Cha et al. (2021).

### E.4   HYPERPARAMETERS

In the case of ResNet architectures, `NAC-ME` computation is performed by using the neurons in *layer-4*. For ResNet-50, layer-4 consists of 2048 neurons, while ResNet-18 has 512 neurons. As for vision transformers, `NAC-ME` computation utilizes the neurons in the attention layer of *block-11*. In the case of Vit-b16, we utilize 768 neurons, while for Vit-t16, we employ 192 neurons. During this series of experiments, we employ the source-domain training data to formulate the NAC function. Besides, to mitigate the noises in training samples, we merely utilize training data that can be correctly classified to build the NAC function.

In order to determine the best hyperparameters of `NAC-ME` for all models, we utilize the InD validation data for parameter search based on the distribution outlined in Table 13. Specifically, given the unavailability of OOD data in this context, we select `NAC-ME` hyperparameters based on the rank correlation with the InD validation accuracy. This is motivated by the fact that the validation accuracy can provide some insights into the model learning progress.

| **Dataset** | No. of bins $M$ | Sigmoid steepness $\alpha$ | No. of samples for bin filling $O^*$ |
|---|---|---|---|
| VLCS / PACS / OfficeHome / TerraInc | [50, 1000] | [1, 500, 5000] / [0.01, 0.1, 0.5] / [0.01, 1, 100] / [0.01, 0.1] | [1, 500, 5000, 10000] if not TerraInc else [5, 10, 30, 50] |

Table 13: Hyperparameters of our `NAC-ME` and their distributions for random search. Note that $r$ can be computed based on $O^*$, as illustrated in Appendix C.2

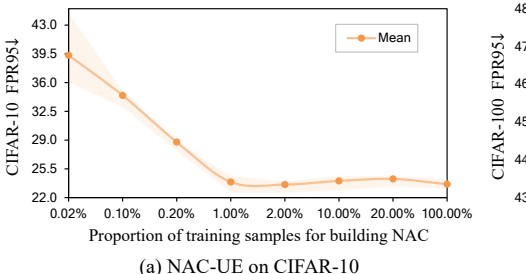 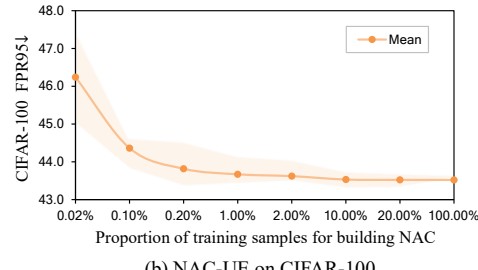

(a) NAC-UE on CIFAR-10  (b) NAC-UE on CIFAR-100

Figure 7: Ablation studies on the number of training samples for building NAC. `NAC-UE` achieves promising performance though only 1% of the training data are utilized, demonstrating the efficiency of our NAC-based approaches.

## F  REPRODUCIBILITY

We will publicly release our code with detailed instructions.

### F.1  SOFTWARE AND HARDWARE

All experiments are performed on a single NVIDIA GeForce RTX 3090 GPU, with Python version 3.8.11. The deep learning framework used is PyTorch 1.10.0, and Torchvision version 0.11.1 is utilized for image processing. We leverage CUDA 11.3 for GPU acceleration.

### F.2  RUNTIME ANALYSIS

The total runtime of the experiments varies depending on the tasks and datasets. In the following, we provide details for two OOD tasks with resent50 architecture, using a single NVIDIA GeForce RTX 3090 GPU. For OOD detection, the experiments (*e.g.*, inference during the test phase) take approximately 10 minutes for all benchmarks. For OOD generalization, the experiments on average take approximately 4 hours for PACS and VLCS, 8 hours for OfficeHome, 8.5 hours for TerraInc.

## G  ADDITIONAL EXPERIMENTAL RESULTS

### G.1  EFFICIENCY ANALYSIS

**Efficient NAC modeling.** As previously mentioned in the main paper, the NAC function is constructed using the InD training data. Specifically, we utilize a subset with 1,000 training images on the CIFAR-10 and CIFAR-100 benchmarks, representing approximately 2% of the total training set. In the case of ImageNet, we employ 1,000 and 50,000 images for ResNet-50 and Vit-b16, respectively, which correspond to approximately 0.1% and 5% of the complete training set.

Here, to gain further insights into the efficiency of our approach, we analyze the performance of `NAC-UE` when constructing the NAC function with varying numbers of training samples. Figure 7 illustrates the results on CIFAR-10 and CIFAR-100 benchmarks, where we randomly sample training images at different ratios and repeat this process five times to ensure the validity of the results. Notably, even when utilizing only 1% of the training data, `NAC-UE` demonstrates remarkable performance that is comparable to the scenario where 100% of the training data is used. This demonstrates the efficiency of our approach, especially in situations with limited data availability.

**Computational Cost Analysis.** To provide a comprehensive view of our approach, we further analyze the computational costs of our proposed NAC-UE method. Specifically, we select the top-3 performing methods from Table 2 as baselines, and compare them with NAC-UE in terms of preprocessing and inference time on the ImageNet benchmark. From the results exhibited in Table 14, the following two observations can be drawn:

1) *Preprocessing Time*: From Table 14, we can see that NAC-UE significantly reduces the preprocessing time compared to the most competitive ViM and SHE, *e.g.*, 7.75s (NAC-UE) vs. 1019.34s (ViM). This finding aligns with our previous experiments (Figure 7), where we show that NAC-UE achieves favorable performance despite utilizing only 1% of the training data for NAC modeling.

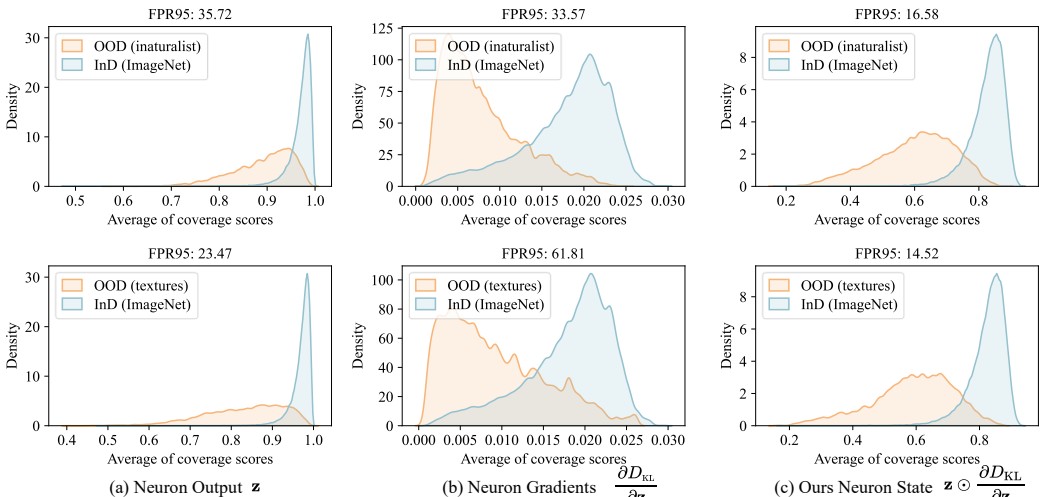

Figure 8: Ablation studies on the neuron activation states $\hat{z}$. We visualize the distribution of averaged coverage scores *w.r.t* all neurons (See Eq.(5)) on the ImageNet benchmark.

| Method | Preprocessing Time (s) | Total Inference Time (s) | AUROC↑ |
|---|---|---|---|
| GEN (Liu et al., 2023) | $0.00 \pm 0.0$ | $43.33 \pm 0.3$ | 89.76 |
| ViM (Wang et al., 2022) | $1087.82 \pm 9.0$ | $48.10 \pm 0.4$ | 92.68 |
| SHE (Zhang et al., 2023b) | $1019.34 \pm 2.2$ | $41.85 \pm 0.5$ | 90.92 |
| NAC-UE (layer4) | $5.43 \pm 0.3$ | $\mathbf{39.63 \pm 0.2}$ | 94.57 |
| NAC-UE (layer4+layer3) | $6.75 \pm 0.3$ | $46.09 \pm 0.7$ | 95.05 |
| NAC-UE (layer4+layer3+layer2) | $7.75 \pm 0.2$ | $69.73 \pm 0.4$ | **95.23** |

Table 14: Computational time comparison between NAC-UE and three SoTA OOD detection methods. Preprocessing and inference time are assessed on the ImageNet benchmark with ResNet-50, which are averaged over five trials. Appendix F.1 provides the details for hardware configurations.

2) *Inference Time*: While NAC-UE requires more inference time with an increase in the number of layers, it is able to outperform SoTA methods in terms of both inference time and detection performance. Remarkably, when utilizing just a single layer (layer4), NAC-UE achieves an AUROC of 94.57% with an inference time of 39.63 seconds. In contrast, GEN achieves only 89.76% AUROC with an inference time of 43.33 seconds. This highlights the efficiency of our approach.

Besides the above analysis, it is also worth noting that there are numerous ongoing research efforts dedicated to facilitating gradient calculation (e.g., Lee et al. (2019)), which could potentially complement our proposed method.

### G.2 Ablation on Neuron Activation State $\hat{z}$

In the main paper (Figure 5), we analyze the formulation of neuron activation state $\hat{z}$ with two neuron examples. In this section, we provide additional experiments to further verify the superiority of $\hat{z}$.

**Distribution of coverage scores under InD and OOD.** To complement the previous analysis which mainly centers on individual neurons, we first investigate the overall neuron activities under different form of neuron states, *i.e.*, raw neuron output $z$, neuron gradients $\partial D_{\mathrm{KL}}/\partial z$, and ours $z \odot \partial D_{\mathrm{KL}}/\partial z$. Figure 8 illustrates the results, where we visualize the InD and OOD distributions of averaged coverage scores *w.r.t* all neurons (See Eq.(5)) on the ImageNet benchmark. We provide the main observations in the following:

Firstly, among all the three variants, $z \odot \partial D_{\mathrm{KL}}/\partial z$ method performs the best, as it inherits the advantages from both $z$ and $\partial D_{\mathrm{KL}}/\partial z$. This spotlights the superiority of our defined neuron state again. Secondly, it can also be found that OOD samples generally present lower coverage scores compared to InD samples. This demonstrates that OOD data tend to provoke abnormal neuron behaviors in comparison to InD data, which confirms the rationale behind our NAC-based approaches.

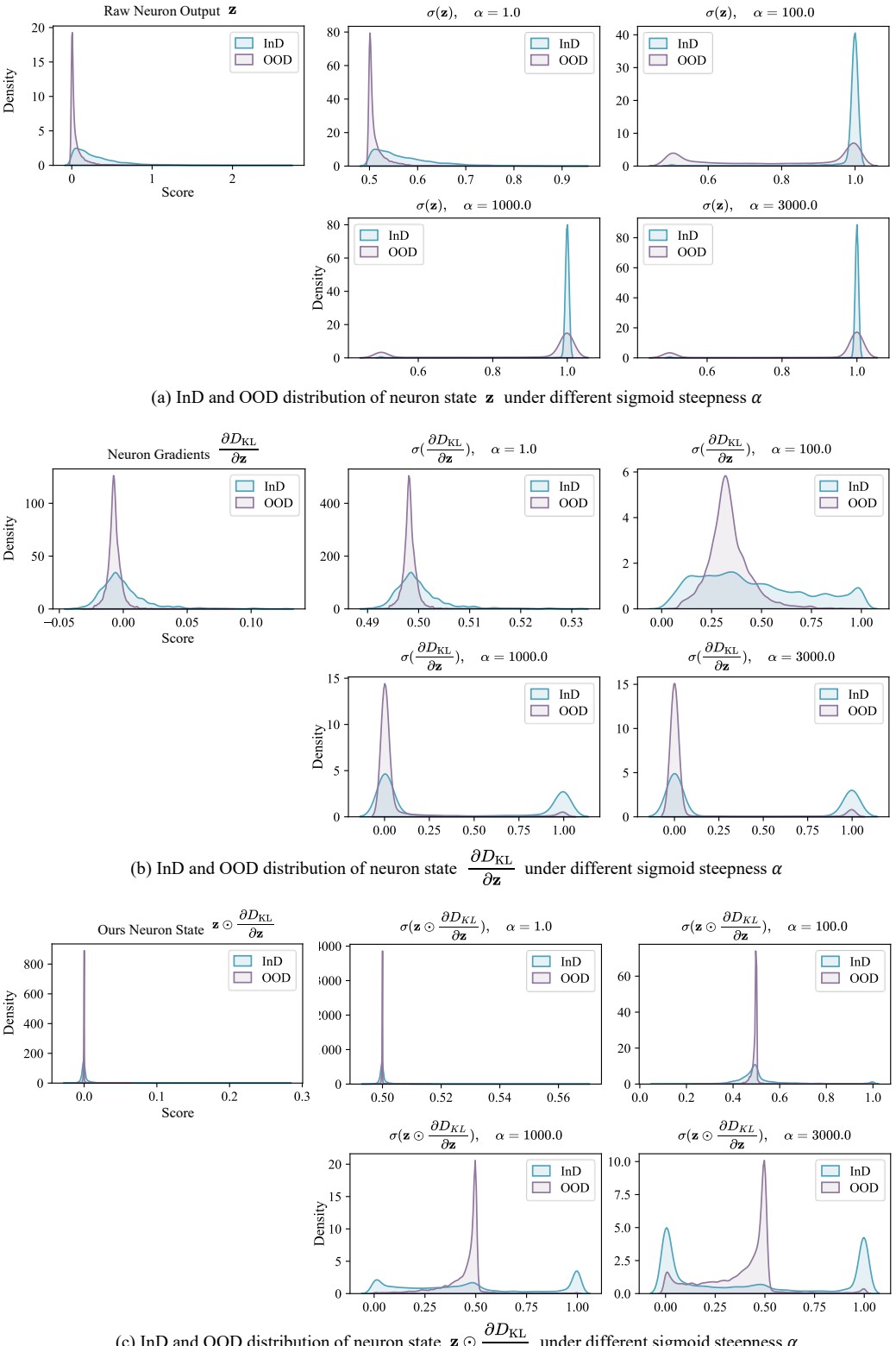

(a) InD and OOD distribution of neuron state $\mathbf{z}$ under different sigmoid steepness $\alpha$

(b) InD and OOD distribution of neuron state $\dfrac{\partial D_{\mathrm{KL}}}{\partial \mathbf{z}}$ under different sigmoid steepness $\alpha$

(c) InD and OOD distribution of neuron state $\mathbf{z} \odot \dfrac{\partial D_{\mathrm{KL}}}{\partial \mathbf{z}}$ under different sigmoid steepness $\alpha$

Figure 9: Ablation studies on the form of neuron activation states with varying sigmoid steepness $\alpha$. We visualize InD and OOD distributions for the **layer4 unit-894** on ResNet-50. NAC-UE achieves best performance when $\alpha = 3000$ (using state $\sigma(\mathbf{z} \odot \frac{\partial D_{\mathrm{KL}}}{\partial \mathbf{z}})$), which outperforms other forms of neuron states, *i.e.*, $\sigma(z)$ and $\sigma(\frac{\partial D_{\mathrm{KL}}}{\partial \mathbf{z}})$.

| Method | CIFAR-100 | | Tiny ImageNet | | Average | |
|---|---|---|---|---|---|---|
| | FPR95↓ | AUROC↑ | FPR95↓ | AUROC↑ | FPR95↓ | AUROC↑ |
| OpenMax | $48.06_{\pm3.25}$ | $86.91_{\pm0.31}$ | $39.18_{\pm1.44}$ | $88.32_{\pm0.28}$ | $43.62_{\pm2.27}$ | $87.62_{\pm0.29}$ |
| MSP | $53.08_{\pm4.86}$ | $87.19_{\pm0.33}$ | $43.27_{\pm3.00}$ | $88.87_{\pm0.19}$ | $48.17_{\pm3.92}$ | $88.03_{\pm0.25}$ |
| TempScale | $55.81_{\pm5.07}$ | $87.17_{\pm0.40}$ | $46.11_{\pm3.63}$ | $89.00_{\pm0.23}$ | $50.96_{\pm4.32}$ | $88.09_{\pm0.31}$ |
| ODIN | $77.00_{\pm5.74}$ | $82.18_{\pm1.87}$ | $75.38_{\pm6.42}$ | $83.55_{\pm1.84}$ | $76.19_{\pm6.08}$ | $82.87_{\pm1.85}$ |
| MDS | $52.81_{\pm3.62}$ | $83.59_{\pm2.27}$ | $46.99_{\pm4.36}$ | $84.81_{\pm2.53}$ | $49.90_{\pm3.98}$ | $84.20_{\pm2.40}$ |
| MDSEns | $91.87_{\pm0.10}$ | $61.29_{\pm0.23}$ | $92.66_{\pm0.42}$ | $59.57_{\pm0.53}$ | $92.26_{\pm0.20}$ | $60.43_{\pm0.26}$ |
| RMDS | $43.86_{\pm3.49}$ | $88.83_{\pm0.35}$ | $33.91_{\pm1.39}$ | $90.76_{\pm0.27}$ | $38.89_{\pm2.39}$ | $89.80_{\pm0.28}$ |
| Gram | $91.68_{\pm2.24}$ | $58.33_{\pm4.49}$ | $90.06_{\pm1.59}$ | $58.98_{\pm5.19}$ | $90.87_{\pm1.91}$ | $58.66_{\pm4.83}$ |
| EBO | $66.60_{\pm4.46}$ | $86.36_{\pm0.58}$ | $56.08_{\pm4.83}$ | $88.80_{\pm0.36}$ | $61.34_{\pm4.63}$ | $87.58_{\pm0.46}$ |
| OpenGAN | $94.84_{\pm3.83}$ | $52.81_{\pm7.69}$ | $94.11_{\pm4.21}$ | $54.62_{\pm7.68}$ | $94.48_{\pm4.01}$ | $53.71_{\pm7.68}$ |
| GradNorm | $94.54_{\pm1.11}$ | $54.43_{\pm1.59}$ | $94.89_{\pm0.60}$ | $55.37_{\pm0.41}$ | $94.72_{\pm0.82}$ | $54.90_{\pm0.98}$ |
| ReAct | $67.40_{\pm7.34}$ | $85.93_{\pm0.83}$ | $59.71_{\pm7.31}$ | $88.29_{\pm0.44}$ | $63.56_{\pm7.33}$ | $87.11_{\pm0.61}$ |
| MLS | $66.59_{\pm4.44}$ | $86.31_{\pm0.59}$ | $56.06_{\pm4.82}$ | $88.72_{\pm0.36}$ | $61.32_{\pm4.62}$ | $87.52_{\pm0.47}$ |
| KLM | $90.55_{\pm5.83}$ | $77.89_{\pm0.75}$ | $85.18_{\pm7.60}$ | $80.49_{\pm0.85}$ | $87.86_{\pm6.37}$ | $79.19_{\pm0.80}$ |
| VIM | $49.19_{\pm3.15}$ | $87.75_{\pm0.28}$ | $40.49_{\pm1.55}$ | $89.62_{\pm0.33}$ | $44.84_{\pm2.31}$ | $88.68_{\pm0.28}$ |
| KNN | $37.64_{\pm0.31}$ | $89.73_{\pm0.14}$ | $30.37_{\pm0.65}$ | $91.56_{\pm0.26}$ | $34.01_{\pm0.38}$ | $90.64_{\pm0.20}$ |
| DICE | $73.71_{\pm7.67}$ | $77.01_{\pm0.88}$ | $66.37_{\pm7.68}$ | $79.67_{\pm0.87}$ | $70.04_{\pm7.64}$ | $78.34_{\pm0.79}$ |
| RankFeat | $65.32_{\pm3.48}$ | $77.98_{\pm2.24}$ | $56.44_{\pm5.76}$ | $80.94_{\pm2.80}$ | $60.88_{\pm4.60}$ | $79.46_{\pm2.52}$ |
| ASH | $87.31_{\pm2.06}$ | $74.11_{\pm1.55}$ | $86.25_{\pm1.58}$ | $76.44_{\pm0.61}$ | $86.78_{\pm1.82}$ | $75.27_{\pm1.04}$ |
| SHE | $81.00_{\pm3.42}$ | $80.31_{\pm0.69}$ | $78.30_{\pm3.52}$ | $82.76_{\pm0.43}$ | $79.65_{\pm3.47}$ | $81.54_{\pm0.51}$ |
| GEN | $58.75_{\pm3.97}$ | $87.21_{\pm0.36}$ | $48.59_{\pm2.34}$ | $89.20_{\pm0.25}$ | $53.67_{\pm3.14}$ | $88.20_{\pm0.30}$ |
| NAC-UE | $\mathbf{35.06}_{\pm0.30}$ | $\mathbf{89.78}_{\pm0.31}$ | $\mathbf{26.53}_{\pm0.21}$ | $\mathbf{91.98}_{\pm0.24}$ | $\mathbf{30.80}_{\pm0.13}$ | $\mathbf{90.88}_{\pm0.25}$ |

Table 15: Near-OOD detection results on the CIFAR-100 and Tiny ImageNet datasets. Following OpenOOD, we employ ResNet-18 model, which is trained solely on the InD dataset, *i.e.*, CIFAR-10. ↑ denotes the higher value is better, while ↓ indicates lower values are better.

**Distribution of neuron states with varying $\alpha$ under InD and OOD.** As illustrated in Table 5, choosing a suitable sigmoid steepness $\alpha$ is crucial for the OOD detection of NAC-UE. To further investigate if this factor also affects other forms of neuron states (*e.g.*, $\mathbf{z}$), we visualize the distribution of different neuron states with varying $\alpha$ under InD and OOD.

We present the results in Figure 9. It can be observed that when the sigmoid steepness $\alpha$ is increased, the neurons behaviors of InD and OOD become more distinguishable in the form of $\mathbf{z} \odot \partial D_{\mathrm{KL}}/\partial\mathbf{z}$. This leads to the superior performance of NAC-UE in OOD detection. On the other hand, when using the vanilla form of $\mathbf{z}$ and $\partial D_{\mathrm{KL}}/\partial\mathbf{z}$, the varying number of $\alpha$ has less of an effect. This result is consistent with our previous finding in Figure 5, which further demonstrates the unique characteristic of our neuron activation state $\hat{\mathbf{z}}$ in distinguishing InD and OOD data points.

**Respective power of $\mathbf{z}$, $\nabla g(\mathbf{z})$, and $\mathbf{p}-\mathbf{u}$..** To assess the individual contributions of different components in our neuron states, we conduct ablation studies to evaluate the respective power of each component: 1) neuron output $\mathbf{z}$, 2) neuron gradients $\nabla g(\mathbf{z})$, and 3) model prediction deviation $\mathbf{p}-\mathbf{u}$. We provide the results in Table 16.

These results reveal two key findings. Firstly, the formulation that includes all three components performs the best among all variants, demonstrating the superiority of our state $\hat{\mathbf{z}}$. Moreover, arbitrary combinations of $\mathbf{z}$, $\nabla g(\mathbf{z})$, and $\mathbf{p}-\mathbf{u}$ can lead to improvements compared to using a single component alone. For instance, utilizing $\mathbf{z} \odot \nabla g(\mathbf{z})$ yields better performance than using either $\mathbf{z}$

| $\mathbf{z}$ | $\nabla g(\mathbf{z})$ | $\mathbf{p}-\mathbf{u}$ | FPR95↓ | AUROC↑ |
|---|---|---|---|---|
| ✓ | | | 45.70 | 89.42 |
| | ✓ | | 84.20 | 64.13 |
| | | ✓ | 59.39 | 80.96 |
| ✓ | ✓ | | 43.43 | 88.9 |
| | ✓ | ✓ | 49.29 | 87.85 |
| ✓ | | ✓ | 44.71 | 89.47 |
| ✓ | ✓ | ✓ | **26.89** | **94.57** |

Table 16: Ablation studies on our defined neuron state. The results are obtained from the ImageNet benchmark for OOD detection.

or $\nabla g(\mathbf{z})$ in isolation. This suggest that all three components encode meaningful information in OOD scenarios, further supporting the rationale behind our proposed states.

## G.3 NEAR-OOD ANALYSIS

Near-OOD detection considers more challenging scenarios, where OOD data points often exhibit characteristics that lie in proximity to InD data distribution (Fort et al., 2021). In this section, we conduct a series of experiments to explore the potential of our approach in handling near-ood scenarios. We employ ResNet-18, trained on CIFAR-10, as the foundation for our experiments. The evaluation

of OOD detection methods is performed on two near-ood datasets: CIFAR-100 (Krizhevsky, 2009) and Tiny ImageNet (Le & Yang, 2015). We carefully follow the evaluation protocol of OpenOOD, and illustrate the results in Table 15. Remarkably, NAC-UE continues to outperform existing 21 SoTA methods on two near-ood datasets. By comparison with the best-performing method KNN, our NAC-UE achieves a 30.80% FPR95 with 3.3% gain. This finding further confirms the effectiveness and robustness of our proposed approach.

## G.4 Weighted NAC for OOD Detection

As illustrated in Eq. (6), we calculate the NAC-UE by considering multiple layers and averaging the coverage scores across layers to obtain the final uncertainty of test data. However, since different layers may contribute differently to the model predictions, it is worth exploring a weighted version of NAC that takes layer difference into account. To do so, we conduct a series of experiments on the CIFAR-10 benchmark, examining our NAC-UE in the weighted version. Specifically, we randomly search the weight for each layer within the same space: [0.2, 0.4, 0.6, 0.8, 1.0], and combine these weighted neural layers for uncertainty estimation. Note that in line with our previous experiments, we first utilize the validation set to search hyperparameters and then test our NAC-UE.

Table 17 illustrates the results. As can be seen, NAC-UE can be further improved in this weighted version, *e.g.*, 2.47% gain on the average FPR95. The again demonstrates the potential of our NAC-based approaches. Interestingly, we also notice that assigning larger weights to the deeper layers often results in better performance for NAC-UE. For instance, the best weight suite found during the random search was [0.4, 0.8, 0.2, 0.4] for [layer4, layer3, layer2, layer1]. We conjecture this is due to that deeper layers often encode richer semantic information than shallow layers, making them crucial in detection problems.

| Method | MINIST | | SVHN | | Textures | | Places365 | | Average | |
|---|---|---|---|---|---|---|---|---|---|---|
| | FPR95↓ | AUROC↑ | FPR↓ | AUROC↑ | FPR95↓ | AUROC↑ | FPR95↓ | AUROC↑ | FPR95↓ | AUROC↑ |
| NAC-UE | $15.14_{\pm2.6}$ | $94.86_{\pm1.4}$ | $14.33_{\pm1.2}$ | $96.05_{\pm0.5}$ | $17.03_{\pm0.6}$ | $95.64_{\pm0.4}$ | $26.73_{\pm0.8}$ | $91.85_{\pm0.3}$ | $18.31_{\pm0.92}$ | $94.60_{\pm0.5}$ |
| NAC-UE (weighted) | $\mathbf{13.94}_{\pm2.4}$ | $\mathbf{95.55}_{\pm1.1}$ | $\mathbf{9.90}_{\pm1.1}$ | $\mathbf{98.10}_{\pm0.2}$ | $\mathbf{13.36}_{\pm0.7}$ | $\mathbf{97.25}_{\pm0.2}$ | $\mathbf{26.16}_{\pm0.8}$ | $\mathbf{92.31}_{\pm0.3}$ | $\mathbf{15.84}_{\pm0.7}$ | $\mathbf{95.80}_{\pm0.2}$ |

Table 17: OOD detection results on the CIFAR-10 benchmark. *NAC-UE (weighted)* denotes our method performed with weighted layer combinations.

## G.5 Maximum NAC Entropy for OOD Generalization

In addition to evaluating model robustness using NAC-ME, in this section, we also investigate the potential of NAC in the training and regularization. Specifically, we propose to improve model generalization ability with the NAC entropy:

$$H(\mathbf{z}) = -\sum_{i=1}^{N} p_i(z_i) \log p_i(z_i), \tag{18}$$

where $p_i(z_i)$ represents the probability associated with the $i$-th neuron output $z_i$ over its NAC distribution, and $N$ is the total number of neurons. To simplify the computation, we directly utilize the raw neuron output $\mathbf{z}$ for NAC modeling, instead of our rectified neuron states $\hat{\mathbf{z}}$. This is because optimizing $\hat{\mathbf{z}}$ could involve second-order gradient calculation, which may result in the extra computational burden and decelerate the learning. Concretely, we propose two loss functions that incorporate NAC entropy for regularization, 1) Minimum NAC entropy loss: $\mathcal{L}_{ce} + \lambda H(\mathbf{z})$ and 2) Maximum NAC entropy loss: $\mathcal{L}_{ce} - \lambda H(\mathbf{z})$, where $\mathcal{L}_{ce}$ denotes the traditional cross entropy loss and $\lambda$ is the regularization coefficient.

We conduct experiments on the PACS dataset using a ResNet-18 backbone, and Table 18 illustrates the results. Interestingly, we can see that maximizing NAC entropy leads to improved performance. This finding also aligns with the intuitive understanding presented in Dubey et al. (2018). By maximizing NAC entropy, we encourage the activation of neurons in unexplored regions over NAC distribution, thus diversifying the neuron activities and improving the model robustness. Conversely, minimizing entropy may result in collapsed neuron behavior.

| Algorithm | Art | Cartoon | Photo | Sketch | Average |
|---|---|---|---|---|---|
| ERM | $77.32_{\pm 0.7}$ | $71.91_{\pm 0.7}$ | $72.36_{\pm 1.1}$ | $94.44_{\pm 0.2}$ | 79.01 |
| NAC (Minimizing Entropy) | $77.28_{\pm 0.2}$ | $69.17_{\pm 0.2}$ | $93.21_{\pm 0.2}$ | $66.73_{\pm 1.2}$ | 76.60 |
| NAC (Maximizing Entropy) | $\mathbf{78.64}_{\pm 0.5}$ | $\mathbf{72.97}_{\pm 0.3}$ | $\mathbf{72.39}_{\pm 0.3}$ | $\mathbf{95.09}_{\pm 0.1}$ | **79.77** |

Table 18: OOD generalization results on the PACS dataset. We implement NAC as an entropy loss, which improves OOD generalization performance.

| OOD Dataset | RMS Calibration Error↓ | | | MAD Calibration Error↓ | | |
|---|---|---|---|---|---|---|
| | MSP | Temperature | NAC-UE | MSP | Temperature | NAC-UE |
| CIFAR100 | 50.62 | 43.01 | **33.04** | 42.56 | 36.64 | **26.92** |
| Tiny ImageNet | 48.01 | 40.25 | **31.99** | 38.86 | 32.88 | **26.25** |
| MNIST | 71.74 | 60.91 | **51.30** | 67.81 | 57.16 | **49.45** |
| SVHN | 65.82 | 56.41 | **45.32** | 59.57 | 51.05 | **41.60** |
| Texture | 42.65 | 35.19 | **28.74** | 32.37 | 26.90 | **23.72** |
| Places365 | 68.85 | 59.67 | **48.65** | 64.65 | 56.02 | **45.33** |

Table 19: Calibration results on five OOD datasets. To evaluate the calibration performance, we follow the evaluation protocol of Hendrycks et al. (2019a), and utilize two metrics: RMS and MAD.

### G.6 UNCERTAINTY CALIBRATION ANALYSIS

Uncertainty calibration plays a pivotal role in achieving reliable and accurate predictions. In this section, we evaluate our NAC-UE specifically focusing on its uncertainty calibration capabilities. We follow the experimental setup outlined in Hendrycks et al. (2019a), and employ two calibration error metrics: Root Mean Square (RMS) and Mean Absolute Deviation (MAD) calibration error. We mainly compare NAC-UE with two simple baselines, MSP (Hendrycks & Gimpel, 2017) and Temperature (Guo et al., 2017), which are officially implemented by OpenOOD.

For the calibration evaluation, we utilize a pretrained model on the CIFAR-10 dataset as the foundation, and assess the calibration errors on both InD and OOD test data. Since OOD points are commonly misclassified and their labels are often not included in the output space of model, confidence estimation methods should assign these OOD points with low confidence. We illustrate the results in Table 19. As can be seen, NAC-UE significantly outperforms two baseline approaches, which demonstrates its potential in prediction calibration.

## H DISCUSSIONS

**NAC vs. SparseIRM.** For OOD generalization, NAC is differs from SparseIRM (Zhou et al., 2022) in two aspects: 1) SparseIRM concentrates on refining model training. In contrast, our NAC focuses on the robustness evaluation of existing models, which provides a different perspective; 2) Drawing parallels with system testing coverage criteria, NAC tracks neuron behaviors in the whole network. However, feature sparsity, as addressed in SparseIRM, is primarily concerned with feature representation, specifically identifying areas where most features are zero or irrelevant. Hence, these two methods are different in their measurement and targets.

**NAC vs. Neural Mean Discrepancy (NMD).** We outline the differences between NAC and NMD (Dong et al., 2022) in three-fold: Firstly, NMD primarily investigates the raw neuron output, while our paper centers on a new formulation of neuron states, which can be decoupled as the neuron gradients, neuron output, and model prediction deviations. This offers a fresh interpretation of neurons in OOD scenarios. Secondly, our NAC specifically focuses on the distribution of neuron states, while NMD examines the mean of neuron output. This distinctive perspective makes our NAC more comprehensive and superior in understanding neuron behaviors. Thirdly, while NMD could effectively detect OOD samples, it requires an additional classifier during the inference phase. Instead, NAC directly calculates the coverage scores in a parameter-free manner, serving as an efficient measure for both OOD detection and generalization.

**NAC vs. SCONE.** While our NAC and SCONE (Bai et al., 2023) both focus on OOD detection and generalization, they are actually different in their targets, design choices, and experimental settings. Specifically, 1) Target: our NAC aims to provide an off-the-shelf/post-hoc tool that efficiently detects OOD data and evaluates model robustness. In contrast, SCONE targets an effective learning strategy, which trains the network to overcome OOD scenarios. 2) Design: NAC directly leverages neuron distributions to reflect model status under OOD scenarios, while SCONE enforces energy margin during the training phase. 3) Experimental setup: our paper focuses on the prevalent OOD detection and generalization setup, where the InD and OOD data are clearly separated. Instead, SCONE centers on the wild scenarios, where data distribution is a mixed version of InD and OOD, turning the OOD into valuable learning resources.

**What makes NAC effective for both OOD detection and generalization?** Conventionally, OOD detection and generalization are perceived as distinct problems: the former primarily addresses semantic (concept) shift while the latter considers covariate shift. Despite agreeing with this traditional perspective, we also should recognize the overlapping nature of these two problem areas. Indeed, a number of prior research studies have examined the role of covariate shift in the context of OOD detection (Tian et al., 2021; Averly & Chao, 2023; Yang et al., 2023), and the impact of semantic shift on OOD generalization (Zhang et al., 2023c; Rostami & Galstyan, 2023). This overlap constitutes a fundamental rationale for why NAC is adept at addressing both of these OOD challenges. Additionally, NAC exhibits unique advantages such as:

1) *NAC benefits from data-centric modeling*: Our NAC method is rooted in a data-centric approach, leveraging the neuron distributions within InD training data to characterize model status. This data-centric modeling enables NAC to effectively capture the intrinsic patterns and characteristics of the model (i.e., from a neuron level), thus serving as an effective tool for uncertainty estimation (OOD detection) and model robustness evaluation (OOD generalization). This also aligns with the principles of DNN defect detection / network quality assessment, in system testing (Xie et al., 2022; Ma et al., 2018).

2) *Shallow to deep layers account for covariate and semantic shifts*: As per research studies (Yang et al., 2023), shallow layers in models often closely correlate with the image style information (covariate level), while deep layers capture semantic information. Since our NAC often works by leveraging multiple layers spanning from shallow to deep, it naturally accounts for both covariate and semantic shifts. This demonstrates its potential in addressing various OOD problems.

**Why NAC-UE exhibits higher improvements on CIFAR compared to ImageNet?** From Table 1 and 2, we can see that NAC-UE often shows higher improvements on the CIFAR compared to ImageNet benchmarks. We conjecture that this phenomenon can be attributed to an intrinsic model bias, where the model generally performs poorly on the challenging ImageNet dataset. For example, the InD accuracy of the model on CIFAR-10 is 95.06, whereas the accuracy over ImageNet is 76.18. This poor performance on ImageNet indicates the worse learning of models, thus potentially raising unstable behaviors in neurons and impacting the performance of our NAC-UE. This also explains the performance gap of NAC-UE on Places365 between CIFAR-10 and CIFAR-100. Since the model trained on CIFAR-100 achieves only 77.25 accuracy, it leads to higher neuron instability and subsequently affects the performance of NAC-UE.

# I  FULL CIFAR RESULTS

| Method | MINIST | | SVHN | | Textures | | Places365 | | Average | |
|---|---|---|---|---|---|---|---|---|---|---|
| | FPR95↓ | AUROC↑ | FPR95↓ | AUROC↑ | FPR95↓ | AUROC↑ | FPR95↓ | AUROC↑ | FPR95↓ | AUROC↑ |
| *CIFAR-10 Benchmark* | | | | | | | | | | |
| OpenMax | $23.33_{\pm4.67}$ | $90.50_{\pm0.44}$ | $25.40_{\pm1.47}$ | $89.77_{\pm0.45}$ | $31.50_{\pm4.05}$ | $89.58_{\pm0.60}$ | $38.52_{\pm2.27}$ | $88.63_{\pm0.28}$ | $29.69_{\pm1.21}$ | $89.62_{\pm0.19}$ |
| MSP | $23.64_{\pm5.81}$ | $92.63_{\pm1.57}$ | $25.82_{\pm1.64}$ | $91.46_{\pm0.40}$ | $34.96_{\pm4.64}$ | $89.89_{\pm0.71}$ | $42.47_{\pm3.81}$ | $88.92_{\pm0.47}$ | $31.72_{\pm1.84}$ | $90.73_{\pm0.43}$ |
| TempScale | $23.53_{\pm7.05}$ | $93.11_{\pm1.77}$ | $26.97_{\pm2.65}$ | $91.66_{\pm0.52}$ | $38.16_{\pm5.89}$ | $90.01_{\pm0.74}$ | $45.27_{\pm4.50}$ | $89.11_{\pm0.52}$ | $33.48_{\pm2.39}$ | $90.97_{\pm0.52}$ |
| ODIN | $23.83_{\pm12.34}$ | $95.24_{\pm1.96}$ | $68.61_{\pm0.52}$ | $84.58_{\pm0.77}$ | $67.70_{\pm11.06}$ | $86.94_{\pm2.26}$ | $70.36_{\pm6.96}$ | $85.07_{\pm1.24}$ | $57.62_{\pm4.24}$ | $87.96_{\pm0.61}$ |
| MDS | $27.30_{\pm3.55}$ | $90.10_{\pm2.41}$ | $25.96_{\pm2.52}$ | $91.18_{\pm0.47}$ | $27.94_{\pm4.20}$ | $92.69_{\pm1.06}$ | $47.67_{\pm4.54}$ | $84.90_{\pm2.54}$ | $32.22_{\pm3.40}$ | $89.72_{\pm1.36}$ |
| MDSEns | $\mathbf{1.30}_{\pm0.51}$ | $\mathbf{99.17}_{\pm0.41}$ | $74.34_{\pm1.04}$ | $66.56_{\pm0.58}$ | $76.07_{\pm0.17}$ | $77.40_{\pm0.28}$ | $94.16_{\pm0.33}$ | $52.47_{\pm0.15}$ | $61.47_{\pm0.48}$ | $73.90_{\pm0.27}$ |
| RMDS | $21.49_{\pm2.32}$ | $93.22_{\pm0.80}$ | $23.46_{\pm1.48}$ | $91.84_{\pm0.26}$ | $25.25_{\pm0.53}$ | $92.23_{\pm0.23}$ | $31.20_{\pm0.28}$ | $91.51_{\pm0.11}$ | $25.35_{\pm0.73}$ | $92.20_{\pm0.21}$ |
| Gram | $70.30_{\pm8.96}$ | $72.64_{\pm2.34}$ | $33.91_{\pm17.35}$ | $91.52_{\pm4.45}$ | $94.64_{\pm2.71}$ | $62.34_{\pm8.27}$ | $90.49_{\pm1.93}$ | $60.44_{\pm3.41}$ | $72.34_{\pm6.73}$ | $71.73_{\pm3.20}$ |
| EBO | $24.99_{\pm12.93}$ | $94.32_{\pm2.53}$ | $35.12_{\pm6.11}$ | $91.79_{\pm0.98}$ | $51.82_{\pm6.11}$ | $89.47_{\pm0.70}$ | $54.85_{\pm6.52}$ | $89.25_{\pm0.78}$ | $41.69_{\pm5.32}$ | $91.21_{\pm0.92}$ |
| OpenGAN | $79.54_{\pm19.71}$ | $56.14_{\pm24.08}$ | $75.27_{\pm26.93}$ | $52.81_{\pm27.60}$ | $83.95_{\pm14.89}$ | $56.14_{\pm18.26}$ | $95.32_{\pm4.45}$ | $53.34_{\pm5.79}$ | $83.52_{\pm11.63}$ | $54.61_{\pm15.51}$ |
| GradNorm | $85.41_{\pm4.85}$ | $63.72_{\pm7.37}$ | $91.65_{\pm2.42}$ | $53.91_{\pm6.36}$ | $98.09_{\pm0.49}$ | $52.07_{\pm4.09}$ | $92.46_{\pm2.28}$ | $60.50_{\pm5.33}$ | $91.90_{\pm2.23}$ | $57.55_{\pm3.22}$ |
| ReAct | $33.77_{\pm18.00}$ | $92.81_{\pm3.03}$ | $50.23_{\pm15.98}$ | $89.12_{\pm3.19}$ | $51.42_{\pm11.42}$ | $89.38_{\pm1.49}$ | $44.20_{\pm3.35}$ | $90.35_{\pm0.78}$ | $44.90_{\pm8.37}$ | $90.42_{\pm1.41}$ |
| MLS | $25.06_{\pm12.87}$ | $94.15_{\pm2.48}$ | $35.09_{\pm6.09}$ | $91.69_{\pm0.94}$ | $51.73_{\pm6.13}$ | $89.41_{\pm0.71}$ | $54.84_{\pm6.51}$ | $89.14_{\pm0.76}$ | $41.68_{\pm5.27}$ | $91.10_{\pm0.89}$ |
| KLM | $76.22_{\pm12.09}$ | $85.00_{\pm2.04}$ | $59.47_{\pm7.06}$ | $84.99_{\pm1.18}$ | $81.95_{\pm9.95}$ | $82.35_{\pm0.33}$ | $95.58_{\pm2.12}$ | $78.37_{\pm0.33}$ | $78.31_{\pm4.84}$ | $82.68_{\pm0.21}$ |
| VIM | $18.36_{\pm1.42}$ | $94.76_{\pm0.38}$ | $19.29_{\pm0.41}$ | $94.50_{\pm0.48}$ | $21.14_{\pm1.83}$ | $95.15_{\pm0.34}$ | $41.43_{\pm2.17}$ | $89.49_{\pm0.39}$ | $25.05_{\pm0.52}$ | $93.48_{\pm0.24}$ |
| KNN | $20.05_{\pm1.36}$ | $94.26_{\pm0.38}$ | $22.60_{\pm1.26}$ | $92.67_{\pm0.30}$ | $24.06_{\pm0.55}$ | $93.16_{\pm0.24}$ | $30.38_{\pm0.63}$ | $91.77_{\pm0.23}$ | $24.27_{\pm0.40}$ | $92.96_{\pm0.14}$ |
| DICE | $30.83_{\pm10.54}$ | $90.37_{\pm5.97}$ | $36.61_{\pm4.74}$ | $90.02_{\pm1.77}$ | $62.42_{\pm4.79}$ | $81.86_{\pm2.35}$ | $77.19_{\pm12.60}$ | $74.67_{\pm4.98}$ | $51.76_{\pm4.42}$ | $84.23_{\pm1.89}$ |
| RankFeat | $61.86_{\pm12.78}$ | $75.87_{\pm5.22}$ | $64.49_{\pm7.38}$ | $68.15_{\pm7.44}$ | $59.71_{\pm9.79}$ | $73.46_{\pm6.49}$ | $43.70_{\pm7.39}$ | $85.99_{\pm3.04}$ | $57.44_{\pm7.99}$ | $75.87_{\pm5.06}$ |
| ASH | $70.00_{\pm10.56}$ | $83.16_{\pm4.66}$ | $83.64_{\pm6.48}$ | $73.46_{\pm6.41}$ | $84.59_{\pm1.74}$ | $77.45_{\pm2.39}$ | $77.89_{\pm7.28}$ | $79.89_{\pm3.69}$ | $79.03_{\pm4.22}$ | $78.49_{\pm2.58}$ |
| SHE | $42.22_{\pm20.59}$ | $90.43_{\pm4.76}$ | $62.74_{\pm4.01}$ | $86.38_{\pm1.32}$ | $84.60_{\pm5.30}$ | $81.57_{\pm1.21}$ | $76.36_{\pm5.32}$ | $82.89_{\pm1.22}$ | $66.48_{\pm5.98}$ | $85.32_{\pm1.43}$ |
| GEN | $23.00_{\pm7.75}$ | $93.83_{\pm2.14}$ | $28.14_{\pm2.59}$ | $91.97_{\pm0.66}$ | $40.74_{\pm6.61}$ | $90.14_{\pm0.76}$ | $47.03_{\pm3.22}$ | $89.46_{\pm0.65}$ | $34.73_{\pm1.58}$ | $91.35_{\pm0.69}$ |
| **NAC-UE** | $15.14_{\pm2.60}$ | $94.86_{\pm1.36}$ | $\mathbf{14.33}_{\pm1.24}$ | $\mathbf{96.05}_{\pm0.47}$ | $\mathbf{17.03}_{\pm0.59}$ | $\mathbf{95.64}_{\pm0.44}$ | $26.73_{\pm0.80}$ | $91.85_{\pm0.28}$ | $\mathbf{18.31}_{\pm0.92}$ | $\mathbf{94.60}_{\pm0.50}$ |

Table 20: OOD detection results on the CIFAR-10 benchmark. We format **first**, second, and third results. Following OpenOOD, we report the performance averaged over three checkpoints of ResNet-18, which are trained solely on the InD dataset, *i.e.*, CIFAR-10. ↑ denotes the higher value is better, while ↓ indicates lower values are better.

| Method | MINIST | | SVHN | | Textures | | Places365 | | Average | |
|---|---|---|---|---|---|---|---|---|---|---|
| | FPR95↓ | AUROC↑ | FPR95↓ | AUROC↑ | FPR95↓ | AUROC↑ | FPR95↓ | AUROC↑ | FPR95↓ | AUROC↑ |
| *CIFAR-100 Benchmark* | | | | | | | | | | |
| OpenMax | $53.82_{\pm4.74}$ | $76.01_{\pm1.39}$ | $53.20_{\pm1.78}$ | $82.07_{\pm1.53}$ | $56.12_{\pm1.91}$ | $80.56_{\pm0.09}$ | $54.85_{\pm1.42}$ | $79.29_{\pm0.40}$ | $54.50_{\pm0.68}$ | $79.48_{\pm0.41}$ |
| MSP | $57.23_{\pm4.68}$ | $76.08_{\pm1.86}$ | $59.07_{\pm2.53}$ | $78.42_{\pm0.89}$ | $61.88_{\pm1.28}$ | $77.32_{\pm0.71}$ | $56.62_{\pm0.87}$ | $79.22_{\pm0.29}$ | $58.70_{\pm1.06}$ | $77.76_{\pm0.44}$ |
| TempScale | $56.05_{\pm4.61}$ | $77.27_{\pm1.85}$ | $57.71_{\pm2.68}$ | $79.79_{\pm1.05}$ | $61.56_{\pm1.43}$ | $78.11_{\pm0.72}$ | $56.46_{\pm0.94}$ | $79.80_{\pm0.25}$ | $57.94_{\pm1.14}$ | $78.74_{\pm0.51}$ |
| ODIN | $45.94_{\pm3.29}$ | $83.79_{\pm1.31}$ | $67.41_{\pm3.88}$ | $74.54_{\pm0.76}$ | $62.37_{\pm2.96}$ | $79.33_{\pm1.08}$ | $59.71_{\pm2.02}$ | $79.45_{\pm0.26}$ | $58.86_{\pm0.79}$ | $79.28_{\pm0.21}$ |
| MDS | $71.72_{\pm2.94}$ | $67.47_{\pm0.81}$ | $67.21_{\pm6.09}$ | $70.68_{\pm6.40}$ | $70.49_{\pm2.48}$ | $76.26_{\pm0.69}$ | $79.61_{\pm0.34}$ | $63.15_{\pm0.49}$ | $72.26_{\pm1.56}$ | $69.39_{\pm1.39}$ |
| MDSEns | $\mathbf{2.83}_{\pm0.86}$ | $\mathbf{98.21}_{\pm0.78}$ | $82.57_{\pm2.58}$ | $53.76_{\pm1.63}$ | $84.94_{\pm0.83}$ | $69.75_{\pm1.14}$ | $96.61_{\pm0.17}$ | $42.27_{\pm0.73}$ | $66.74_{\pm1.04}$ | $66.00_{\pm0.69}$ |
| RMDS | $52.05_{\pm6.28}$ | $79.74_{\pm2.49}$ | $51.65_{\pm3.68}$ | $84.89_{\pm1.10}$ | $53.99_{\pm1.06}$ | $83.65_{\pm0.51}$ | $53.57_{\pm0.43}$ | $83.40_{\pm0.46}$ | $52.81_{\pm0.63}$ | $82.92_{\pm0.42}$ |
| Gram | $53.53_{\pm7.45}$ | $80.71_{\pm4.15}$ | $20.06_{\pm1.96}$ | $95.55_{\pm0.60}$ | $89.51_{\pm2.54}$ | $70.79_{\pm1.32}$ | $94.67_{\pm0.60}$ | $46.38_{\pm1.21}$ | $64.44_{\pm2.37}$ | $73.36_{\pm1.08}$ |
| EBO | $52.62_{\pm3.83}$ | $79.18_{\pm1.37}$ | $53.62_{\pm3.14}$ | $82.03_{\pm1.74}$ | $62.35_{\pm2.06}$ | $78.35_{\pm0.83}$ | $57.75_{\pm0.86}$ | $79.52_{\pm0.23}$ | $56.59_{\pm1.38}$ | $79.77_{\pm0.61}$ |
| OpenGAN | $63.09_{\pm23.25}$ | $68.14_{\pm18.78}$ | $70.35_{\pm2.06}$ | $68.40_{\pm2.15}$ | $74.77_{\pm1.78}$ | $65.84_{\pm3.43}$ | $73.75_{\pm8.32}$ | $69.13_{\pm7.08}$ | $70.49_{\pm7.38}$ | $67.88_{\pm7.16}$ |
| GradNorm | $86.97_{\pm1.44}$ | $65.35_{\pm1.12}$ | $69.90_{\pm7.94}$ | $76.95_{\pm4.73}$ | $92.51_{\pm0.92}$ | $64.58_{\pm0.13}$ | $85.32_{\pm0.44}$ | $69.14_{\pm1.05}$ | $83.68_{\pm1.92}$ | $69.14_{\pm1.05}$ |
| ReAct | $56.04_{\pm5.66}$ | $78.37_{\pm1.59}$ | $50.41_{\pm2.02}$ | $83.01_{\pm0.97}$ | $55.04_{\pm0.82}$ | $80.15_{\pm0.46}$ | $55.30_{\pm0.41}$ | $80.03_{\pm0.11}$ | $54.20_{\pm1.56}$ | $80.39_{\pm0.49}$ |
| MLS | $52.95_{\pm3.82}$ | $78.91_{\pm1.47}$ | $53.90_{\pm3.04}$ | $81.65_{\pm1.49}$ | $62.39_{\pm2.13}$ | $78.39_{\pm0.84}$ | $57.68_{\pm0.91}$ | $79.75_{\pm0.24}$ | $56.73_{\pm1.33}$ | $79.67_{\pm0.57}$ |
| KLM | $73.09_{\pm6.67}$ | $74.15_{\pm2.59}$ | $50.30_{\pm7.04}$ | $79.34_{\pm0.44}$ | $81.80_{\pm5.80}$ | $75.77_{\pm0.45}$ | $81.40_{\pm1.58}$ | $75.70_{\pm0.24}$ | $71.65_{\pm2.01}$ | $76.24_{\pm0.52}$ |
| VIM | $48.32_{\pm1.07}$ | $81.89_{\pm1.02}$ | $46.22_{\pm5.46}$ | $83.14_{\pm3.71}$ | $46.86_{\pm2.29}$ | $85.91_{\pm0.78}$ | $61.57_{\pm0.77}$ | $75.85_{\pm0.37}$ | $50.74_{\pm1.00}$ | $81.70_{\pm0.62}$ |
| KNN | $48.58_{\pm4.67}$ | $82.36_{\pm1.52}$ | $51.75_{\pm3.12}$ | $84.15_{\pm1.09}$ | $53.56_{\pm2.32}$ | $83.66_{\pm0.83}$ | $60.70_{\pm1.03}$ | $79.43_{\pm0.47}$ | $53.65_{\pm0.28}$ | $82.40_{\pm0.17}$ |
| DICE | $51.79_{\pm3.67}$ | $79.86_{\pm1.89}$ | $49.58_{\pm3.32}$ | $84.22_{\pm2.00}$ | $64.23_{\pm1.65}$ | $77.63_{\pm0.34}$ | $59.39_{\pm1.25}$ | $78.33_{\pm0.66}$ | $56.25_{\pm0.60}$ | $80.01_{\pm0.18}$ |
| RankFeat | $75.01_{\pm5.83}$ | $63.03_{\pm3.86}$ | $58.49_{\pm2.30}$ | $72.14_{\pm1.39}$ | $66.87_{\pm3.80}$ | $69.40_{\pm3.08}$ | $77.42_{\pm1.96}$ | $46.38_{\pm1.83}$ | $69.45_{\pm1.01}$ | $67.10_{\pm1.42}$ |
| ASH | $66.58_{\pm3.88}$ | $77.23_{\pm0.46}$ | $46.00_{\pm2.67}$ | $85.60_{\pm1.40}$ | $61.27_{\pm2.74}$ | $80.72_{\pm0.70}$ | $62.95_{\pm0.99}$ | $78.76_{\pm0.16}$ | $59.20_{\pm2.46}$ | $80.58_{\pm0.66}$ |
| SHE | $58.78_{\pm2.70}$ | $76.76_{\pm1.07}$ | $59.15_{\pm7.61}$ | $80.97_{\pm3.98}$ | $73.29_{\pm3.22}$ | $73.64_{\pm1.28}$ | $65.24_{\pm0.98}$ | $76.30_{\pm0.51}$ | $64.12_{\pm2.70}$ | $76.92_{\pm1.16}$ |
| GEN | $53.92_{\pm5.71}$ | $78.29_{\pm2.05}$ | $55.45_{\pm2.76}$ | $81.41_{\pm1.50}$ | $61.23_{\pm1.40}$ | $78.74_{\pm0.81}$ | $56.25_{\pm1.01}$ | $80.28_{\pm0.27}$ | $56.71_{\pm1.59}$ | $79.68_{\pm0.75}$ |
| **NAC-UE** | $21.97_{\pm6.62}$ | $93.15_{\pm1.63}$ | $24.39_{\pm4.66}$ | $92.40_{\pm1.26}$ | $\mathbf{40.65}_{\pm1.94}$ | $\mathbf{89.32}_{\pm0.55}$ | $73.57_{\pm1.16}$ | $73.05_{\pm0.68}$ | $\mathbf{40.14}_{\pm1.86}$ | $\mathbf{86.98}_{\pm0.37}$ |

Table 21: OOD detection results on the CIFAR-100 benchmark. We format **first**, second, and third results. Following OpenOOD, we report the performance averaged over three checkpoints of ResNet-18, which are trained solely on the InD dataset, *i.e.*, CIFAR-100. ↑ denotes the higher value is better, while ↓ indicates lower values are better.

## J  FULL IMAGENET RESULTS

| Method | iNaturalist | | | OpenImage-O | | | Textures | | |
|---|---|---|---|---|---|---|---|---|---|
| | ResNet-50 | Vit-b16 | Average | ResNet-50 | Vit-b16 | Average | ResNet-50 | Vit-b16 | Average |
| OpenMax | 92.05 | 94.93 | 93.49 | 87.62 | 87.36 | 87.49 | 88.10 | 85.52 | 86.81 |
| MSP | 88.41 | 88.19 | 88.30 | 84.86 | 84.86 | 84.86 | 82.43 | 85.06 | 83.75 |
| TempScale | 90.50 | 88.54 | 89.52 | 87.22 | 85.04 | 86.13 | 84.95 | 85.39 | 85.17 |
| ODIN | 91.17 | / | 91.17 | 88.23 | / | 88.23 | 89.00 | / | 89.00 |
| MDS | 63.67 | 96.01 | 79.84 | 69.27 | **92.38** | 80.83 | 89.80 | 89.41 | 89.61 |
| MDSEns | 61.82 | / | 61.82 | 60.80 | / | 60.80 | 79.94 | / | 79.94 |
| RMDS | 87.24 | **96.10** | 91.67 | 85.84 | 92.32 | 89.08 | 86.08 | 89.38 | 87.73 |
| Gram | 76.67 | / | 76.67 | 74.43 | / | 74.43 | 88.02 | / | 88.02 |
| EBO | 90.63 | 79.30 | 84.97 | 89.06 | 76.48 | 82.77 | 88.70 | 81.17 | 84.94 |
| OpenGAN | / | / | / | / | / | / | / | / | / |
| GradNorm | 93.89 | 42.42 | 68.16 | 84.82 | 37.82 | 61.32 | 92.05 | 44.99 | 68.52 |
| ReAct | 96.34 | 86.11 | 91.23 | 91.87 | 84.29 | 88.08 | 92.79 | 86.66 | 89.73 |
| MLS | 91.17 | 85.29 | 88.23 | 89.17 | 81.60 | 85.39 | 88.39 | 83.74 | 86.07 |
| KLM | 90.78 | 89.59 | 90.19 | 87.30 | 87.03 | 87.17 | 84.72 | 86.49 | 85.61 |
| VIM | 89.56 | 95.72 | 92.64 | 90.50 | 92.18 | 91.34 | **97.97** | 90.61 | 94.29 |
| KNN | 86.41 | 91.46 | 88.94 | 87.04 | 89.86 | 88.45 | 97.09 | 91.12 | 94.11 |
| DICE | 92.54 | 82.50 | 87.52 | 88.26 | 82.22 | 85.24 | 92.04 | 82.21 | 87.13 |
| RankFeat | 40.06 | / | 40.06 | 50.83 | / | 50.83 | 70.90 | / | 70.90 |
| ASH | **97.07** | 50.62 | 73.85 | **93.26** | 55.51 | 74.39 | 96.90 | 48.53 | 72.72 |
| SHE | 92.65 | 93.57 | 93.11 | 86.52 | 91.04 | 88.78 | 93.60 | 92.65 | 93.13 |
| GEN | 92.44 | 93.54 | 92.99 | 89.26 | 90.27 | 89.77 | 87.59 | 90.23 | 88.91 |
| **NAC-UE** | 96.52 | 93.72 | **95.12** | 91.45 | 91.58 | **91.52** | 97.9 | **94.17** | **96.04** |

Table 22: OOD detection results on the ImageNet benchmark. We format **first**, second, and third results. Following OpenOOD, we report the AUROC↑ scores over two backbones (ResNet-50 and Vit-b16), which are trained solely on the InD dataset, *i.e.*, ImageNet-1k.

## K  FULL DOMAINBED RESULTS

| | Method | Caltech101 | | LabelMe | | SUN09 | | VOC2007 | | Average | |
|---|---|---|---|---|---|---|---|---|---|---|---|
| | | RC | ACC | RC | ACC | RC | ACC | RC | ACC | RC | ACC |
| RN18 | Oracle | - | $97.00_{\pm0.6}$ | - | $65.60_{\pm0.3}$ | - | $71.44_{\pm0.8}$ | - | $76.64_{\pm0.5}$ | - | 77.67 |
| | Validation | $36.03_{\pm17.3}$ | $95.38_{\pm0.9}$ | $17.57_{\pm13.2}$ | $63.62_{\pm1.1}$ | $50.33_{\pm13.6}$ | $67.73_{\pm0.6}$ | $33.17_{\pm15.7}$ | $\mathbf{73.75}_{\pm0.7}$ | 34.27 | 75.12 |
| | NAC-ME | $\mathbf{67.73}_{\pm3.0}$ | $\mathbf{96.41}_{\pm0.5}$ | $7.52_{\pm3.4}$ | $\mathbf{63.72}_{\pm0.8}$ | $\mathbf{64.22}_{\pm7.2}$ | $\mathbf{70.89}_{\pm1.1}$ | $\mathbf{61.68}_{\pm10.2}$ | $72.29_{\pm0.5}$ | $\mathbf{50.29}$ | $\mathbf{75.83}$ |
| RN50 | Oracle | - | $98.53_{\pm0.3}$ | - | $68.69_{\pm0.8}$ | - | $73.88_{\pm0.5}$ | - | $78.07_{\pm0.3}$ | - | 79.79 |
| | Validation | $20.75_{\pm17.0}$ | $98.00_{\pm0.2}$ | $\mathbf{35.29}_{\pm13.2}$ | $\mathbf{65.16}_{\pm1.4}$ | $\mathbf{33.01}_{\pm3.1}$ | $70.37_{\pm0.6}$ | $\mathbf{36.68}_{\pm4.3}$ | $\mathbf{77.28}_{\pm0.3}$ | $\mathbf{31.43}$ | $\mathbf{77.70}$ |
| | NAC-ME | $\mathbf{54.90}_{\pm2.6}$ | $\mathbf{98.50}_{\pm0.3}$ | $-2.04_{\pm2.7}$ | $60.27_{\pm0.6}$ | $28.27_{\pm14.0}$ | $\mathbf{70.88}_{\pm2.1}$ | $33.58_{\pm8.9}$ | $76.00_{\pm1.0}$ | 28.68 | 76.41 |
| Vit-t16 | Oracle | - | $98.88_{\pm0.1}$ | - | $66.65_{\pm0.3}$ | - | $74.78_{\pm0.2}$ | - | $76.14_{\pm0.3}$ | - | 79.11 |
| | Validation | $\mathbf{25.57}_{\pm8.8}$ | $\mathbf{98.32}_{\pm0.3}$ | $41.01_{\pm4.6}$ | $63.87_{\pm0.6}$ | $47.14_{\pm2.7}$ | $72.44_{\pm0.1}$ | $38.07_{\pm12.3}$ | $\mathbf{75.08}_{\pm0.6}$ | 37.95 | 77.43 |
| | NAC-ME | $24.02_{\pm0.2}$ | $98.26_{\pm0.1}$ | $\mathbf{69.69}_{\pm3.6}$ | $\mathbf{64.30}_{\pm0.2}$ | $\mathbf{49.51}_{\pm6.2}$ | $\mathbf{74.36}_{\pm0.4}$ | $\mathbf{55.15}_{\pm9.0}$ | $74.95_{\pm0.3}$ | $\mathbf{49.59}$ | $\mathbf{77.97}$ |
| Vit-b16 | Oracle | - | $98.65_{\pm0.1}$ | - | $67.18_{\pm0.5}$ | - | $78.24_{\pm0.4}$ | - | $79.77_{\pm0.5}$ | - | 80.96 |
| | Validation | $-6.45_{\pm10.2}$ | $95.49_{\pm0.7}$ | $\mathbf{43.30}_{\pm14.1}$ | $\mathbf{64.67}_{\pm0.6}$ | $12.83_{\pm12.2}$ | $76.68_{\pm0.9}$ | $25.57_{\pm26.9}$ | $\mathbf{77.96}_{\pm0.9}$ | 18.81 | 78.70 |
| | NAC-ME | $\mathbf{47.79}_{\pm2.2}$ | $\mathbf{97.44}_{\pm0.1}$ | $38.48_{\pm10.4}$ | $64.30_{\pm1.4}$ | $\mathbf{30.07}_{\pm11.6}$ | $\mathbf{77.22}_{\pm0.4}$ | $\mathbf{33.33}_{\pm4.1}$ | $77.85_{\pm0.4}$ | $\mathbf{37.42}$ | $\mathbf{79.20}$ |

Table 23: OOD generalization results on VLCS dataset (Fang et al., 2013). *Oracle* denotes the upper bound, which uses OOD test data to evaluate models. The training strategy is ERM (Vapnik, 1999). All scores are averaged over 3 random trials.

| | Method | Art | | Cartoon | | Photo | | Sketch | | Average | |
|---|---|---|---|---|---|---|---|---|---|---|---|
| | | RC | ACC | RC | ACC | RC | ACC | RC | ACC | RC | ACC |
| RN18 | Oracle | - | $78.52_{\pm0.2}$ | - | $75.09_{\pm0.8}$ | - | $94.96_{\pm0.3}$ | - | $73.47_{\pm1.5}$ | - | 80.51 |
| | Validation | $72.22_{\pm5.1}$ | $77.32_{\pm0.7}$ | $65.20_{\pm6.6}$ | $\mathbf{71.91}_{\pm0.7}$ | $60.87_{\pm7.1}$ | $94.44_{\pm0.2}$ | $76.55_{\pm1.2}$ | $\mathbf{72.36}_{\pm1.1}$ | 68.71 | $\mathbf{79.01}$ |
| | NAC-ME | $\mathbf{75.49}_{\pm5.8}$ | $\mathbf{77.89}_{\pm0.3}$ | $\mathbf{74.84}_{\pm1.3}$ | $71.54_{\pm0.8}$ | $\mathbf{65.36}_{\pm6.0}$ | $\mathbf{94.64}_{\pm0.2}$ | $\mathbf{80.96}_{\pm1.9}$ | $71.34_{\pm2.4}$ | $\mathbf{74.16}$ | 78.85 |
| RN50 | Oracle | - | $86.78_{\pm0.5}$ | - | $81.31_{\pm0.5}$ | - | $98.43_{\pm0.0}$ | - | $77.87_{\pm0.4}$ | - | 86.10 |
| | Validation | $70.26_{\pm9.1}$ | $\mathbf{86.72}_{\pm0.5}$ | $65.93_{\pm10.3}$ | $78.86_{\pm1.3}$ | $\mathbf{38.73}_{\pm12.3}$ | $\mathbf{97.83}_{\pm0.1}$ | $59.23_{\pm11.4}$ | $74.87_{\pm1.1}$ | 58.54 | 84.57 |
| | NAC-ME | $\mathbf{73.61}_{\pm1.4}$ | $86.56_{\pm0.4}$ | $\mathbf{76.14}_{\pm5.0}$ | $\mathbf{80.22}_{\pm1.1}$ | $30.15_{\pm15.3}$ | $97.68_{\pm0.1}$ | $\mathbf{68.38}_{\pm8.8}$ | $\mathbf{76.66}_{\pm1.2}$ | $\mathbf{62.07}$ | $\mathbf{85.28}$ |
| Vit-t16 | Oracle | - | $75.84_{\pm0.1}$ | - | $66.01_{\pm0.7}$ | - | $96.31_{\pm0.2}$ | - | $49.79_{\pm1.6}$ | - | 71.99 |
| | Validation | $\mathbf{88.97}_{\pm3.7}$ | $\mathbf{75.66}_{\pm0.2}$ | $92.32_{\pm1.6}$ | $\mathbf{65.41}_{\pm0.4}$ | $93.79_{\pm1.8}$ | $\mathbf{96.16}_{\pm0.2}$ | $82.27_{\pm3.7}$ | $42.10_{\pm2.2}$ | 89.34 | 69.83 |
| | NAC-ME | $88.15_{\pm3.8}$ | $75.64_{\pm0.2}$ | $\mathbf{92.57}_{\pm0.5}$ | $64.04_{\pm0.6}$ | $\mathbf{95.02}_{\pm2.0}$ | $96.11_{\pm0.2}$ | $\mathbf{86.93}_{\pm2.4}$ | $\mathbf{48.20}_{\pm1.9}$ | $\mathbf{90.67}$ | $\mathbf{70.99}$ |
| Vit-b16 | Oracle | - | $94.81_{\pm0.3}$ | - | $86.57_{\pm0.2}$ | - | $99.65_{\pm0.0}$ | - | $79.89_{\pm0.6}$ | - | 90.23 |
| | Validation | $\mathbf{22.96}_{\pm7.7}$ | $92.58_{\pm0.2}$ | $47.96_{\pm4.3}$ | $84.54_{\pm0.3}$ | $\mathbf{55.64}_{\pm5.2}$ | $\mathbf{99.43}_{\pm0.0}$ | $38.97_{\pm3.1}$ | $74.66_{\pm2.8}$ | 41.38 | 87.80 |
| | NAC-ME | $17.73_{\pm3.9}$ | $\mathbf{93.25}_{\pm0.5}$ | $\mathbf{63.24}_{\pm3.1}$ | $\mathbf{85.09}_{\pm1.1}$ | $37.17_{\pm7.7}$ | $99.33_{\pm0.1}$ | $\mathbf{62.01}_{\pm6.0}$ | $\mathbf{77.66}_{\pm0.4}$ | $\mathbf{45.04}$ | $\mathbf{88.83}$ |

Table 24: OOD generalization results on PACS dataset (Li et al., 2017). *Oracle* denotes the upper bound, which uses OOD test data to evaluate models. The training strategy is ERM (Vapnik, 1999). All scores are averaged over 3 random trials.

| | Method | Art | | Clipart | | Product | | Real | | Average | |
|---|---|---|---|---|---|---|---|---|---|---|---|
| | | RC | ACC | RC | ACC | RC | ACC | RC | ACC | RC | ACC |
| RN18 | Oracle | - | $48.04_{\pm0.2}$ | - | $41.99_{\pm0.2}$ | - | $66.26_{\pm0.2}$ | - | $68.41_{\pm0.2}$ | - | 56.18 |
| | Validation | $\mathbf{86.36}_{\pm1.9}$ | $\mathbf{47.68}_{\pm0.3}$ | $75.33_{\pm3.2}$ | $41.16_{\pm0.6}$ | $88.73_{\pm3.3}$ | $65.82_{\pm0.1}$ | $83.58_{\pm3.1}$ | $67.73_{\pm0.4}$ | 83.50 | 55.60 |
| | NAC-ME | $86.19_{\pm2.5}$ | $\mathbf{47.68}_{\pm0.1}$ | $\mathbf{77.45}_{\pm5.8}$ | $41.16_{\pm0.6}$ | $\mathbf{91.83}_{\pm1.2}$ | $\mathbf{66.15}_{\pm0.2}$ | $\mathbf{84.15}_{\pm4.5}$ | $\mathbf{68.04}_{\pm0.3}$ | $\mathbf{84.91}$ | $\mathbf{55.76}$ |
| RN50 | Oracle | - | $60.20_{\pm0.3}$ | - | $51.76_{\pm0.2}$ | - | $75.49_{\pm0.1}$ | - | $76.37_{\pm0.3}$ | - | 65.95 |
| | Validation | $71.32_{\pm4.2}$ | $59.01_{\pm0.5}$ | $53.43_{\pm6.5}$ | $\mathbf{50.29}_{\pm0.4}$ | $\mathbf{81.21}_{\pm5.7}$ | $\mathbf{74.96}_{\pm0.5}$ | $\mathbf{65.77}_{\pm7.0}$ | $\mathbf{75.88}_{\pm0.2}$ | 67.93 | 65.04 |
| | NAC-ME | $\mathbf{78.68}_{\pm7.0}$ | $\mathbf{60.20}_{\pm0.3}$ | $\mathbf{59.15}_{\pm3.1}$ | $50.19_{\pm0.4}$ | $78.68_{\pm5.3}$ | $74.66_{\pm0.4}$ | $60.13_{\pm7.3}$ | $75.86_{\pm0.1}$ | $\mathbf{69.16}$ | $\mathbf{65.23}$ |
| Vit-t16 | Oracle | - | $56.97_{\pm0.1}$ | - | $43.58_{\pm0.4}$ | - | $71.82_{\pm0.1}$ | - | $73.41_{\pm0.1}$ | - | 61.44 |
| | Validation | $\mathbf{98.77}_{\pm0.3}$ | $56.39_{\pm0.4}$ | $98.45_{\pm0.1}$ | $43.47_{\pm0.5}$ | $98.28_{\pm0.6}$ | $71.62_{\pm0.2}$ | $99.35_{\pm0.3}$ | $\mathbf{73.41}_{\pm0.1}$ | 98.71 | 61.22 |
| | NAC-ME | $\mathbf{98.77}_{\pm0.5}$ | $56.39_{\pm0.4}$ | $\mathbf{98.86}_{\pm0.4}$ | $\mathbf{43.55}_{\pm0.4}$ | $\mathbf{99.35}_{\pm0.3}$ | $\mathbf{71.73}_{\pm0.1}$ | $\mathbf{99.59}_{\pm0.1}$ | $73.39_{\pm0.1}$ | $\mathbf{99.14}$ | $\mathbf{61.26}$ |
| Vit-b16 | Oracle | - | $78.94_{\pm0.2}$ | - | $68.12_{\pm0.3}$ | - | $87.93_{\pm0.1}$ | - | $89.91_{\pm0.0}$ | - | 81.23 |
| | Validation | $54.66_{\pm4.7}$ | $77.77_{\pm0.3}$ | $56.70_{\pm2.4}$ | $66.49_{\pm0.3}$ | $\mathbf{61.03}_{\pm5.9}$ | $87.19_{\pm0.0}$ | $\mathbf{60.78}_{\pm9.2}$ | $88.99_{\pm0.1}$ | 58.29 | 80.11 |
| | NAC-ME | $\mathbf{70.83}_{\pm1.0}$ | $\mathbf{78.03}_{\pm0.3}$ | $\mathbf{65.03}_{\pm2.3}$ | $\mathbf{67.52}_{\pm0.7}$ | $56.13_{\pm3.6}$ | $\mathbf{87.43}_{\pm0.3}$ | $60.70_{\pm3.2}$ | $\mathbf{89.12}_{\pm0.2}$ | $\mathbf{63.17}$ | $\mathbf{80.52}$ |

Table 25: OOD generalization results on OfficeHome dataset (Venkateswara et al., 2017). *Oracle* denotes the upper bound, which uses OOD test data to evaluate models. The training strategy is ERM (Vapnik, 1999). All scores are averaged over 3 random trials.

| | Method | Loc100 | | Loc38 | | Loc43 | | Loc46 | | Average | |
|---|---|---|---|---|---|---|---|---|---|---|---|
| | | RC | ACC | RC | ACC | RC | ACC | RC | ACC | RC | ACC |
| RN18 | Oracle | - | $54.94_{\pm1.3}$ | - | $35.64_{\pm0.7}$ | - | $52.32_{\pm0.1}$ | - | $35.14_{\pm0.6}$ | - | 44.51 |
| | Validation | $\mathbf{12.01}_{\pm11.9}$ | $40.60_{\pm2.5}$ | $49.75_{\pm10.9}$ | $28.41_{\pm2.9}$ | $\mathbf{58.17}_{\pm12.8}$ | $48.31_{\pm1.5}$ | $38.40_{\pm10.3}$ | $32.12_{\pm0.8}$ | 39.58 | 37.36 |
| | NAC-ME | $10.29_{\pm13.2}$ | $\mathbf{41.31}_{\pm2.5}$ | $\mathbf{53.19}_{\pm9.4}$ | $\mathbf{33.23}_{\pm0.7}$ | $54.49_{\pm8.2}$ | $\mathbf{50.26}_{\pm0.5}$ | $\mathbf{43.71}_{\pm10.6}$ | $\mathbf{33.01}_{\pm0.2}$ | $\mathbf{40.42}$ | $\mathbf{39.45}$ |
| RN50 | Oracle | - | $55.62_{\pm0.5}$ | - | $45.12_{\pm1.1}$ | - | $58.75_{\pm0.3}$ | - | $43.55_{\pm0.8}$ | - | 50.76 |
| | Validation | $43.95_{\pm7.6}$ | $49.08_{\pm3.5}$ | $\mathbf{36.60}_{\pm13.6}$ | $37.44_{\pm2.3}$ | $\mathbf{28.02}_{\pm8.6}$ | $\mathbf{56.12}_{\pm0.3}$ | $39.71_{\pm15.0}$ | $\mathbf{41.63}_{\pm0.5}$ | 37.07 | 46.07 |
| | NAC-ME | $\mathbf{48.28}_{\pm7.0}$ | $\mathbf{50.94}_{\pm2.5}$ | $34.07_{\pm15.4}$ | $\mathbf{40.93}_{\pm2.0}$ | $26.06_{\pm8.4}$ | $55.95_{\pm0.6}$ | $\mathbf{52.21}_{\pm15.1}$ | $40.59_{\pm0.9}$ | $\mathbf{40.16}$ | $\mathbf{47.10}$ |
| Vit-t16 | Oracle | - | $52.03_{\pm0.3}$ | - | $27.38_{\pm3.0}$ | - | $49.61_{\pm0.4}$ | - | $36.14_{\pm0.1}$ | - | 41.29 |
| | Validation | $21.24_{\pm11.8}$ | $43.51_{\pm2.8}$ | $13.15_{\pm4.0}$ | $\mathbf{20.85}_{\pm2.1}$ | $\mathbf{20.02}_{\pm18.6}$ | $46.55_{\pm0.1}$ | $36.44_{\pm14.2}$ | $34.20_{\pm0.7}$ | 22.71 | 36.28 |
| | NAC-ME | $\mathbf{21.65}_{\pm12.1}$ | $\mathbf{44.37}_{\pm3.3}$ | $\mathbf{15.77}_{\pm1.5}$ | $20.23_{\pm0.7}$ | $18.30_{\pm17.9}$ | $\mathbf{46.77}_{\pm0.2}$ | $\mathbf{37.34}_{\pm13.9}$ | $\mathbf{35.39}_{\pm0.5}$ | $\mathbf{23.26}$ | $\mathbf{36.69}$ |
| Vit-b16 | Oracle | - | $62.23_{\pm0.4}$ | - | $46.94_{\pm1.7}$ | - | $57.45_{\pm0.5}$ | - | $42.29_{\pm0.1}$ | - | 52.23 |
| | Validation | $-1.31_{\pm3.1}$ | $53.13_{\pm2.0}$ | $-16.91_{\pm13.4}$ | $36.78_{\pm2.2}$ | $-3.27_{\pm9.5}$ | $\mathbf{54.19}_{\pm0.2}$ | $\mathbf{25.16}_{\pm7.0}$ | $37.84_{\pm0.4}$ | 0.92 | 45.49 |
| | NAC-ME | $\mathbf{32.60}_{\pm11.5}$ | $\mathbf{58.98}_{\pm0.7}$ | $\mathbf{11.44}_{\pm19.7}$ | $\mathbf{40.48}_{\pm2.6}$ | $\mathbf{15.60}_{\pm19.7}$ | $53.63_{\pm0.6}$ | $21.24_{\pm2.7}$ | $\mathbf{38.35}_{\pm0.4}$ | $\mathbf{20.22}$ | $\mathbf{47.86}$ |

Table 26: OOD generalization results on TerraInc dataset (Beery et al., 2018). *Oracle* denotes the upper bound, which uses OOD test data to evaluate models. The training strategy is ERM (Vapnik, 1999). All scores are averaged over 3 random trials.

