# OpenReview forum: "Neuron Activation Coverage: Rethinking Out-of-distribution Detection and Generalization"
_ICLR.cc/2024/Conference — ICLR 2024 spotlight_

### Official Review · Reviewer_iXGU · 2023-10-17

**Soundness:** 2 fair
**Presentation:** 2 fair
**Contribution:** 2 fair
**Rating:** 5
**Confidence:** 4

**Summary:**

The study focuses on the issue of out-of-distribution (OOD) in neural networks by examining the activity of neurons. A notion termed as "neuron activation coverage" (NAC) is introduced to describe the operation of neurons under in-distribution (ID) and OOD data. The researchers utilize NAC to attain good OOD detection results and demonstrate a positive association between NAC and the capacity of model generalization. Models that are more robust are chosen through the NAC-based standard and it presents a higher correlation with OOD test outcomes compared to conventional validation standards.

**Strengths:**

1. The holistic approach of tackling both OOD detection and generalization issues concurrently is not only commendable but also crucial.

2. The meticulous evaluation, inclusive of visualization and ablation study, is praiseworthy. This level of precision in assessment bolsters the understanding of the research outcomes and solidifies the integrity and robustness of the findings.

**Weaknesses:**

1. The introduction contains some issues: The methods like ASH, React do not negatively impact the ID accuracy. The model can employ an unchanged classifier for the ID task, which only introduces a negligible computational cost.

2. The motivation is somewhat perplexing: The author suggests that OOD is more prone to activate neurons with lower activation frequency. However, this paper fails to provide empirical evidence to back up this assertion.

3. Also, I think that the confusion between OOD and ID images stems from the fact that OOD images can potentially activate the same neurons as ID images. How does the author's approach works in such a scenario? It would be beneficial for the author to include near-ood experiments [1] to enhance the comprehensiveness of the study.

4. The performance of NAC-UE, as depicted in Tables 1 and 2, is not consistent and competitive across all OOD datasets, indicating a considerable bias in its effectiveness on different OOD datasets. For example, when CIFAR100 is used as ID and Places365 as OOD, NAC-UE only yields a 73.05% AUROC and 73.57 FPR95.

5. The author has overlooked a discussion on a comparable study [2].

[1]  Fort, Stanislav, Jie Ren, and Balaji Lakshminarayanan. "Exploring the limits of out-of-distribution detection." Advances in Neural Information Processing Systems 34 (2021): 7068-7081.

[2] Bai, Haoyue, et al. "Feed two birds with one scone: Exploiting wild data for both out-of-distribution generalization and detection." International Conference on Machine Learning. PMLR, 2023.

**Questions:**

The absence of open-source code is a setback as it impedes the reproducibility and validation of the results presented in the study.

---

> ### Author Response · Authors · 2023-11-17
> **Thank you; justifications and new near-ood experiments [1/3]**
>
> Dear Reviewer,
>
> Thank you for dedicating your time and effort to reviewing our paper. We are glad that you find our approach holistic and evaluation meticulous. We carefully considered your suggestions and conducted additional experiments to improve our paper. Please find our point-to-point responses below.
>
> ---
>
> **Q1:** The introduction contains some issues: The methods like ASH, React do not negatively impact the ID accuracy. The model can employ an unchanged classifier for the ID task, which only introduces a negligible computational cost.
>
> **A1:** We apologize for the confusion. In practice, our claim that "ASH, ReAct decrease the model classification ability" is referred from the ASH paper [R1], where they clearly state that ReAct and their ASH methods lead to InD accuracy drops due to pruning. We would like to invite the reviewer to check **Figure 2 in ASH paper** [R1] for the details.
>
> [R1] Extremely Simple Activation Shaping for Out-of-Distribution Detection. ICLR, 2023. https://openreview.net/pdf?id=ndYXTEL6cZz
>
> Despite the above clarification, we do agree with the reviewer that using an unchanged classifier can be an efficient solution. However, it is also important to recognize that the success of **this solution hinges on the strict assumption that only later layers are utilized by neuron pruning**. This restricted perspective may largely limit the potential of neuron-based methods, since shallow layers also offer valuable information, as demonstrated in Table 4 of our paper. Therefore, it is imperative to approach this solution with caution and careful consideration.
>
> To further address your concern, **we have added a footnote in our Introduction** to clarify this point: "While it may be argued that maintaining neuron outputs for double-propagation preserves InD accuracy with low computational cost,  it relies on the assumption that only later layers are utilized in neuron pruning, thus undermining the potential of these neuron-based methods."
>
> Lastly, we would also like to remind the reviewer that our **NAC-UE significantly outperforms ReAct and ASH on three benchmarks**. For your convenience, we provide their averaged AUROC results below. The scores are sourced from our Table 1 and 2, where ReAct and ASH are officially implemented by OpenOOD.
>
> || ReAct     | ASH   | NAC-UE |
> |----|---|----|---|
> | CIFAR-10  | 90.42 | 78.49  | **94.60** |
> | CIFAR-100 | 80.39 | 80.58  | **86.98** |
> | ImageNet  | 89.68 | 73.65  | **94.22** |
>
> [To be continued]

---

> > ### Author Response · Authors · 2023-11-17
> > **Thank you; justifications and new near-ood experiments [2/3]**
> >
> > ---
> >
> > **Q2:** The motivation is somewhat perplexing: The author suggests that OOD is more prone to activate neurons with lower activation frequency. However, this paper fails to provide empirical evidence to back up this assertion.
> >
> > **A2:** We are sorry for the confusion. In practice, **our paper has already provided both qualitative and quantitative results** to support our motivation:
> >
> > 1) For qualitative results, we kindly request the reviewer to refer to our **Figure 5 (rightmost column)**. This figure effectively illustrates that when considering our defined neuron states, InD and OOD data elicit distinct activation patterns in neurons. Specifically, OOD data tends to activate neurons with states around 0.5, representing the "center" of the activation space. Instead, InD data primarily activates neurons located at the "corners" of the space. This contrasting behavior shows that OOD data potentially activate neuron states that are less frequently observed with InD data.
> >
> > 2) For quantitative results, we would invite the reviewer to refer to **Figure 8 in Appendix G.2**. In this figure, we present the distribution of averaged coverage scores w.r.t all neurons, which reveal that **OOD samples generally present lower coverage scores compared to InD samples**. Since a low coverage score signifies a high inconsistency in neuron behavior between test samples and InD training data (See Eq. (5)), we can further confirm that OOD data tend to provoke abnormal neuron behaviors in comparison to InD data.
> >
> > Finally, it is worth noting that there also are many existing studies revealing that InD and OOD trigger distinct activation patterns (e.g., [R2, R3]), which is consistent with our findings and supports the validity of our results.
> >
> > [R2] ReAct: Out-of-distribution Detection With Rectified Activations. NeurIPS, 2021\
> > [R3] Line: Out-of-distribution detection by leveraging important neurons. CVPR, 2023
> >
> > ---
> >
> > **Q3:** Also, I think that the confusion between OOD and ID images stems from the fact that OOD images can potentially activate the same neurons as ID images. How does the author's approach works in such a scenario?
> >
> > **A3:** We appreciate this valuable comment, and agree with the reviewer that it is possible that OOD images trigger the same neuron states as InD images. However, as a network often comprises numerous neurons, it is also often the case that some neuron activations are triggered differently by OOD data, as evidenced by our qualitative and quantitative results (explained in **A2**).
> >
> > To further address your concern, **we have conducted a series of experiments on near-ood situations**, similar to [1]. Specifically, we utilized a model trained on CIFAR-10 as the foundation, and evaluated OOD detection methods on near-ood datasets: CIFAR-100 and Tiny ImageNet. We carefully follow the evaluation protocol of OpenOOD, and present the results below. For the sake of brevity, we merely report the top-5 performed methods as well as ReAct and ASH. The detailed experiments and full results for 21 baselines will be included in Appendix G.
> >
> >
> > | Method | CIFAR-100 || TIN-IMGNET ||Average||
> > |---|---|---|---|---|---|---|
> > | | FPR↓  | AUROC↑  | FPR↓ | AUROC↑  | FPR↓   | AUROC↑  |
> > | ReAct | 67.40 ± 7.34 | 85.93 ± 0.83 | 59.71 ± 7.31 | 88.29 ± 0.44 | 63.56 ± 7.33 | 87.11 ± 0.61 |
> > | ASH  | 87.31 ± 2.06 | 74.11 ± 1.55 | 86.25 ± 1.58 | 76.44 ± 0.61 | 86.78 ± 1.82 | 75.27 ± 1.04 |
> > | TempScale | 55.81 ± 5.07 | 87.17 ± 0.40 | 46.11 ± 3.63 | 89.00 ± 0.23 | 50.96 ± 4.32 | 88.09 ± 0.31 |
> > | RMDS | 43.86 ± 3.49 | 88.83 ± 0.35 | 33.91 ± 1.39 | 90.76 ± 0.27 | 38.89 ± 2.39 | 89.80 ± 0.28 |
> > | VIM  | 49.19 ± 3.15 | 87.75 ± 0.28 | 40.49 ± 1.55 | 89.62 ± 0.33 | 44.84 ± 2.31 | 88.68 ± 0.28 |
> > | KNN  | 37.64 ± 0.31 | 89.73 ± 0.14 | 30.37 ± 0.65 | 91.56 ± 0.26 | 34.01 ± 0.38 | 90.64 ± 0.20 |
> > | GEN| 58.75 ± 3.97 | 87.21 ± 0.36 | 48.59 ± 2.34 | 89.20 ± 0.25 | 53.67 ± 3.14 | 88.20 ± 0.30 |
> > | NAC-UE  | **35.06** ± 0.30 | **89.78** ± 0.31 | **26.53** ± 0.21 | **91.98** ± 0.24 | **30.80** ± 0.13 | **90.88** ± 0.25 |
> >
> > From the above results, we can see that **NAC-UE still outperforms existing SoTA methods on the near-ood benchmark**. This further confirms the effectiveness and robustness of our proposed approach.
> >
> > [1] Exploring the limits of out-of-distribution detection. NeurIPS, 2021.
> >
> > [To be continued]

---

> > > ### Author Response · Authors · 2023-11-17
> > > **Thank you; justifications and new near-ood experiments [3/3]**
> > >
> > > ---
> > >
> > > **Q4:** The performance of NAC-UE, as depicted in Tables 1 and 2, is not consistent and competitive across all OOD datasets, indicating a considerable bias in its effectiveness on different OOD datasets.
> > >
> > > **A4:**
> > > **There is no single winner on all OOD datasets**, as evidenced by [R8]. More crucially, most previously published methods (e.g., [R4-R7]) do exhibit limitations on one or several OOD datasets as well.
> > >
> > > We want to emphasize again that our NAC method is effective by surpassing 21 OOD detection baselines across three benchmarks. This has showcased its stronger ability compared to existing OOD detection methods, not to mention its potential in evaluating model robustness for OOD generalization. By considering a broader range of previously published works, we hope the reviewer could recognize the superiority of our work.
> > >
> > > [R4] GEN: Pushing the Limits of Softmax-Based Out-of-Distribution Detection. CVPR, 2023\
> > > [R5] MOS: Towards Scaling Out-of-Distribution Detection for Large Semantic Space. CVPR, 2021\
> > > [R6] How to Exploit Hyperspherical Embeddings for Out-of-Distribution Detection? ICLR, 2023\
> > > [R7] ViM: Out-Of-Distribution with Virtual-logit Matching. CVPR, 2022\
> > > [R8] OpenOOD v1.5: Enhanced Benchmark for Out-of-Distribution Detection. Arxiv, 2023
> > >
> > > ---
> > >
> > > **Q5:** The author has overlooked a discussion on a comparable study [2].
> > >
> > > **A5:** We thank the reviewer for this valuable comment. We will cite the paper and add a discussion section in Appendix H.
> > >
> > > While our NAC and [2] both focus on OOD detection and generalization, **they are actually different in their targets, design choices, and experimental settings**.
> > >
> > > 1) **Target**: Our NAC aims to provide an off-the-shelf/post-hoc tool that efficiently detects OOD data and evaluates model robustness. In contrast, [2] targets an effective learning strategy, which trains the network to overcome OOD scenarios.
> > >
> > > 2) **Design**: NAC directly leverages neuron distributions to reflect model status under OOD scenarios, while [2] enforces energy margin during the training phase.
> > >
> > > 3) **Experimental setup**:  Our paper focuses on the prevalent OOD detection and generalization setup, where the InD and OOD data are clearly separated. Instead, [2] centers on the wild scenarios, where data distribution is a mixed version of InD and OOD, turning the OOD into valuable learning resources.
> > >
> > > [2] Feed two birds with one scone: Exploiting wild data for both out-of-distribution generalization and detection. ICML, 2023.
> > >
> > > ---
> > >
> > > **Q6:** The absence of open-source code is a setback.
> > >
> > > **A6:** We are sorry for the confusion. As illustrated in Appendix F, **we will release our code with detailed instructions**. We also have outlined the hyperparameter choices under different datasets to promote reproducibility (Appendix D.4 and E.4).

---

> > > > ### Comment · Reviewer_iXGU · 2023-11-22
> > > >
> > > > Thank you to the author for the detailed response, which has cleared up most of my doubts, though some questions remain:
> > > > * The use of an unchanged classifier for ID classification tasks is clearly stated in DICE and ReAct's original papers. I also notice the expression in ASH Figure 2. However, in reading various post-hoc OOD detection method articles, it's common to find claims that one objective of OOD detection is to maintain ID classification performance unchanged [1]. Hence, I suggest the author should be cautious about considering a decline problem in ID classification accuracy as the motivation.
> > > > * As for the performance issue, I suggest the author provide explanations for the poor performance on certain OOD datasets. Notably, NAC evidently excels in small-scale datasets (like CIFAR) compared to the large-scale benchmarks, potentially indicating some limitations of the NAC method. Additionally, when using CIFAR100 as ID and Places365 as OOD, NAC-UE achieves only 73.05% AUROC and 73.57 FPR95. It would be beneficial to explain why NAC-UE underperforms on Places365.
> > > >
> > > > Overall, I will raise my score.
> > > >
> > > > [1] Yang, Jingkang, et al. "Generalized out-of-distribution detection: A survey." arXiv preprint arXiv:2110.11334 (2021).

---

> > > > > ### Author Response · Authors · 2023-11-22
> > > > > **Thank you for the reply**
> > > > >
> > > > > Thank you for the valuable comments. We provide the point-to-point responses below.
> > > > >
> > > > > > The use of an unchanged classifier for ID classification tasks is clearly stated in DICE and ReAct's original papers. I also notice the expression in ASH Figure 2. However, in reading various post-hoc OOD detection method articles, it's common to find claims that one objective of OOD detection is to maintain ID classification performance unchanged [1]. Hence, I suggest the author should be cautious about considering a decline problem in ID classification accuracy as the motivation.
> > > > >
> > > > > We appreciate this comment. As mentioned in our previous response (**A1**), we agree with the reviewer that using an unchanged classifier can be an efficient solution. But it is also important to acknowledge that **this solution hinges on the strict assumption that only later layers are involved in neuron pruning**. This restricted assumption can largely limit the potential of these pruning methods, since shallow layers also offer valuable information (Table 4 in our paper). **We have added a footnote in the Introduction to clarify this point**.
> > > > >
> > > > > Besides, we also want to emphasize again that **even though these pruning methods could preserve InD classification accuracy, our NAC still establishes its superiority by significantly outperforming them across three benchmarks** (results are summarized in **A1**). Additionally, our NAC stands out from traditional neuron pruning methods, given that it evaluates model status using natural neuron distributions, thus introducing a fresh perspective to OOD detection.
> > > > > We sincerely hope that the reviewer could recognize the novelty of our method and consider its contributions accordingly.
> > > > >
> > > > >
> > > > > > As for the performance issue, I suggest the author provide explanations for the poor performance on certain OOD datasets. Notably, NAC evidently excels in small-scale datasets (like CIFAR) compared to the large-scale benchmarks, potentially indicating some limitations of the NAC method. Additionally, when using CIFAR100 as ID and Places365 as OOD, NAC-UE achieves only 73.05% AUROC and 73.57 FPR95. It would be beneficial to explain why NAC-UE underperforms on Places365.
> > > > >
> > > > >
> > > > > Thank you for this comment. We acknowledge that our NAC-UE exhibits suboptimal performance on Places365 over the CIFAR-100. However, it is necessary to highlight that when considering the overall statistics (i.e., 3 benchmarks with a total of 11 results across 7 datasets), our method showcases strong performance. Specifically, **aside from our record-breaking average performance, our NAC-UE still consistently ranks top-3 across 21 SoTA methods, in 10 out of 11 individual results.**
> > > > >
> > > > > It is worth noting that the limitations on one or more OOD datasets are not unique to our approach. **Even highly-regarded papers, such as [1-5], have also displayed similar limitations.** This seems unfair if the reviewer dismisses our approach just based on such an isolated observation.
> > > > >
> > > > > On the other hand, we also agree with the reviewer that NAC shows higher improvements on CIFAR compared to ImageNet. We conjecture that **this phenomenon can be attributed to an intrinsic model bias**, where the model generally performs poorly on the challenging ImageNet dataset. For example, the InD accuracy of the model on CIFAR-10 is 95.06, whereas the accuracy over ImageNet is 76.18. This poor performance on ImageNet indicates the worse learning of models, thus potentially raising unstable behaviors in neurons and impacting the performance of our NAC-UE. This also explains the performance gap of NAC-UE on Places365 between CIFAR-10 and CIFAR-100. Since the model trained on CIFAR-100 achieves only 77.25 accuracy, it leads to higher neuron instability and subsequently affects the performance of NAC-UE. To address your concern, **we have added a discussion in Appendix H to explain this point**.
> > > > >
> > > > >
> > > > > [1] GEN: Pushing the Limits of Softmax-Based Out-of-Distribution Detection. CVPR, 2023\
> > > > > [2] MOS: Towards Scaling Out-of-Distribution Detection for Large Semantic Space. CVPR, 2021\
> > > > > [3] How to Exploit Hyperspherical Embeddings for Out-of-Distribution Detection? ICLR, 2023\
> > > > > [4] ViM: Out-Of-Distribution with Virtual-logit Matching. CVPR, 2022\
> > > > > [5] Extremely Simple Activation Shaping for Out-of-Distribution Detection. ICLR, 2023

---

> > > > > > ### Author Response · Authors · 2023-11-23
> > > > > > **Thank you; invitation to reply**
> > > > > >
> > > > > > Dear reviewer,
> > > > > >
> > > > > > We sincerely appreciate your dedicated time and effort in reviewing our paper. As we near the end of the discussion period, we would like to take this opportunity to further engage with you.
> > > > > >
> > > > > > In response to your valuable feedback, we have carefully provided clarifications in the Introduction (i.e., a **footnote** to clarify the confusion surrounding the neuron pruning methods), and included a detailed **discussion** in Appendix H (i.e., explaining the improvement on CIFAR benchmarks and the performance on Places365).
> > > > > >
> > > > > > We are looking forward to your feedback on these revisions and kindly request your consideration in revising the scores accordingly. Your continued engagement and support are immensely appreciated.

---

### Official Review · Reviewer_sJ3q · 2023-10-31

**Soundness:** 3 good
**Presentation:** 3 good
**Contribution:** 3 good
**Rating:** 8
**Confidence:** 3

**Summary:**

This paper propose a new metric named neuron activation coverage (NAC) to quantify the out-of-distribution. It uses neuron behaviors to quantify data distribution. It extends from original raw neuron output, models the neuron influence with combine neuron output with gradient from KL divergence of network output and a uniform vector. It is a simple metric while achieve SOTA results compared to previous methods on three benchmarks. It also shows correlation with generation, and could be used as a metric to quantify the robustness of model, and also correlate with OOD test performance.

**Strengths:**

Method:

1. Combine neuron output with gradient from KL divergence of network output and a uniform vector is a novel contribution and well-motivated approach, and it is simple while shows effective results.
2. It preserves more information in a continuous fashion, without discretizing it to become binary values compared to previous approach.
3. The measurement is further extended for uncertainty estimation for test samples with using average of all neurons.

Evaluation:
1. The method demonstrates its soundness on multiple benchmarks and outperforms existing SOTA.
2. Multiple ablation studies are performed.
3. Guidance and decisions of hyper parameters choice is discussed.

Writing:
1. The paper is well-organized, and the presentation is in good structure.

**Weaknesses:**

Time complexity:

1. Given the time-consuming process of using KL gradient, computational cost of proposed method with other SOTA should be reported.

Evaluation:

1. The proposed metric is extended for uncertainty estimation, while no quantification or evaluation (calibration of uncertainty) on this dimension.

**Questions:**

1. How does the choice of rank correlation instead Pearson correlation might affect the results?

2. How does this metric generalized to regression problems?

---

> ### Author Response · Authors · 2023-11-17
> **Thank you; new experiments and explanations [1/2]**
>
> Dear Reviewer,
>
> Thank you for dedicating your time and effort to reviewing our paper. We sincerely appreciate your positive remarks regarding the writing, novelty of the method, motivation, and robustness of our evaluations. We carefully considered your suggestions and conducted additional experiments to improve our paper. Please find our point-to-point responses below.
>
> ---
>
> **Q1:** Given the time-consuming process of using KL gradient, computational cost of proposed method with other SOTA should be reported.
>
> **A1:** Thanks for your valuable comment! Following your suggestion, we have conducted a series of experiments to analyze the efficiency of our method. To provide a comprehensive comparison, we selected the top-3 performed methods from Table 2 as baselines, and compared them with our NAC-UE in terms of preprocessing and inference time on ImageNet. For consistency, we used the same software and hardware configurations as in our previous experiments, which are detailed in Appendix F.
>
> || Preprocessing Time (s)| Total Inference Time (s) | AUROC↑|
> |---|---|---|---|
> | GEN *(CVPR' 2023)* | 0.00 ± 0.0 | 43.33 ± 0.3	| 89.76 |
> | ViM *(CVPR' 2022)*  | 1087.82 ± 9.0 | 48.10 ± 0.4 | 92.68 |
> | SHE *(ICLR' 2023)* | 1019.34 ± 2.2 | 41.85 ± 0.5 | 90.92 |
> | NAC-UE (layer4) | 5.43 ± 0.3 | **39.63** ± 0.2 | 94.57 |
> | NAC-UE (layer4+layer3)  | 6.75 ± 0.3  | 46.09 ± 0.7 | 95.05 |
> | NAC-UE (layer4+layer3+layer2) | 7.75 ± 0.2  | 69.73 ± 0.4 | **95.23** |
>
>
> From the above results, the following observations can be made:
>
> 1) While NAC-UE requires more inference time as more layers are considered, **it significantly reduces preprocessing time compared to ViM and SHE**, e.g., 7.75s (NAC-UE) vs. 1019.34s (ViM). This finding coincides with our previous experiments (Appendix G.1), where we show that NAC-UE achieves favorable performance despite utilizing only 1% of the training data for NAC modeling.
>
> 2) Remarkably, even when utilizing just a single layer (layer4), **NAC-UE outperforms SoTA methods in terms of both inference time and detection performance**. For example, NAC-UE achieves an AUROC of 94.57 (39.63s inference time), whereas GEN achieves only 89.76 on AUROC with 43.33s inference time. This highlights the efficiency of our approach.
>
> Additionally, it is also worth noting that there are numerous ongoing research efforts dedicated to facilitating gradient calculation [R1, R2], which could potentially complement our proposed method. We will include the above experiments in Appendix G.
>
> [R1] Low Complexity Gradient Computation Techniques to Accelerate Deep Neural Network Training, TNNLS, 2021.\
> [R2] Acceleration of DNN Backward Propagation by Selective Computation of Gradients, DAC, 2019.
>
> ---
>
> **Q2:** The proposed metric is extended for uncertainty estimation, while no quantification or evaluation (calibration of uncertainty) on this dimension.
>
> **A2:** We are so sorry for the confusion. In line with many prevalent OOD uncertainty estimators (e.g., Gram [R3], ReAct [R4], MOS [R5]), our uncertainty estimation mainly serves for OOD detection problems: _predicting whether a test sample is from OOD or not_. Thus, our experiments primarily emphasize OOD detection performance rather than calibration errors, similar to the aforementioned studies.
>
> In response to your suggestions, **we have conducted additional experiments to further explore the potential of our NAC-UE**. Our experiments carefully align with [R6], where two calibration error measures (RMS and MAD) are employed. For the calibration evaluation, we utilized a pretrained model on the CIFAR-10 dataset as the foundation, and assessed the calibration errors on both InD and OOD test data. We mainly compare NAC-UE with two simple baselines: MSP (Hendrycks & Gimpel, 2017) and Temperature (Guo et al., 2017).
>
> |OOD Dataset| | RMS↓ ||| MAD↓ ||
> |---|---|---|---|---|---|---|
> |  | MSP| Temp | NAC-UE | MSP | Temp | NAC-UE |
> | CIFAR100| 50.62 | 43.01 | **33.04** | 42.56| 36.64  | **26.92** |
> | Tiny ImageNet | 48.01 | 40.25 | **31.99** | 38.86| 32.88  | **26.25** |
> | MNIST | 71.74  | 60.91| **51.30** | 67.81 | 57.16  | **49.45** |
> | SVHN  | 65.82 | 56.41| **45.32** | 59.57 | 51.05  | **41.60** |
> | Texture | 42.65 | 35.19 | **28.74** | 32.37| 26.90  | **23.72** |
> | Places365 | 68.85 | 59.67  | **48.65** | 64.65  | 56.02  | **45.33** |
>
> From the above results, we can see that our NAC-UE significantly outperforms two baseline approaches, which demonstrates its potential in prediction calibration. We will include the above experiments in Appendix G.
>
> [R3] On the Importance of Gradients for Detecting Distributional Shifts in the Wild. NeurIPS, 2021\
> [R4] ReAct: Out-of-distribution Detection With Rectified Activations. NeurIPS, 2021\
> [R5] MOS: Towards Scaling Out-of-Distribution Detection for Large Semantic Space. CVPR, 2021\
> [R6] Deep Anomaly Detection with Outlier Exposure. ICLR, 2019
>
> [To be continued]

---

> ### Author Response · Authors · 2023-11-17
> **Thank you; new experiments and explanations [2/2]**
>
> ---
>
> **Q3:** How does the choice of rank correlation instead Pearson correlation might affect the results?
>
> **A3:** Great question! The choice between rank correlation (RC) and Pearson correlation (PC) can have an impact on the results. More specifically, as supported by [R7], **RC is more suited to our situations**.
>
> RC measures the **monotonic relationship** between two variables, while PC assesses the **linear relationship**. In our situations where the target is to select the best model based on InD evaluation criteria, the monotonic relationship becomes crucial [R7].
> To gain a clearer understanding, let's consider the following example:
>
> A (Test Accuracy): [0, **0.7**, 0.6, 0.5];
> B ( Val. Accuracy): [0, 0.8, 0.8, **0.9**],
>
> Here, sets A and B consist of test and validation accuracy at each epoch, respectively. PC between A and B is 0.93, while RC is 0.3.
> If we rely on PC, we might conclude that validation accuracy is highly correlated with test performance, and safely choose the best model with the highest validation accuracy (0.9). However, such a model only achieves a test accuracy of 0.5, contradicting our expectations. This example highlights the limitation of PC in our scenario, where the linear relationship could be misleading.
>
> In addition to the above toy example, **we have also conducted an analysis by measuring PC on Vit-b16 model**. As exhibited below, results mostly show surprisingly high PC values, i.e., 0.98/0.99 on three datasets. This contrasts sharply with our RC results in Table 8, where RC is hardly larger than 0.5. As the performance gap between the oracle-selected and validation-selected/NAC-selected model consistently exists, this again leads to the deduction that Pearson correlation is not as suitable as rank correlation in our situations.
>
> || VLCS   | PACS  | OfficeHome | TerraInc |
> |----|----|----|----|----|
> | InD Val. vs. OOD Test ACC | 0.994 | 0.983      | 0.999          | 0.866 |
> | InD NAC-ME vs. OOD Test Acc    | 0.995 | 0.983      | 0.999          | 0.893 |
>
> [R7] Ensemble of Averages: Improving Model Selection and Boosting Performance in Domain Generalization. NeurIPS 2022.
>
> ---
>
> **Q4:** How does this metric generalized to regression problems?
>
> **A4:** Thank you for this insightful question. Our proposed metric NAC, which focuses on neuron distributions, can indeed be generalized to regression problems without any changes to its definition. The only update required is about the formulation of neuron states. Since the current formulation involves KL divergence which is not suitable for regression problems, we need to simplify the neuron states from $ z \odot g'(z) (p-u)$ to $ z \odot g'(z) $. This simplified version corresponds to the Input $\times$ Gradient explanation, which is also meaningful in explaining neuron behaviors and provides insights into the model status.

---

> > ### Comment · Reviewer_sJ3q · 2023-11-22
> >
> > Thanks for the authors' clarifications and efforts to add new experiment results. Most of my concerns are well addressed. I maintain my original recommendation of acceptance.

---

> > > ### Author Response · Authors · 2023-11-22
> > > **Thank you so much**
> > >
> > > Dear reviewer,
> > >
> > > We sincerely appreciate your recognition and valuable feedback! Your comment has greatly improved the quality of our paper. We are grateful for your time and effort to review our work. Thank you!

---

### Official Review · Reviewer_pybD · 2023-11-01

**Soundness:** 2 fair
**Presentation:** 2 fair
**Contribution:** 2 fair
**Rating:** 6
**Confidence:** 3

**Summary:**

The paper addresses the out-of-distribution (OOD) problem in neural networks, which occurs when the encountered data significantly deviates from the training data distribution (in-distribution, InD). The authors approach the OOD problem by studying neuron activation states, considering both the output of the neuron and its influence on model decisions. They introduce a measure called neuron activation coverage (NAC) to characterize the relationship between neurons and OOD issues under InD data.

By leveraging NAC, the authors demonstrate two key findings. Firstly, they show that InD and OOD inputs can be effectively separated based on neuron behavior. This separation greatly simplifies the OOD detection problem and outperforms 21 previous methods across three benchmark datasets (CIFAR10, CIFAR-100, and ImageNet-1K). Secondly, they observe a consistent positive correlation between NAC and model generalization ability across different architectures and datasets. This correlation enables the use of NAC as a criterion for evaluating model robustness. In comparison to prevalent InD validation criteria, the authors demonstrate that NAC not only selects more robust models but also exhibits a stronger correlation with OOD test performance.

Overall, the paper proposes a novel approach to the OOD problem by studying neuron activation states and introducing NAC as a measure. The findings highlight the effectiveness of NAC in separating InD and OOD inputs, as well as its potential for evaluating model robustness.

**Strengths:**

Novel approach: The paper introduces a fresh perspective on addressing the OOD problem by studying neuron activation states and proposing the concept of neuron activation coverage (NAC). This approach offers a new and innovative methodology for tackling the OOD problem, contributing to the advancement of the field.

Effective separation of InD and OOD inputs: The paper demonstrates that the proposed neuron behavior-based approach, leveraging NAC, can effectively separate InD and OOD inputs. This separation significantly eases the OOD detection problem and outperforms 21 previous methods across multiple benchmark datasets. The ability to accurately distinguish between InD and OOD inputs is crucial for improving the reliability and robustness of neural networks.

Consistent correlation with model generalization ability: The authors observe a consistent positive correlation between NAC and model generalization ability across different architectures and datasets. This correlation indicates that NAC can serve as a reliable criterion for evaluating model robustness. By considering neuron activation behaviors, NAC provides valuable insights into the model's ability to generalize well beyond the training distribution, facilitating OOD generation.

Practical applicability: The proposed approach holds promise for real-world applications. By focusing on neuron activation states, the method does not require modifications to the neural network architecture, making it easily implementable in existing systems. Furthermore, the demonstrated effectiveness of NAC in separating InD and OOD inputs across different benchmark datasets reinforces its potential for diverse application scenarios.

**Weaknesses:**

The definitions of OOD are different for OOD Detection and OOD Generalisation. From my view, the former focuses on the semantic shift and the latter focuses on the covariate shift. Such a difference may raise two problems. First, it seems better if the authors could discern OOD in detection and generalisation, with further discussion and detailed definition. Second, why the purposed method, i.e., NAC, can handle both of these two cases. Heuristically, the proposed method is motivated by previous works in OOD detection (i.e., using gradient information), so why it is also applicable for OOD generalization.

There are many dimensions that are useless, and even worse, components/directions [1], instead of individual elements, can model the contribution of model outputs in making prediction. It means that there may exist misleading information and redundant message when using NAC to compute the activated rates, of which the performance cannot be guaranteed, at least from my personal view.

When it comes to OOD generalization, the authors suggest use NAC-based learning objective to improve OOD performance. It seems that it just makes the embedding features to be sparse, which has been discussed in previous works in OOD generalization, such as [2]. Moreover, the gradient-based learning objective can be computational hard, since the second order gradients are needed to be computed in each optimisation step.

Detailed discussion about the hyper parameter setups should be given, especially for the hyper parameter tuning. More ablation studies should  be given to test the respective power of z \times g'(z) and p-u.

[1] Intriguing Properties of Neural Networks

[2] Sparse Invariant Risk Minimisation.

**Questions:**

please see the weaknesses above

---

> ### Author Response · Authors · 2023-11-17
> **Thank you; new experiments and justifications [1/2]**
>
> Dear Reviewer,
>
> Thank you for dedicating your time and effort to reviewing our paper. We are glad that you think our method is novel, effective, and practically applicable. We took your suggestions very carefully, and have conducted additional experiments to update our paper.
>
> **Clarification:** Before diving into detailed responses, we hope to firstly clarify one fundamental aspect of our NAC, which seems misunderstood by the reviewer. That is, **our NAC-based methods all operate in a post-hoc fashion, which do not involve any training process**. Specifically, for OOD generalization, our NAC-ME is designed to assess model robustness for the purpose of best model selection, rather than directly improving OOD performance through optimization.
>
> ---
>
> **Q1:** The definitions of OOD are different for OOD Detection and OOD Generalisation...... so why it is also applicable for OOD generalization.
>
> **A1:** Thank you for this insightful comment. The response to your valuable question can be three-fold:
>
> 1) **The overlap between OOD detection and generalization exists**: While we agree with the reviewer that OOD detection/generalization mainly centers on semantic shift/covariate shift, the overlap between these two problems indeed exists. Previous studies have also explored covariate shift for OOD detection [R1, R2, R6], as well as semantic shift for OOD generalization [R3, R4]. Given such situations, it seems **less suitable if we consider these two problems as entirely separate**. We will include a discussion section in Appendix H.
>
> 2) **NAC benefits from data-centric modeling (Reason 1)**: Our NAC method is rooted in a data-centric approach, leveraging the neuron distributions within InD training data to characterize model status. This data-centric modeling enables NAC to effectively capture the intrinsic patterns and characteristics of the model (i.e., from a neuron level), thus serving as an effective tool for model robustness evaluation (i.e., applicable to OOD generalization). This also aligns with the principles of network quality assessment in system testing [R5].
>
> 3) **Shallow to deep layers account for covariate and semantic shifts (Reason 2)**: As noted by [R6], shallow layers in models often closely correlate with the image style information (*covariate* level), while deep layers capture *semantic* information. Since our NAC often works by leveraging multiple layers spanning from shallow to deep, it naturally accounts for both covariate and semantic shifts.
>
> [R1] Exploring Covariate and Concept Shift for Detection and Calibration of Out-of-Distribution Data. NeurIPS workshop, 2021.\
> [R2] Unified Out-Of-Distribution Detection: A Model-Specific Perspective. ICCV, 2023\
> [R3] Overcoming Concept Shift in Domain-Aware Settings through Consolidated Internal Distributions. AAAI, 2023\
> [R4] NICO++: Towards Better Benchmarking for Domain Generalization. CVPR 2023\
> [R5] NPC: Neuron Path Coverage via Characterizing Decision Logic of Deep Neural Networks. ACM Trans. Softw. Eng. Methodol. 2022.\
> [R6] Full-Spectrum Out-of-Distribution Detection. IJCV, 2023.
>
> ---
>
> **Q2:** There are many dimensions that are useless, and even worse......there may exist misleading information and redundant message when using NAC to compute the activated rates, of which the performance cannot be guaranteed.....
>
> **A2:** We appreciate this valuable comment, and concur with the reviewer regarding the possible redundancy and noise in multiple dimensions/neurons. Nonetheless, it is still worth noting that there are also **numerous neurons that are meaningful** and can encode significant information [R7, R8]. This observation underscores the viability of neuron-based methods (which is also supported by our Figure 4 and 5).
>
> On the other hand, our **NAC method actually goes beyond considering individual neurons alone**. It models the neuron distributions on InD data, and leverages averaged statistics of neuron behavior to reflect model status. This also aligns with the widely adopted statistical method in many network testing methods (e.g., [R9, R10]), which allows for the modeling of more abundant information.
>
> Despite the above clarification, we do acknowledge that filtering noisy neurons in NAC-based methods can be a viable strategy to further improve the performance. But we still want to remind the reviewer that **our NAC method has already demonstrated strong results** on three benchmarks for OOD detection (i.e., outperforming 21 baselines), and four datasets for OOD generalization. This has showcased the effectiveness and robustness of our approach.
>
>
> [R7] Understanding the role of individual units in a deep neural network. PNAS, 2020\
> [R8] Interpreting Deep Visual Representations via Network Dissection. TPAMI, 2018\
> [R9] DeepGauge: Multi-Granularity Testing Criteria for Deep Learning Systems. ASE, 2018\
> [R10] DeepXplore: Automated Whitebox Testing of Deep Learning Systems. SOSP, 2017
>
> [To be continued]

---

> ### Author Response · Authors · 2023-11-17
> **Thank you; new experiments and justifications [2/2]**
>
> ---
>
> **Q3:** When it comes to OOD generalization, the authors suggest use NAC-based learning objective to improve OOD performance. It seems that it just makes the embedding features to be sparse, which has been discussed in previous works in OOD generalization, such as [2]. Moreover, the gradient-based learning objective can be computational hard, since the second order gradients are needed to be computed in each optimisation step.
>
> **A3:** We are so sorry for the confusion. As we clarified earlier, our NAC-based method specifically targets the evaluation of model robustness for OOD generalization, which actually does not involve optimization processes. Thus, **we do not need to calculate 2nd-order gradients**, which saves computational cost.
>
> On the other hand, NAC also differs from [2] in the following aspects:
>
> 1) As noted, unlike the approach in [2] which concentrates on refining model training, our NAC focuses on the evaluation of existing models, thus providing a different perspective.
>
> 2) Drawing parallels with system testing coverage criteria, **NAC tracks neuron behaviors in the whole network**. In contrast, feature sparsity, as addressed in [2], is primarily concerned with feature representation, specifically identifying areas where most features are zero or irrelevant. Hence, these two methods differ in their measurement and targets.
>
> We will cite [2] and add a discussion in Appendix H.
>
> [2] Sparse Invariant Risk Minimisation. ICML, 2022
>
>
> ---
>
> **Q4:** Detailed discussion about the hyper parameter setups should be given, especially for the hyper parameter tuning. More ablation studies should be given to test the respective power of z \times g'(z) and p-u.
>
> **A4:** Thank you for this valuable comment. In practice, our **Appendix D.4 and E.4 have already provided the details for all employed hyperparameters**.
>
> To further address your concern, **we have added more ablation studies** to analyze the respective power of z, g'(z), and p-u. We provided the results on ImageNet below. Firstly, it can be drawn that z\*g'(z)\*(p-u) does perform best among all variants of formulations, showing its superiority.
> Moreover, arbitrary combinations of z, g'(z), and p-u can lead to improvements compared to using a single component alone. For instance, utilizing z*(p-u) yields better performance than using either z or p-u in isolation.
> These findings suggest that all three components encode meaningful information in OOD scenarios, further supporting the rationale behind our proposed neuron states. We will include these experiments in Appendix G.
>
>
>  |Formulation | z | g'(z) | p-u | FPR95↓ | AUROC↑ |
> |---|---|:-:|---|---|---|
> | z | ✔ |  | |45.7  | 89.42 |
> | g'(z)   | | ✔  | |84.2  | 64.13 |
> | p-u  || | ✔|59.39 | 80.96 |
> | z*g'(z)  | ✔| ✔  | |43.43 | 88.9  |
> | g'(z)*(p-u) || ✔ | ✔  |49.29 | 87.85 |
> | z*(p-u) | ✔| | ✔|44.71 | 89.47 |
> | z\*g'(z)\*(p-u) | ✔ | ✔ | ✔  | **26.89** | **94.57** |

---

> > ### Comment · Reviewer_pybD · 2023-11-23
> > **Many thanks for the response**
> >
> > The authors more or less address my concerns, and I would like to raise my score to 6.

---

> > > ### Author Response · Authors · 2023-11-23
> > > **Thank you so much**
> > >
> > > Dear reviewer,
> > >
> > > We are glad to hear that our response helps address your concern. Your comment has greatly improved the quality of our paper. Thank you for your time and effort to review our work!

---

> ### Comment · Area_Chair_TDdM · 2023-11-21
> **Please provide further feedback**
>
> Dear reviewer,
>
> It'd be helpful if you could provide feedback and engage with the authors further. Let me know if you need any help.
>
> AC

---

### Official Review · Reviewer_xpKd · 2023-11-01

**Soundness:** 3 good
**Presentation:** 4 excellent
**Contribution:** 3 good
**Rating:** 8
**Confidence:** 3

**Summary:**

This paper introduces the concept of neuron activation coverage (NAC) to characterize neuron behaviors under in-distribution (InD) and out-of-distribution (OOD) data. The key idea is to quantify the coverage degree of neuron activation states using training data, such that rarely covered states likely contain defects that could lead to poor OOD performance. The authors formulate neuron states by combining raw outputs and KL divergence gradients, and model coverage through a probability density function. NAC is applied to improve OOD detection via uncertainty estimation (NAC-UE), and OOD generalization through model evaluation (NAC-ME). Experiments demonstrate state-of-the-art OOD detection results on CIFAR and ImageNet benchmarks. NAC-ME also shows favorable correlation with OOD test accuracy for model selection.

**Strengths:**

* Novel perspective of leveraging neuron coverage to reflect OOD issues. This is intuitive and well-motivated through the analogy to software testing.
* Simple yet effective formulation of neuron activation states and coverage function.
* Improvements over strong baselines across multiple datasets and tasks. NAC-UE sets new state-of-the-art results while preserving model accuracy.

**Weaknesses:**

* For NAC-UE uncertainty estimation, the paper averages coverage scores across layers when applying to multiple layers. Is it possible to weight the contribution of different layers especially given that earlier layers tend to be more general?
* Missing a related work that shares a similar motivation of leveraging neuron behaviors. [1] uses neural mean discrepancy of neuron activations for OOD detection. Discussing relations to such works could provide additional insights.

[1] Dong, Xin, et al. "Neural mean discrepancy for efficient out-of-distribution detection." Proceedings of the IEEE/CVF Conference on Computer Vision and Pattern Recognition. 2022

**Questions:**

* Is it possible to directly train or regularize models based on NAC to improve robustness. For example, one could minimize the entropy of NAC distributions to encourage balanced coverage. Evaluating such NAC-driven training schemes could further validate its usefulness.

---

> ### Author Response · Authors · 2023-11-17
> **Thank you; new experiments and discussions**
>
> Dear Reviewer,
>
> Thank you for dedicating your time and effort to reviewing our paper. We are glad that you think our method is well-motivated and presents strong improvements. We took your suggestions very seriously, and have conducted additional experiments to update our paper. Please see below for our point-to-point responses.
>
> ---
>
> **Q1:** For NAC-UE uncertainty estimation, the paper averages coverage scores across layers when applying to multiple layers. Is it possible to weight the contribution of different layers especially given that earlier layers tend to be more general?
>
> **A1:** Nice suggestion! Inspired by your question, we have conducted a series of experiments on the CIFAR-10 benchmark, where we randomly searched the weight for each layer within the same space: [0.2, 0.4, 0.6, 0.8, 1.0]. We found that our **NAC-UE can be further improved in this weighted version** (e.g., 2.47% gain on average FPR):
>
> |  |   MNIST  || SVHN         || Textures     || Places365    || Average  ||
> |------|---|--------|------|------|--------|-----|-----|-------|----|-----|
> |                       | FPR↓               | AUROC↑          | FPR↓          | AUROC↑          | FPR↓         | AUROC↑          | FPR↓          | AUROC↑          | FPR↓          | AUROC↑          |
> | NAC-UE            | 15.14 ± 2.60 | 94.86 ± 1.36 | 14.33 ± 1.24 | 96.05 ± 0.47 | 17.03 ± 0.59 | 95.64 ± 0.44 | 26.73 ± 0.80 | 91.85 ± 0.28 | 18.31 ± 0.92 | 94.60 ± 0.50 |
> | NAC-UE (weighted) | 13.94 ± 2.42 | 95.55 ± 1.08 | 9.90 ± 1.09  | 98.10 ± 0.18 | 13.36 ± 0.65 | 97.25 ± 0.21 | 26.16 ± 0.79 | 92.31 ± 0.26 | 15.84 ± 0.65 | 95.80 ± 0.24 |
>
> What's interesting is that we noticed assigning larger weights to the deeper layers often results in better performance. We conjecture this is due to that deeper layers often encode richer semantic information than shallow layers, making them crucial in detection problems. We will include such experiments in Appendix G.
>
> ---
>
> **Q2:** Missing a related work that shares a similar motivation of leveraging neuron behaviors. [1] uses neural mean discrepancy of neuron activations for OOD detection. Discussing relations to such works could provide additional insights.
>
> **A2:** Thank you for pointing out the missed related work. We will cite the paper and add a discussion section in Appendix H. Here are the main differences outlined:
> 1)  [1] primarily investigates the raw neuron output, while **our paper centers on a new formulation of neuron states**. This formulation can be decoupled as the neuron gradients, neuron output, and model prediction deviations, thus offering a fresh interpretation of neurons in OOD scenarios.
> 2) Our NAC specifically focuses on the **distribution** of neuron states, while [1] mainly examines the **mean** of neuron output. This distinctive perspective makes our NAC more comprehensive and superior in understanding neuron behaviors.
> 3) While Neural Mean Discrepancy [1] may effectively detect OOD samples, it requires an additional classifier during the inference phase. Instead, NAC directly calculates the coverage scores in a **parameter-free** manner, serving as an efficient measure for both OOD detection and generalization.
>
> [1] Dong, Xin, et al. "Neural mean discrepancy for efficient out-of-distribution detection." CVPR, 2022
>
> ---
>
> **Q3:** Is it possible to directly train or regularize models based on NAC to improve robustness. For example, one could minimize the entropy of NAC distributions to encourage balanced coverage. Evaluating such NAC-driven training schemes could further validate its usefulness.
>
> **A3:** Yes, it is possible! Based on your suggestion, we have tried to regularize the model using our NAC entropy loss:
> $H(\mathbf{z})= -\sum_{i=1}^{N}p_i(z_i) \log{p_i(z_i)}$, where $p_i$ associates with the NAC distribution of $i$-th neuron.  We conducted experiments on the PACS dataset, and the results are shown below:
>
> || Art      | Cartoon     | Photo       | Sketch      | Average     |
> |----|---|----|-----|----|----|
> | ERM| 77.32 ± 0.7 | 71.91 ± 0.7 | 72.36 ± 1.1 | 94.44 ± 0.2 | 79.01 |
> | ERM+NAC (Minimizing Entropy)| 77.28 ± 0.2 | 69.17 ± 0.2 | 66.73 ± 1.2 | 93.21 ± 0.2 | 76.60 |
> | ERM+NAC (Maximizing Entropy)| 78.64 ± 0.5 | 72.97 ± 0.3 | 72.39 ± 0.3 | 95.09 ± 0.1 | 79.77 |
>
>
> Interestingly, **we find that maximizing NAC entropy leads to improved performance**. This finding also aligns with the intuitive understanding presented in [2]. By maximizing NAC entropy, we encourage the activation of neurons in unexplored regions over NAC distribution, thus diversifying the neuron activities and improving the model robustness. Conversely, minimizing entropy may result in collapsed neuron behavior. We will include the above experiments in Appendix G.
>
> [2] Dubey, Abhimanyu, et al. "Maximum-Entropy Fine-Grained Classification". NeurIPS 2018.

---

> > ### Comment · Reviewer_xpKd · 2023-11-22
> >
> > Thanks for the experiments and discussion from the authors.
> > It is interesting to see that NAC-UE can be further improved in weighted version and NAC entropy leads to improved performance.
> > These results make the work stronger.
> > I would like to raise my score.

---

> > > ### Author Response · Authors · 2023-11-22
> > > **Thank you so much!**
> > >
> > > Dear reviewer,
> > >
> > > Thank you for the recognition and valuable feedback! Your comment has greatly improved the quality of our paper. We appreciate your time and effort in reviewing our work.

---

### Author Response · Authors · 2023-11-19
**Summary of added experiments and updates**

We sincerely thank the reviewers for their time and effort in reviewing our paper. Taking their valuable feedback into consideration, we have made several additions to our work, which are **highlighted in red** in the revised draft. The major updates are as follows:

- **Computation cost analysis (Appendix G.1)**: we compared NAC with three top-performing baselines (SHE, ViM, and GEN) and demonstrated the efficiency of NAC in terms of preprocessing and inference. We appreciate the suggestions provided by *Reviewer sJ3q*.

- **Ablation studies on the respective power of  $\mathbf{z}$, $\nabla g(\mathbf{z})$, and $\mathbf{p}-\mathbf{u}$ (Appendix G.2)**, revealing that all these three components encode meaningful information and contribute to our defined neuron states. Thanks to the suggestions of *Reviewer pybD*.

- **Near-ood analysis (Appendix G.3)**: we analyzed NAC in the more challenging near-ood situations, demonstrating that NAC-UE continues to outperform 21 SoTA baseline methods. Thanks to the suggestions of *Reviewer iXGU*.

- **Weighted NAC for OOD detection (Appendix G.4)**: we implemented NAC with weighted layer combinations and observed further improvements in its performance. We appreciate the suggestions made by *Reviewer xpKd*.

- **Maximum NAC entropy for OOD generalization (Appendix G.5)**: we explored NAC entropy for model regularization and found that maximizing NAC entropy has the potential to enhance model performance. We thank *Reviewer xpKd* for this insightful suggestion.

- **Uncertainty calibration analysis (Appendix G.6)**: we included experiments on uncertainty calibration, where NAC outperforms two simple baselines and shows its potential. Thanks to the suggestions of *Reviewer sJ3q*.

- **Discussions (Appendix H)**: we dedicated this section to discuss the differences between NAC and SparseIRM [1] (*Reviewer pybD*), Neural Mean Discrepancy [2] (*Reviewer xpKd*), and SCONE [3] (*Reviewer iXGU*).
Additionally, we also offered explanations on the effectiveness of NAC in handling two OOD problems (*Reviewer pybD*), and discussed the inherent model bias that influences the performance of NAC-UE (*Reviewer iXGU*).

We also added **a footnote in the Introduction section**, which clarifies the argument surrounding the decrease in InD accuracy when using neuron pruning methods.  Thanks to the suggestions of *Reviewer iXGU*.

[1] Sparse Invariant Risk Minimisation. ICML, 2022\
[2] Neural mean discrepancy for efficient out-of-distribution detection. CVPR, 2022\
[3] Feed two birds with one scone: Exploiting wild data for both out-of-distribution generalization and detection. ICML, 2023.

---

### Meta-Review · Area_Chair_TDdM · 2023-12-09

**Metareview:**

This paper tackles the problem of out-of-distribution (OOD) detection and generalization with a new concept called concept of neuron activation coverage (NAC). The main idea of NAC (or of almost any neural activation based OOD detection method) is to define a measure of neuron behavior that effectively separates in-distribution (InD) and OOD data. The paper does so by combining raw outputs and KL divergence gradients, and a probability density function to quantify the "coverage degree" of "neuron activation states" with training (aka InD) data. Then the paper applies NAC in two avenues: 1) to improve OOD detection via uncertainty estimation (NAC-UE), and 2) to study OOD generalization through model evaluation (NAC-ME).

In terms of results, the authors demonstrate two key findings. Firstly, they show that InD and OOD inputs can be effectively separated by NAC. Experiments show that their method outperforms 21 previous methods across 3 benchmark datasets. Secondly, they observe a consistent positive correlation between NAC and model generalization ability across different architectures and datasets. This correlation enables the use of NAC as a metric to quantify the robustness of model, and also correlate with OOD test performance.

Overall, the paper proposes a novel approach to the OOD problem (both detection and generalization), provided comprehensive analyses and ablation studies, and demonstrated strong results. It is also well-written, doubling as a great literature survey for the OOD field of research.

**Justification For Why Not Higher Score:**

The study of neural activation behavior is not entirely novel — many previous methods have attempted to do so. The proposed method, while seemingly outperforming others, can be unnecessarily complicated.

**Justification For Why Not Lower Score:**

The proposed method is new. The experiment set up is very comprehensive, covering a great number of existing methods and benchmarks, with additional variants of model architecture. The results are also very strong, showing clear SoTA.

---

### Decision · Program_Chairs · 2024-01-16

Accept (spotlight)